# PTTA: Purifying Malicious Samples for Test-Time Model Adaptation

**Jing Ma** [* 1]   **Hanlin Li** [* 1]   **Xiang Xiang** [* 1 2 3]

## Abstract

Test-Time Adaptation (TTA) enables deep neural networks to adapt to arbitrary distributions during inference. Existing TTA algorithms generally tend to select benign samples that help achieve robust online prediction and stable self-training. Although malicious samples that would undermine the model's optimization should be filtered out, it also leads to a waste of test data. To alleviate this issue, we focus on how to make full use of the malicious test samples for TTA by transforming them into benign ones, and propose a plug-and-play method, PTTA. The core of our solution lies in the purification strategy, which retrieves benign samples having opposite effects on the objective function to perform Mixup with malicious samples, based on a saliency indicator for encoding benign and malicious data. This strategy results in effective utilization of the information in malicious samples and an improvement of the models' online test accuracy. In this way, we can directly apply the purification loss to existing TTA algorithms without the need to carefully adjust the sample selection threshold. Extensive experiments on four types of TTA tasks as well as classification, segmentation, and adversarial defense demonstrate the effectiveness of our method. Code is available at `https://github.com/HAIV-Lab/PTTA`.

## 1. Introduction

Test-Time Adaptation (TTA) is a way that goes beyond large-scale and heavy pre-training, and utilizes unlabeled test samples to enhance the online prediction performance

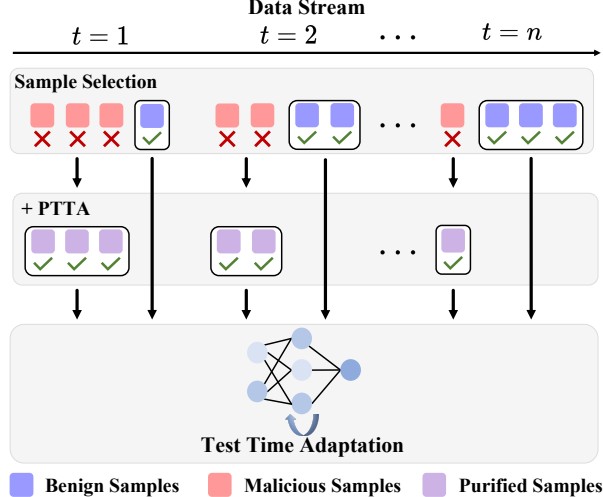

Figure 1: Illustration of purifying malicious test samples for test-time model adaptation using the proposed PTTA.

of pretrained models. TTA approaches break free from the curse of *i.i.d.* assumption and explore the adaptive potential of widely used models in unseen scenarios. A series of works have been proposed to achieve the stable and robust prediction of Deep Neural Networks (DNNs) in complex and dynamically changing environments (Wang et al., 2021; Niu et al., 2022a; Lee et al., 2024; Chen et al., 2024).

Since TTA algorithms seek to avoid human supervision and intervention, most of them adopt self-supervised tasks, such as Entropy Minimization (Wang et al., 2021) and Consistency Learning (Wang et al., 2022). This leads to the amplification of models' biases and catastrophic forgetting of original knowledge during continuous adaptation (Zhang et al., 2024; Niu et al., 2022a). In response to this challenge, recent studies have shifted their focus towards selecting benign samples useful for TTA algorithms from the test data (Niu et al., 2022a; Lee et al., 2024; Zhang et al., 2024). However, existing approaches incorporate sample selection criteria to filter and discard a portion of test samples that could otherwise be utilized. This practice results in data waste for TTA tasks, which grapple with data scarcity issues. Simultaneously, previous TTA methods encounter a trade-off between stable optimization and comprehensive use of test samples. This trade-off necessitates the cautious selection of hyperparameters and thresholds of the criteria.

---

[*] Equal contribution, co-first author. [1] National Key Lab of Multi-Spectral Info. Intelligent Processing Tech., School of Artificial Intelligence and Automation, Huazhong University of Science and Tech. (HUST), Wuhan, China. [2] Peng Cheng National Lab, Shenzhen, China. [3] School of Computer Science and Technology, HUST, China. Correspondence to: Xiang Xiang <xex@hust.edu.cn>.

*Proceedings of the 42ⁿᵈ International Conference on Machine Learning*, Vancouver, Canada. PMLR 267, 2025. Copyright 2025 by the author(s).

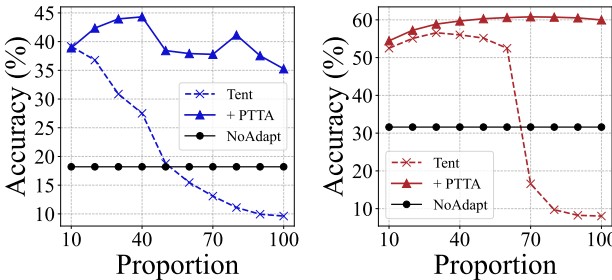

Figure 2: We vary the proportion of selected benign (low-entropy) samples and plot the test accuracy curve of ResNet50 (left) and ViT-B/16 (right).

To illustrate this issue, we conduct a pilot experiment to empirically demonstrate the deficiency of sample selection. Specifically, we employ ResNet50 and ViT-B/16 pretrained on ImageNet-1K to conduct experiments on continual TTA task constructed from ImageNet-C. The results, as shown in Fig. 2, reveal that the threshold for benign sample selection controls the testing accuracy of DNNs. A minor change in the threshold will lead to severe performance degradation, *e.g.*, 36% accuracy drop when proportion changes from 0.6 to 0.7 in Fig. 2 (right). High sensitivity to thresholds limits the scalability of existing TTA algorithms.

Although the filtered malicious samples[1] may undermine the stability of TTA algorithms, they nonetheless reflect the distribution of the test data. If it is possible to transform malicious samples into benign ones and subsequently make full use of them, the optimization efficiency can be enhanced. Motivated by adversarial purification (Tang & Zhang, 2024; Shi et al., 2021), we aim to utilize benign samples to purify the malicious ones. Instead of adding noise to the pixels, we retrieve for a malicious sample the counterpart that has an opposite impact on the objective function. We employ the simple Mixup technique (Zhang et al., 2018) to accomplish malicious sample purification, due to its advantages of improving the calibration of DNNs and reducing over-confident predictions. We also demonstrate the benefits of our proposed **P**urification for **T**est-**T**ime **A**daptation (PTTA) in Fig. 2. We find that the performance of existing TTA algorithm is significantly improved by PTTA, and its sensitivity to the threshold is reduced. In addition, the test accuracy of the model grows as the increase in the utilization rate of test samples, as shown in Fig. 2 (right).

In this paper, we propose a plug-and-play solution to purify malicious test samples. To accurately encode the impacts of benign and malicious samples on TTA, we propose a *saliency indicator*, which measures the contribution of an individual test sample to the objective function at the pixel,

feature, and logit levels. The saliency indicator is also an efficient retrieval metric for retrieving benign samples in a memory bank or within the mini-batch. Based on this indicator, we design a pipeline of *benign sample retrieval* and *malicious sample purification*, as shown in Fig. 1. Specifically, we employ existing sample selection criteria for TTA to distinguish benign and malicious samples (Niu et al., 2022a; Lee et al., 2024; Zhang et al., 2024). Then we retrieve the counterpart from benign samples that has the most opposite effect on the objective function compared to a malicious sample. Two matched samples are Mixuped to calculate the purification loss. In this way, we effectively use the malicious samples that should have been filtered out, thereby improving the utilization rate of test data. We demonstrate the effectiveness of PTTA and its universality for different DNNs through extensive experiments, including episodic, single, continual, and lifelong TTA tasks, as well as image classification, semantic segmentation, and adversarial defense tasks.

In summary: 1) We analyze the hazards of learning malicious test samples for TTA and propose a saliency indicator to encode benign and malicious samples effectively; 2) We put forward a plug-and-play solution, PTTA, to purify malicious samples by using retrieved benign samples that have the opposite impact on the objective function. It improves the utilization rate of test data and enhances models' test accuracy, while reducing sensitivity to the thresholds; and 3) We conduct extensive experiments to demonstrate the generality of PTTA across various tasks and models, as well as its expandability on existing TTA algorithms.

## 2. Preliminaries

**Notations.** Considering the $C$-way classification, we define the label space $Y \in \mathbb{R}^C$ and the sample space $X \in \mathbb{R}^m$. Following the assumptions of existing TTA methods, we assume that all unlabeled samples $x \in X$ arriving at test time could be categorized into one of the known classes in the label space, *i.e.*, $y(x) \in Y$. We define a Deep Neural Network $f$ and its parameters $\theta$. The TTA task follows the paradigm of online learning. That is, at any time $t$, only $N_{bs}$ unlabeled samples $\mathbf{x}_t = \{x_{ti}\}_{i=1}^{N_{bs}}$ are available. The predicted output of the model $f$ for sample $x$ is defined as logits $\mathbf{z} = f(x) = [z_1, ..., z_C]$. Generally, a Softmax function $\sigma(\cdot)$ is applied to compute the probability vector $\mathbf{p} = \sigma(f(x)) = [p_1, ..., p_C]$. A widely used approach is to decompose the DNN $f$ into a feature extractor $\phi$ and a classifier $\varphi$. The $n$-dimensional features extracted by $\phi$ are called latent code defined as $\mathbf{h} = \phi(x) = [h_1, ..., h_n]$.

**Entropy Minimization.** Shannon entropy (Shannon, 1948) is often used to measure the confidence of model predictions. Wang *et al.* (2021) establish a connection between entropy and the error rate, and show that more confident predictions

---

[1]Different from defining malicious samples in adversarial defense as those attacked by adding noise, we define malicious samples as those filtered out by sample selection criteria. Essentially, both definitions refer to the samples that undermine TTA methods.

are all-in-all correct. On top of this, Entropy Minimization is taken as the optimization objective function for TTA:

$$\min_{\theta} \; -\frac{1}{N_{bs}} \sum_{i=1}^{N_{bs}} \sum_{j=1}^{C} \mathbb{1}_{\{S(x_{ti}) > S_0\}} \cdot p_j(x_{ti}, \theta) \log p_j(x_{ti}, \theta),$$
(1)

where $t$ is the time step. Minimizing the conditional entropy reduces the overlap of output class distributions, leading to the low-density separation between classes.

**Sample Selection Criteria.** Not all test samples are helpful for TTA. Therefore, selecting informative benign samples is crucial, which is denoted as an indicator function $\mathbb{1}\{S(x_{ti}) > S_0\}$ for reweighting in Eq. 1, where $S(\cdot)$ is a scoring function for each sample, and $S_0$ is a preset threshold. A series of selection criteria has been proposed. ETA (Niu et al., 2022a) selects diverse samples with low entropy, *i.e.*, $\mathbb{1}\{\text{Ent}(x_{ti}) < \text{Ent}_0\} \cap \mathbb{1}\{\text{Cos}(f(x_{ti}), \mathbf{m}^{t-1}) < \epsilon\}$. DeYO (Lee et al., 2024) introduces the PLPD and add a criterion, *i.e.*, $\mathbb{1}\{\text{PLPD}(x_{ti}) > \tau\}$. CPL (Zhang et al., 2024) proposes the intra- and inter-instance label selection, *i.e.*, $\mathbb{1}\{j \in \text{MinSize}(\{j| \sum_{j=1}^{C} p_j(x_{ti}) \geq \tau\})\} \cup \mathbb{1}\{j \in \{j|p_j(x_{ti}) > \text{Quantile}(\text{Sort}(\mathbf{p}_j(\mathbf{x}_t)), \beta)\}_{j=1}^{C}\}$.
These sample selection criteria divide test data into benign and malicious samples, and then filter out the malicious ones, resulting in a data waste. In this paper, we will demonstrate how to use purification to turn malicious samples into benign ones so as to make full use of test data.

## 3. Method

### 3.1. Malicious Sample Hazards

Vanilla Test-Time Adaptation (TTA) algorithm treats all the test samples equally. However, according to the model's predictions, samples with high conditional entropy are less accurate than those with low entropy. We verify the relationship between the classification accuracy and the conditional entropy of ResNet50 and ViT-B/16 pre-trained on ImageNet-1K for 15 types of out-of-distribution samples in ImageNet-C, as shown in Fig. 3 (top). Applying Entropy Minimization for high-entropy samples does not bring benefits to TTA algorithms. Instead, it will undermine the stability of DNNs, especially when test data distribution has a large shift from the source training distribution. As shown in Fig. 3 (down), DNNs inherently have relatively high conditional entropy for test domains with low classification accuracy.

A simple solution is to sort the test samples (within a mini-batch) in ascending order according to the conditional entropy, and then take the top-$\mathfrak{p}\%$ low-entropy samples, called benign samples, to apply Entropy Minimization. This sample selection criterion can effectively improve the performance of vanilla TTA algorithm, as shown in Fig. 2. However, this success exhibits high sensitivity to thresholds (typ-

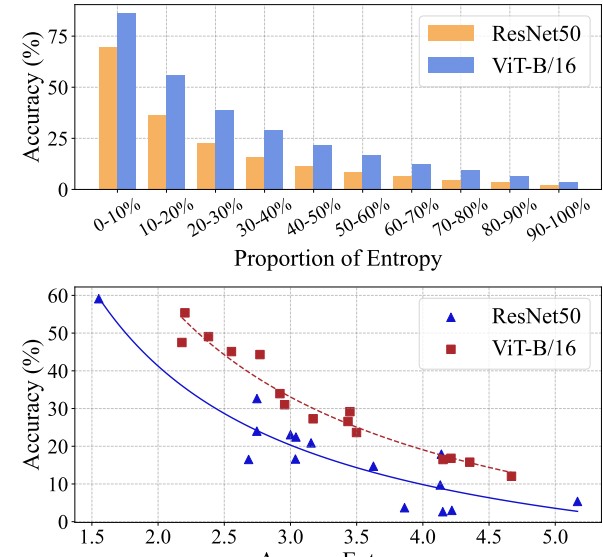

Figure 3: We test the accuracy of top-$\mathfrak{p}\%$ low-entropy samples (top), and visualize the relationship between the averaged prediction entropy and the test accuracy of DNNs for 15 types of out-of-distribution domains (down).

ically treated as hyperparameters) in the criterion, where slight variations lead to substantial performance degradation. Through these observations, it is intuitive to believe that malicious samples exist in an uncertain ratio within mini-batches, and existing sample selection criteria fail to completely eliminate them from test data. This constitutes another form of malicious sample hazards that fundamentally constrains the practical applicability of TTA algorithms.

Instead of designing criteria to detect and exclude malicious samples, we focus on how to transform malicious samples into benign ones. We name this transformation the *malicious sample purification* and propose PTTA, a plug-and-play method, to improve existing TTA algorithms for test data utilization. The pipeline is shown in Fig. 4.

### 3.2. Saliency Indicator

In this subsection, we introduce an indicator for encoding benign and malicious samples, motivated by adversarial purification. The concept of adversarial purification stems from adversarial defense (Tang & Zhang, 2024; Shi et al., 2021). The purpose of purification is to add noise $\gamma$ of a specified magnitude $\xi$ at the pixel level of an image $x$ to minimize the loss of the defense objective function, which is defined as $\gamma = \arg\min_{\|\gamma\|_p \leq \xi} \mathcal{L}(f_\theta(x + \gamma), y(x))$. The noise $\gamma$ is generally calculated as the partial derivative with respect to image $x$, *i.e.*, $\gamma = \xi \cdot \text{sign}(\nabla_x \mathcal{L}(f_\theta(x), y(x)))$. This first-order derivative $\nabla_x \mathcal{L}$, called saliency information, is used to judge how pixels in the image affect the model's final prediction, and it also indicates the importance of each pixel (Simonyan et al., 2014). Another interpolation is that

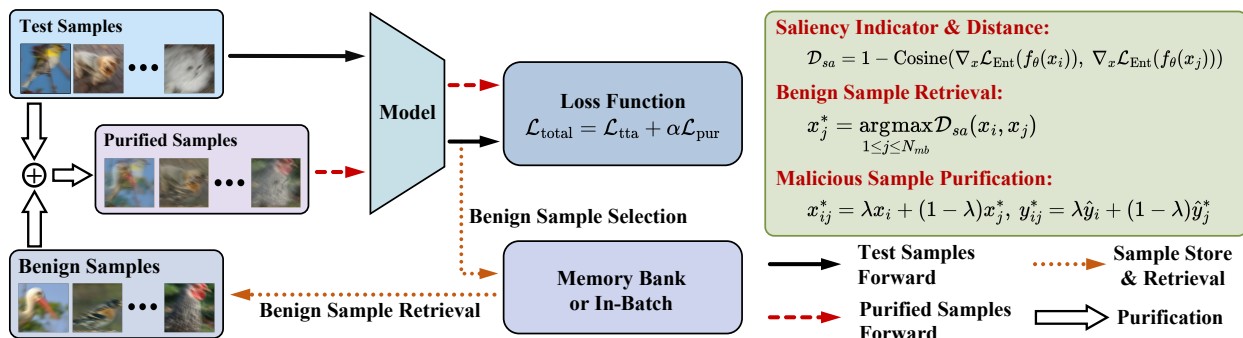

Figure 4: The pipeline of Purification for Test-Time Adaptation (PTTA). We purify malicious test samples using benign samples retrieved from a memory bank or within the mini-batch, based on the saliency indicator we proposed for encoding benign and malicious data. The purified samples are used to calculate the purification loss $\mathcal{L}_{\text{pur}}$, which is directly inserted into the original objective function $\mathcal{L}_{\text{tta}}$ of existing TTA methods.

it measures which individual pixels need to be changed the most in order to minimize the objective function. Consequently, samples contributing similarly to the objective function exhibit aligned directions of their saliency information. Therefore, the saliency information of Entropy Minimization (a widely adopted objective function in TTA methods) with respect to test sample $x$, *i.e.*, $\nabla_x \mathcal{L}_{\text{Ent}}(f_\theta(x))$, can be effectively used for purification. We further demonstrate the flexibility in choosing objective functions in Sec. 4.3.

However, instead of directly superimposing noises on pixels for purification, we utilize noises to encode the gradient direction of objective functions for test samples. Specifically, these unit noise vectors (with a magnitude of 1.0) indicate the direction in which the entropy of model's predictions decreases the fastest in the sample space. Since gradients corresponding to benign samples $x^+$ and those corresponding to malicious samples $x^-$ have opposite directions, *i.e.*,

$$\text{Cosine}(\nabla_x \mathcal{L}_{\text{Ent}}(f_\theta(x^+)), \ \nabla_x \mathcal{L}_{\text{Ent}}(f_\theta(x^-))) < 0. \quad (2)$$

Therefore, we employ the saliency information as an indicator for encoding benign and malicious samples. Meanwhile, we calculate the *cosine distance* of the saliency indicator to retrieve two samples $x_i$ and $x_j$ that have the most opposite contributions to objective functions as follows:

$$\mathcal{D}_{sa} = 1 - \text{Cosine}(\nabla_x \mathcal{L}_{\text{Ent}}(f_\theta(x_i)), \ \nabla_x \mathcal{L}_{\text{Ent}}(f_\theta(x_j))). \quad (3)$$

Besides using the gradients of pixels, the gradients of features and logits can also provide saliency information. We investigate these three indicators in the following.

*Pixel-saliency indicator* uses $\nabla_x \mathcal{L}_{\text{Ent}}$ as the saliency information. It encodes the contribution to objective functions in the high-dimensional sample space, which preserves spatial or temporal information. But it is vulnerable to pixels that are irrelevant to the model prediction.

*Feature-saliency indicator* exploits $\nabla_h \mathcal{L}_{\text{Ent}}$ as the saliency information. The feature extractor compressive encodes im-

ages, it reduces the interference of irrelevant pixels. Meanwhile, the features possess semantic information and are more closely related to the model's final decisions.

*Logit-saliency indicator* employs $\nabla_z \mathcal{L}_{\text{Ent}}$ as the saliency information. $\nabla_z \mathcal{L}_{\text{Ent}}$ can be directly calculated without the need of backpropagating gradients, *i.e.*,

$$\nabla_z \mathcal{L}_{\text{Ent}}(f_\theta(x)) = -(\mathbf{z} - \mathbf{p} \cdot \mathbf{z}) \odot \mathbf{p}. \quad (4)$$

Intuitively, the logit-level saliency represents the changes in the logits output of the model after optimization with respect to the sample $x$. It directly encodes the impact of test samples on the model's final decisions.

We conduct an ablation study on these three indicators, as shown in Table 1. The logit-saliency indicator generally exhibits better performance and lower computational overhead. Meanwhile, we also compare with adversarial purification methods, which directly superimpose noises to malicious samples via FGSM (Tang & Zhang, 2024) or SOAP (Shi et al., 2021). Adversarial purification methods fail to improve and even impair the performance of TTA algorithms.

### 3.3. Purification for Test-Time Adaptation

In this subsection, we elaborate on how to perform malicious sample purification and scale it as a plug-and-play method to existing TTA algorithms. Put it briefly, we mainly adopt the Mixup technique (Zhang et al., 2018) to merge benign and malicious samples, which serves as a data augmentation to make up for the discarding of malicious samples by sample-selection-based TTA algorithms. While simple to implement, Mixup has been shown to improve the calibration of DNNs, *i.e.*, the predicted Softmax probabilities can better indicate the actual likelihood of a correct prediction, and reduce the over-confident predictions of DNNs on out-of-distribution data (Thulasidasan et al., 2019).

**Benign Sample Retrieval.** Next, we focus on determining the retrieval scope and retrieving using saliency indicator.

Table 1: We demonstrate the generality of PTTA across various sample selection criteria. Meanwhile, we compare the performance of PTTA with different saliency indicators, types of retrieval, and sample purification scopes. We count and compare the number of forward (#Fwds) and backward (#Bwds) propagations for different methods.

| METHODS | SINGLE | | | | CONTINUAL | | | | #FWDS | #BWDS |
|---|---|---|---|---|---|---|---|---|---|---|
| | ETA | DeYO | CPL | AVG. | ETA | DeYO | CPL | AVG. | | |
| COMPARISON WITH EXISTING PURIFICATION (PURIF.) METHODS. | | | | | | | | | | |
| BASE ALGORITHM | $48.2_{\pm0.04}$ | $48.6_{\pm0.11}$ | $42.8_{\pm0.04}$ | 46.5 | $62.2_{\pm0.07}$ | $62.2_{\pm0.35}$ | $55.5_{\pm0.13}$ | 60.0 | $N_{batch}$ | $N_{batch}$ |
| + PURIF. W/ FGSM | $45.9_{\pm0.04}$ | $47.0_{\pm0.04}$ | $42.1_{\pm0.09}$ | 45.0 | $61.7_{\pm0.03}$ | $62.2_{\pm0.59}$ | $56.4_{\pm0.08}$ | 60.1 | $N_{batch} \times 2$ | $N_{batch} \times 2$ |
| + PURIF. W/ SOAP (5 ITER) | $47.4_{\pm1.07}$ | $47.7_{\pm0.25}$ | $36.6_{\pm8.80}$ | 43.9 | $57.0_{\pm7.43}$ | $64.1_{\pm0.09}$ | $49.1_{\pm3.22}$ | 56.7 | $N_{batch} \times 6$ | $N_{batch} \times 6$ |
| ABLATION STUDY OF DIFFERENT SALIENCY INDICATORS (IND.). | | | | | | | | | | |
| + PIXEL-SALI. IND. | $49.0_{\pm0.03}$ | $\mathbf{50.0}_{\pm0.09}$ | $43.0_{\pm0.06}$ | 47.3 | $\mathbf{65.6}_{\pm0.02}$ | $\mathbf{66.0}_{\pm0.01}$ | $\underline{59.3}_{\pm0.05}$ | $\mathbf{63.6}$ | $N_{batch} \times 2$ | $N_{batch} \times 2$ |
| + FEATURE-SALI. IND. | $\underline{49.1}_{\pm0.04}$ | $49.9_{\pm0.03}$ | $\underline{43.4}_{\pm0.06}$ | $\underline{47.5}$ | $65.4_{\pm0.05}$ | $65.8_{\pm0.02}$ | $58.9_{\pm0.06}$ | 63.3 | $N_{batch} \times 2$ | $N_{batch} \times 2$ |
| + LOGIT-SALI. IND. | $\mathbf{49.5}_{\pm0.00}$ | $\underline{49.9}_{\pm0.10}$ | $\mathbf{44.6}_{\pm0.06}$ | $\mathbf{48.0}$ | $65.3_{\pm0.06}$ | $65.8_{\pm0.01}$ | $\mathbf{59.6}_{\pm0.01}$ | $\underline{63.5}$ | $N_{batch} \times 2$ | $N_{batch}$ |
| ABLATION STUDY OF DIFFERENT TYPES OF RETRIEVAL (RET.). | | | | | | | | | | |
| + ID RET. ($N_{mb} = 1k$) | $48.2_{\pm0.25}$ | $48.3_{\pm0.22}$ | $43.1_{\pm0.18}$ | 46.5 | $64.4_{\pm0.10}$ | $63.3_{\pm0.49}$ | $58.2_{\pm0.08}$ | 62.0 | $N_{batch} \times 2$ | $N_{batch}$ |
| + OOD RET. ($N_{mb} = 1k$) | $\mathbf{49.5}_{\pm0.00}$ | $\underline{49.9}_{\pm0.10}$ | $\mathbf{44.6}_{\pm0.06}$ | $\mathbf{48.0}$ | $\underline{65.3}_{\pm0.06}$ | $\underline{65.8}_{\pm0.01}$ | $\mathbf{59.6}_{\pm0.01}$ | $\underline{63.5}$ | $N_{batch} \times 2$ | $N_{batch}$ |
| + IB RET. ($N_{mb} = 0$) | $\underline{49.3}_{\pm0.08}$ | $\mathbf{50.0}_{\pm0.04}$ | $43.6_{\pm0.06}$ | $\underline{47.6}$ | $\mathbf{65.5}_{\pm0.03}$ | $65.8_{\pm0.02}$ | $\underline{59.1}_{\pm0.06}$ | $\underline{63.5}$ | $N_{batch} \times 2$ | $N_{batch}$ |
| ABLATION STUDY OF DIFFERENT SAMPLE PURIFICATION SCOPES. | | | | | | | | | | |
| + PURIF. ALL SAMPS. | $\mathbf{49.5}_{\pm0.00}$ | $\mathbf{49.9}_{\pm0.10}$ | $\mathbf{44.6}_{\pm0.06}$ | $\mathbf{48.0}$ | $65.3_{\pm0.06}$ | $65.8_{\pm0.01}$ | $59.6_{\pm0.01}$ | $63.5$ | $N_{batch} \times 2$ | $N_{batch}$ |
| + PURIF. MALI. SAMPS. | $49.1_{\pm0.02}$ | $49.6_{\pm0.08}$ | $44.3_{\pm0.06}$ | 47.7 | $64.7_{\pm0.04}$ | $65.3_{\pm0.06}$ | $59.0_{\pm0.07}$ | 63.0 | $N_{batch} \times 2$ | $N_{batch}$ |

*In-distribution (ID) retrieval* suggests that the samples within source training distribution could be used as benign samples to purify out-of-distribution samples at test time. Specifically, we randomly choose $N_{mb}$ ID samples, stored in a fixed memory bank $\{x_j, \hat{y}_j, \nabla\mathcal{L}_{\text{Ent}}(x_j)\}_{j=1}^{N_{mb}}$, where their probabilities and saliency informations are computed offline. We then calculate the saliency distance $\mathcal{D}_{sa}(x_i, x_j)$ between a online test sample $x_i$ and ID samples, and select one ID sample with the largest distance to perform Mixup.

*Out-of-distribution (OOD) retrieval.* However, model $f_\theta$ will gradually adapt to the test data distribution in TTA tasks, while forgetting the source data (Niu et al., 2022a). At time step $t$, utilizing the benign test samples $\{\mathbf{x}_i^+\}_{i=t-a}^t$ within a historical period $[t-a,\ t]$ is beneficial to current model. Therefore, we define the memory bank as a first-in-first-out queue and update it with $\mathbf{x}_t^+$. We retrieve benign samples in the memory bank by using the saliency distance.

*In-batch (IB) retrieval* retrieves within benign samples $\mathbf{x}_t^+$ in the mini-batch at time step $t$ without an additional memory bank. One drawback is that it limits the retrieval scope.

We compare three retrieval methods in Table 1. Refer to Appendix E.2 for detailed ablation studies.

**Malicious Sample Purification.** For a test sample $x_i$, we retrieve a matched benign sample within a scope $\{x_j\}_{j=1}^{N_{mb}}$,

$$x_j^* = \underset{1 \le j \le N_{mb}}{\arg\max}\ \mathcal{D}_{sa}(x_i, x_j), \qquad (5)$$

where $x_j^*$ is employed to purify the test sample $x_i$. The purified sample $x_{ij}^*$ and its pseudo-label are obtained by using the Mixup technique (Zhang et al., 2018) as follows:

$$\begin{aligned} x_{ij}^* &= \lambda x_i + (1-\lambda)x_j^*, \\ y_{ij}^* &= \lambda \hat{y}_i + (1-\lambda)\hat{y}_j^*, \end{aligned} \qquad (6)$$

where $\hat{y}_i$ is the model's predicted probability for $x_i$. We set $\lambda = 1/(K+1)$, where $K$ decides that the top-$K$ samples with the largest saliency distance are retrieved from the scope. We uniformly set $K = 1$ and conduct an ablation study on the value of $K$ in Sec. 4.4.

Built upon the original objective function $\mathcal{L}_{\text{tta}}$ of existing TTA algorithms, we add a purification loss as

$$\mathcal{L}_{\text{pur}} = -\frac{1}{N_{bs}} \sum_i^{N_{bs}} \sum_c^C (y_{ij}^*)_c \log \sigma_c(f_\theta(x_{ij}^*)). \qquad (7)$$

Note that we purify *all test samples* in the mini-batch for $\mathcal{L}_{\text{pur}}$, considering the performance improvement brought by greater amounts of augmented data. We experimentally demonstrate that it benefits our proposed PTTA in Table 1.

Overall, the total loss function is defined as

$$\mathcal{L}_{\text{total}} = \mathcal{L}_{\text{tta}} + \alpha\mathcal{L}_{\text{pur}}, \qquad (8)$$

where $\alpha$ is a hyperparameter to balance the two loss functions, and we conduct an ablation study on $\alpha$ in Sec. 4.4.

## 4. Experiments

### 4.1. Setups

We describe the benchmarks used in the experiments, the baselines for comparison, and the implementation of methods in the following. Refer to Appendix C for details.

**Benchmarks.** We focus on episodic, single, continual, and lifelong tasks, which are widely concerned in Test-Time Adaptation research. In episodic task, a single batch of test samples is used to optimize the model, and then the updated model makes predictions for current batch. After

Table 2: Experimental results (top-1 classification accuracy (%)) on the lifelong TTA task.

| Methods | Round R-1 | R-2 | R-3 | R-4 | R-5 | R-6 | R-7 | R-8 | R-9 | R-10 | Average |
|---|---|---|---|---|---|---|---|---|---|---|---|
| NoAdapt | $31.6_{\pm0.00}$ | $31.6_{\pm0.00}$ | $31.6_{\pm0.00}$ | $31.6_{\pm0.00}$ | $31.6_{\pm0.00}$ | $31.6_{\pm0.00}$ | $31.6_{\pm0.00}$ | $31.6_{\pm0.00}$ | $31.6_{\pm0.00}$ | $31.6_{\pm0.00}$ | $31.6_{\pm0.00}$ |
| Tent | $8.1_{\pm0.06}$ | $0.1_{\pm0.00}$ | $0.1_{\pm0.00}$ | $0.1_{\pm0.00}$ | $0.1_{\pm0.00}$ | $0.1_{\pm0.00}$ | $0.1_{\pm0.00}$ | $0.1_{\pm0.00}$ | $0.1_{\pm0.00}$ | $0.1_{\pm0.00}$ | $0.9_{\pm0.01}$ |
| + PTTA | $60.0_{\pm0.05}$ | $31.8_{\pm0.11}$ | $0.1_{\pm0.00}$ | $0.1_{\pm0.00}$ | $0.1_{\pm0.00}$ | $0.1_{\pm0.00}$ | $0.1_{\pm0.00}$ | $0.1_{\pm0.00}$ | $0.1_{\pm0.00}$ | $0.1_{\pm0.00}$ | $9.3_{\pm0.02}$ |
| CoTTA | $42.9_{\pm0.25}$ | $40.6_{\pm0.57}$ | $37.0_{\pm1.33}$ | $34.8_{\pm0.82}$ | $33.5_{\pm1.02}$ | $32.0_{\pm0.67}$ | $31.0_{\pm0.50}$ | $30.5_{\pm0.61}$ | $30.7_{\pm0.89}$ | $30.6_{\pm0.90}$ | $34.4_{\pm0.53}$ |
| + PTTA | $52.1_{\pm0.28}$ | $46.3_{\pm0.25}$ | $42.8_{\pm0.04}$ | $40.3_{\pm0.06}$ | $39.2_{\pm0.09}$ | $38.8_{\pm0.01}$ | $38.4_{\pm0.14}$ | $38.2_{\pm0.03}$ | $38.0_{\pm0.05}$ | $37.6_{\pm0.08}$ | $41.2_{\pm0.02}$ |
| SoTTA | $59.5_{\pm0.22}$ | $60.7_{\pm0.15}$ | $61.0_{\pm0.15}$ | $61.3_{\pm0.16}$ | $61.5_{\pm0.13}$ | $61.5_{\pm0.19}$ | $61.6_{\pm0.15}$ | $61.7_{\pm0.15}$ | $61.8_{\pm0.22}$ | $61.9_{\pm0.20}$ | $61.3_{\pm0.24}$ |
| + PTTA | $61.3_{\pm0.39}$ | $62.6_{\pm0.42}$ | $63.1_{\pm0.34}$ | $63.4_{\pm0.31}$ | $63.5_{\pm0.27}$ | $63.7_{\pm0.26}$ | $63.8_{\pm0.27}$ | $63.9_{\pm0.25}$ | $63.9_{\pm0.25}$ | $64.0_{\pm0.24}$ | $63.3_{\pm0.43}$ |
| SAR | $60.0_{\pm0.02}$ | $61.1_{\pm0.02}$ | $61.4_{\pm0.02}$ | $61.6_{\pm0.02}$ | $61.7_{\pm0.02}$ | $61.8_{\pm0.03}$ | $61.7_{\pm0.02}$ | $61.6_{\pm0.08}$ | $59.3_{\pm0.26}$ | $60.4_{\pm0.06}$ | $61.1_{\pm0.04}$ |
| + PTTA | $61.6_{\pm0.01}$ | $63.0_{\pm0.00}$ | $63.4_{\pm0.00}$ | $63.6_{\pm0.03}$ | $63.8_{\pm0.02}$ | $63.9_{\pm0.01}$ | $64.0_{\pm0.02}$ | $64.0_{\pm0.02}$ | $64.1_{\pm0.00}$ | $64.1_{\pm0.02}$ | $63.5_{\pm0.00}$ |
| ETA | $62.2_{\pm0.07}$ | $59.5_{\pm0.13}$ | $55.4_{\pm0.39}$ | $46.6_{\pm7.84}$ | $31.5_{\pm27.2}$ | $29.5_{\pm25.4}$ | $28.2_{\pm24.3}$ | $26.2_{\pm22.6}$ | $26.1_{\pm22.6}$ | $25.1_{\pm21.6}$ | $39.0_{\pm15.2}$ |
| + PTTA | $65.3_{\pm0.06}$ | $65.4_{\pm0.03}$ | $65.3_{\pm0.06}$ | $65.1_{\pm0.01}$ | $64.9_{\pm0.03}$ | $64.7_{\pm0.05}$ | $64.6_{\pm0.03}$ | $64.4_{\pm0.06}$ | $64.3_{\pm0.04}$ | $64.1_{\pm0.04}$ | $64.8_{\pm0.00}$ |
| EATA | $62.5_{\pm0.40}$ | $62.2_{\pm0.39}$ | $61.9_{\pm0.37}$ | $61.7_{\pm0.44}$ | $61.6_{\pm0.48}$ | $61.4_{\pm0.44}$ | $61.3_{\pm0.53}$ | $61.2_{\pm0.38}$ | $61.0_{\pm0.42}$ | $61.0_{\pm0.35}$ | $61.6_{\pm0.41}$ |
| + PTTA | $64.7_{\pm0.31}$ | $65.0_{\pm0.43}$ | $65.0_{\pm0.39}$ | $65.1_{\pm0.43}$ | $65.1_{\pm0.40}$ | $65.0_{\pm0.38}$ | $65.0_{\pm0.45}$ | $65.0_{\pm0.43}$ | $65.0_{\pm0.39}$ | $65.0_{\pm0.41}$ | $65.0_{\pm0.40}$ |
| DeYO | $62.0_{\pm0.50}$ | $43.1_{\pm18.9}$ | $32.7_{\pm29.0}$ | $25.0_{\pm27.8}$ | $17.9_{\pm30.7}$ | $16.8_{\pm28.8}$ | $15.9_{\pm27.4}$ | $3.5_{\pm5.94}$ | $0.1_{\pm0.01}$ | $0.1_{\pm0.00}$ | $21.7_{\pm15.9}$ |
| + PTTA | $65.8_{\pm0.01}$ | $66.0_{\pm0.05}$ | $66.0_{\pm0.04}$ | $65.9_{\pm0.05}$ | $65.9_{\pm0.05}$ | $65.8_{\pm0.07}$ | $65.7_{\pm0.05}$ | $65.5_{\pm0.07}$ | $64.9_{\pm0.91}$ | $61.6_{\pm4.30}$ | $65.3_{\pm0.53}$ |
| CPL | $55.5_{\pm0.13}$ | $57.3_{\pm0.11}$ | $57.8_{\pm0.10}$ | $58.1_{\pm0.15}$ | $58.4_{\pm0.27}$ | $58.5_{\pm0.21}$ | $58.5_{\pm0.20}$ | $58.7_{\pm0.11}$ | $58.8_{\pm0.10}$ | $58.8_{\pm0.21}$ | $58.0_{\pm0.13}$ |
| + PTTA | $59.7_{\pm0.02}$ | $61.5_{\pm0.02}$ | $62.0_{\pm0.03}$ | $62.3_{\pm0.02}$ | $62.6_{\pm0.02}$ | $62.7_{\pm0.00}$ | $62.9_{\pm0.03}$ | $63.0_{\pm0.00}$ | $63.1_{\pm0.01}$ | $63.2_{\pm0.03}$ | $62.3_{\pm0.01}$ |

Table 3: Experimental results (top-1 classification accuracy (%)) on the single and continual TTA tasks for ImageNet-C (left) and other ImageNet variants (right). We highlight the highest accuracy in **bold** and the second best as underline.

| Methods | ResNet50 Single | Contl. | ViT-B/16 Single | Contl. |
|---|---|---|---|---|
| NoAdapt | $18.2_{\pm0.00}$ | $18.2_{\pm0.00}$ | $31.6_{\pm0.00}$ | $31.6_{\pm0.00}$ |
| Tent | $42.8_{\pm0.04}$ | $9.9_{\pm0.22}$ | $51.3_{\pm0.31}$ | $14.2_{\pm10.5}$ |
| + PTTA | $44.9_{\pm0.05}$ | $36.9_{\pm1.82}$ | $52.1_{\pm0.26}$ | $60.1_{\pm0.08}$ |
| CoTTA | $32.9_{\pm0.01}$ | $33.3_{\pm0.04}$ | $38.5_{\pm0.69}$ | $42.9_{\pm0.25}$ |
| + PTTA | $34.0_{\pm0.53}$ | $37.6_{\pm0.02}$ | $48.4_{\pm4.65}$ | $51.3_{\pm0.73}$ |
| SoTTA | $41.5_{\pm0.01}$ | $30.6_{\pm0.26}$ | $57.1_{\pm0.31}$ | $59.5_{\pm0.31}$ |
| + PTTA | $42.9_{\pm0.36}$ | $37.0_{\pm0.69}$ | $58.1_{\pm0.49}$ | $61.3_{\pm0.55}$ |
| SAR | $44.0_{\pm0.15}$ | $40.0_{\pm0.61}$ | $56.7_{\pm0.19}$ | $60.0_{\pm0.04}$ |
| + PTTA | $45.5_{\pm0.93}$ | $46.2_{\pm0.02}$ | $58.0_{\pm0.04}$ | $61.6_{\pm0.03}$ |
| ETA | $48.2_{\pm0.04}$ | $43.7_{\pm0.06}$ | $63.7_{\pm0.03}$ | $62.2_{\pm0.07}$ |
| + PTTA | $49.5_{\pm0.00}$ | $48.5_{\pm0.05}$ | $64.4_{\pm0.04}$ | $65.3_{\pm0.06}$ |
| EATA | $48.0_{\pm0.06}$ | $48.4_{\pm0.03}$ | $61.7_{\pm0.51}$ | $62.2_{\pm0.63}$ |
| + PTTA | $49.1_{\pm0.08}$ | $\mathbf{50.3_{\pm0.04}}$ | $62.7_{\pm0.35}$ | $64.7_{\pm0.31}$ |
| DeYO | $48.6_{\pm0.11}$ | $9.2_{\pm0.25}$ | $64.2_{\pm0.46}$ | $62.2_{\pm0.35}$ |
| + PTTA | $\mathbf{49.9_{\pm0.10}}$ | $25.7_{\pm4.98}$ | $\mathbf{65.3_{\pm0.01}}$ | $\mathbf{65.8_{\pm0.01}}$ |
| CPL | $42.8_{\pm0.04}$ | $42.3_{\pm0.15}$ | $52.4_{\pm0.08}$ | $55.5_{\pm0.13}$ |
| + PTTA | $44.6_{\pm0.06}$ | $45.3_{\pm0.02}$ | $54.6_{\pm0.08}$ | $59.7_{\pm0.02}$ |

| Methods | ImageNet -1K | -A | -V2. | -R. | -S. | Avg. |
|---|---|---|---|---|---|---|
| NoAdapt | $78.0_{\pm0.00}$ | $23.9_{\pm0.00}$ | $66.3_{\pm0.00}$ | $43.2_{\pm0.00}$ | $18.2_{\pm0.00}$ | $45.9_{\pm0.00}$ |
| Tent | $82.0_{\pm0.03}$ | $29.0_{\pm0.57}$ | $70.3_{\pm0.20}$ | $17.6_{\pm0.02}$ | $2.0_{\pm0.24}$ | $40.2_{\pm0.10}$ |
| + PTTA | $82.0_{\pm0.07}$ | $28.7_{\pm0.75}$ | $70.4_{\pm0.29}$ | $19.1_{\pm0.62}$ | $2.5_{\pm0.25}$ | $40.5_{\pm0.83}$ |
| CoTTA | $79.6_{\pm0.18}$ | $26.8_{\pm0.64}$ | $68.3_{\pm0.16}$ | $53.9_{\pm0.98}$ | $13.8_{\pm9.40}$ | $48.5_{\pm1.71}$ |
| + PTTA | $81.0_{\pm0.00}$ | $26.9_{\pm0.66}$ | $70.1_{\pm0.12}$ | $51.1_{\pm0.27}$ | $22.3_{\pm0.45}$ | $50.3_{\pm0.25}$ |
| SoTTA | $82.2_{\pm0.09}$ | $29.6_{\pm0.50}$ | $70.4_{\pm0.04}$ | $57.4_{\pm0.12}$ | $40.8_{\pm0.30}$ | $56.1_{\pm0.11}$ |
| + PTTA | $82.0_{\pm0.04}$ | $30.1_{\pm0.12}$ | $70.5_{\pm0.06}$ | $57.6_{\pm0.25}$ | $40.9_{\pm0.16}$ | $56.2_{\pm0.11}$ |
| SAR | $81.0_{\pm0.05}$ | $30.0_{\pm0.74}$ | $70.3_{\pm0.11}$ | $43.3_{\pm0.18}$ | $23.6_{\pm1.35}$ | $49.6_{\pm0.36}$ |
| + PTTA | $82.1_{\pm0.03}$ | $30.5_{\pm0.23}$ | $70.5_{\pm0.17}$ | $43.2_{\pm0.02}$ | $24.3_{\pm0.55}$ | $50.1_{\pm0.19}$ |
| ETA | $82.3_{\pm0.05}$ | $29.8_{\pm1.19}$ | $71.3_{\pm0.22}$ | $56.5_{\pm0.90}$ | $\underline{45.9_{\pm0.14}}$ | $57.2_{\pm0.18}$ |
| + PTTA | $\mathbf{82.7_{\pm0.02}}$ | $32.7_{\pm2.47}$ | $71.7_{\pm0.06}$ | $\mathbf{61.4_{\pm0.04}}$ | $\mathbf{47.4_{\pm0.09}}$ | $\underline{59.2_{\pm0.53}}$ |
| EATA | $81.7_{\pm0.07}$ | $31.4_{\pm0.79}$ | $70.8_{\pm0.06}$ | $57.7_{\pm0.71}$ | $44.6_{\pm0.58}$ | $57.2_{\pm0.15}$ |
| + PTTA | $81.9_{\pm0.07}$ | $30.8_{\pm0.82}$ | $71.1_{\pm0.11}$ | $59.9_{\pm0.33}$ | $45.6_{\pm0.56}$ | $57.9_{\pm0.08}$ |
| DeYO | $82.2_{\pm0.11}$ | $\underline{39.1_{\pm0.34}}$ | $\underline{71.8_{\pm0.25}}$ | $59.5_{\pm0.28}$ | $40.8_{\pm1.30}$ | $58.7_{\pm0.13}$ |
| + PTTA | $\underline{82.7_{\pm0.10}}$ | $\mathbf{40.7_{\pm0.85}}$ | $\mathbf{71.9_{\pm0.12}}$ | $\underline{61.0_{\pm0.41}}$ | $42.4_{\pm3.18}$ | $\mathbf{59.7_{\pm0.57}}$ |
| CPL | $80.7_{\pm0.17}$ | $27.7_{\pm0.07}$ | $69.3_{\pm0.27}$ | $55.1_{\pm0.11}$ | $39.5_{\pm0.17}$ | $54.5_{\pm0.07}$ |
| + PTTA | $81.2_{\pm0.12}$ | $27.4_{\pm0.32}$ | $69.5_{\pm0.45}$ | $56.0_{\pm0.21}$ | $39.3_{\pm0.27}$ | $54.7_{\pm0.08}$ |

that, model's parameters are reset to the source. For single and continual tasks, the model is iteratively updated in static and dynamically changing environments respectively. Furthermore, lifelong task extends the dynamic environment indefinitely, set as 10 rounds and a total of 150 corruptions. Following previous TTA methods, we employ ImageNet-C, CIFAR100-C (Hendrycks & Dietterich, 2018), ImageNet (Deng et al., 2009) and its variants: -A (Hendrycks et al., 2021b), -V2. (Recht et al., 2019), -R. (Hendrycks et al., 2021a), -S. (Wang et al., 2019) to construct these tasks. ImageNet-C contains 15 types of corruptions applied to the original ImageNet validation images, each having 5 severity levels. We exploit the most severe level (5-th level) for experiments. The same applied to CIFAR100-C. ImageNet variants represent the natural distribution shifts caused by natural adversarial samples, image sources, artistic rendi-

tions, and black-and-white sketches. Beyond image classification, we also consider the semantic segmentation task and employ CarlaTTA dataset (Marsden et al., 2024a) for experiments. CarlaTTA includes 5 gradually changing scenarios: day2night, clear2fog, clear2rain, dynamic, and highway.

**Baselines.** We employ Tent (Wang et al., 2021) and TPT (Shu et al., 2022), two simple and general TTA methods, as baselines. Meanwhile, we compare with state-of-the-art (SOTA) sample-selection-based TTA algorithms, including ETA (Niu et al., 2022a), DeYO (Lee et al., 2024), and CPL (Zhang et al., 2024). Since our proposed PTTA is a plug-and-play solution, we apply PTTA to existing TTA methods and achieve superior performance improvements. Additionally, we compare with previous SOTA TTA approaches such as SAR (Niu et al., 2023), MEMO (Zhang et al., 2022), SoTTA

Table 4: Experimental results (top-1 classification accuracy (%)) for test time prompt tuning on the episodic TTA task.

| METHODS | IMAGENET | | | | | |
|---|---|---|---|---|---|---|
| | -1K | -A | -V2. | -R. | -S. | AVG. |
| *CLIP-RN50* | | | | | | |
| ZERO-SHOT | 58.2±0.00 | 21.8±0.00 | 51.4±0.00 | 56.2±0.00 | 33.4±0.00 | 44.2±0.00 |
| TPT | 58.7±0.05 | 21.9±0.36 | 52.1±0.29 | 57.0±0.14 | 33.3±0.05 | 44.6±0.16 |
| + PTTA (ID) | 59.1±0.01 | 22.6±0.16 | 52.4±0.19 | **58.0**±0.10 | 33.5±0.06 | 45.1±0.05 |
| + PTTA (OOD) | 59.1±0.07 | 22.5±0.44 | 52.3±0.07 | 57.9±0.12 | 33.4±0.04 | 45.0±0.14 |
| TPT (COOP) | **62.8**±0.05 | 22.6±0.24 | 55.1±0.16 | 56.0±0.02 | 34.2±0.07 | 46.1±0.06 |
| + PTTA (ID) | 62.6±0.06 | **23.9**±0.12 | **55.4**±0.04 | 57.5±0.02 | **34.5**±0.06 | **46.8**±0.02 |
| + PTTA (OOD) | 62.7±0.07 | 23.5±0.91 | 55.4±0.42 | 57.1±1.01 | 34.3±0.09 | 46.6±0.46 |
| *CLIP-ViT-B/16* | | | | | | |
| ZERO-SHOT | 66.7±0.00 | 47.9±0.00 | 60.9±0.00 | 74.0±0.00 | 46.1±0.00 | 59.1±0.00 |
| TPT | 66.2±0.05 | 48.1±0.52 | 59.9±0.11 | 74.3±0.12 | 45.8±0.11 | 58.9±0.10 |
| + PTTA (ID) | 66.1±0.09 | 48.7±0.23 | 60.0±0.08 | 75.1±0.08 | 45.7±0.12 | 59.1±0.07 |
| + PTTA (OOD) | 66.6±0.08 | 49.1±0.03 | 60.0±0.08 | 75.3±0.05 | 45.9±0.10 | 59.4±0.05 |
| TPT (COOP) | **71.3**±0.04 | 49.6±0.27 | 63.9±0.12 | 75.0±0.11 | 47.5±0.07 | 61.5±0.08 |
| + PTTA (ID) | 71.1±0.02 | 49.7±0.04 | **64.1**±0.08 | 73.5±0.08 | 47.2±0.04 | 61.1±0.08 |
| + PTTA (OOD) | 71.2±0.08 | 50.0±0.30 | 63.9±0.19 | **75.6**±0.05 | 47.6±0.07 | 61.6±0.05 |

Table 5: Experimental results for semantic segmentation on episodic and continual TTA tasks. We report the mIoU.

| METHODS | DAY2NIGHT | CLEAR2FOG | CLEAR2RAIN | DYNAMIC | HIGHWAY |
|---|---|---|---|---|---|
| NOADAPT | 58.4±0.00 | 52.8±0.00 | 71.8±0.00 | 46.6±0.00 | 28.7±0.00 |
| COTTA | 61.3±0.07 | 56.7±0.07 | 70.8±0.11 | 46.4±0.08 | 33.6±0.03 |
| MEMO-epi. | 61.1±0.01 | 55.1±0.03 | 71.7±0.01 | 50.3±0.01 | 35.3±0.12 |
| + PTTA | 61.4±0.01 | 55.3±0.02 | 72.1±0.01 | 50.9±0.01 | 35.3±0.11 |
| TENT-contl. | 61.5±0.01 | 56.0±0.02 | 70.9±0.02 | 50.4±0.02 | 32.0±0.01 |
| + PTTA | **62.3**±0.03 | **56.8**±0.22 | **72.5**±0.01 | **52.6**±0.03 | 33.7±0.08 |

Table 6: Error Rate (%) of adversarial attacked samples under the indiscriminate and instant attack scenario.

| METHODS | TENT | ETA | DEYO | CPL | AVG. ↓ |
|---|---|---|---|---|---|
| *CIFAR100-C* | | | | | |
| BN ADAPT | 53.3±0.16 | 52.4±0.24 | 54.2±0.09 | 54.9±0.02 | 53.7±1.06 |
| + PTTA | 52.0±0.18 | 51.5±0.23 | 52.3±0.24 | 54.6±0.12 | 52.6±1.40 |
| MEDBN | 46.6±0.34 | 45.9±0.14 | 48.7±0.01 | 47.2±0.03 | 47.1±1.17 |
| + PTTA | **45.7**±0.30 | **45.3**±0.02 | **46.6**±0.10 | **47.1**±0.21 | **46.2**±0.82 |
| *ImageNet-C* | | | | | |
| BN ADAPT | 69.4±0.11 | 63.2±0.02 | 62.6±0.03 | 66.1±0.07 | 65.3±3.11 |
| + PTTA | **65.4**±0.79 | **62.1**±0.03 | **62.1**±0.57 | **64.2**±0.01 | **63.5**±1.63 |

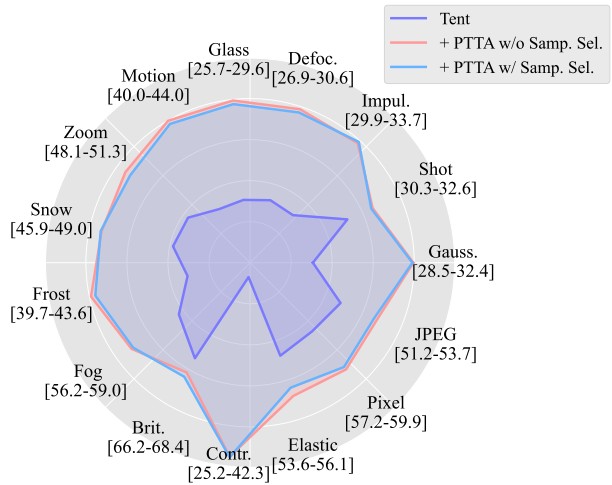

Figure 5: Comparison of selecting benign samples and using all samples as candidates for purifying malicious samples.

(Gong et al., 2023), CoTTA (Wang et al., 2022), and EATA (Niu et al., 2022a). Since CPL is designed for unsupervised domain adaptation, we modify it for TTA tasks.

**Implementations.** Considering that the pipeline of PTTA encompasses sample selection and malicious sample purification. We mainly focus on the latter purification strategy. Therefore, we employ existing sample selection criteria, including those proposed in ETA, DeYO, and CPL, refer to Sec. 2. Meanwhile, we also demonstrate that even without using sample selection, the saliency indicator can accurately be used for retrieval, refer to Sec. 4.3. We default to using the logit-saliency indicator due to its better performance and lower computational overhead. We build a memory bank for OOD retrieval, setting it as a first-in-first-out queue and limiting its maximum length to $1,000$. Although In-Batch retrieval does not require a memory bank, OOD retrieval is used by default as it is more suitable for diverse TTA tasks and existing TTA methods. Uniformly, we set $K = 1$ for Mixup and conduct an ablation study of $K$ in Sec. 4.4. We set $\alpha = 3.0$ for sample-selection-based TTA methods and $\alpha = 1.0$ for selection-free TTA methods. We employ various DNNs for experiments, including ResNet (He et al.,

2016), ViT (Dosovitskiy et al., 2021), CLIP Models (Radford et al., 2021), and DeepLab-V2 (Chen et al., 2017). We run experiments on 3 random seeds and report the average accuracy and the standard deviation.

### 4.2. Main Results

We compare the performance of Tent, CoTTA, SoTTA, SAR, ETA, EATA, DeYO, and CPL on the lifelong TTA task. Additionally, we incorporate our proposed PTTA into these methods, as shown in Table 2. We find that PTTA can alleviate the catastrophic forgetting of sample-selection-based TTA algorithms in lifelong tasks. For example, the accuracy drop from the 1-st to 10-th round is improved from $-37.1\%$ to $-1.2\%$ for ETA and ETA+PTTA. Although using the prior information of in-distribution data (similar to EATA) can also mitigate forgetting. The application of PTTA further enhances the utilization ratio of test data, improves the calibration of DNNs, and reduces over-confident predictions. Therefore, EATA+PTTA maintains the model performance without degradation, *i.e.*, a $0.3\%$ accuracy increase from the 1-st to 10-th round. The same conclusion is verified again in CPL+PTTA with another $3.5\%$ accuracy increase.

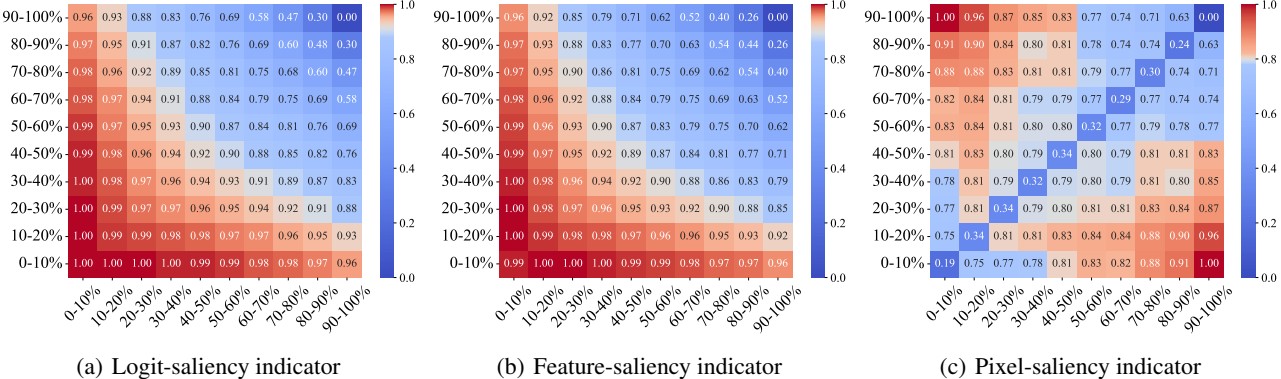

Figure 6: The Saliency Distance (normalized to the range of $0 \sim 1$) among test samples sorted in the ascending order of prediction entropy and split by percentages. A good indicator satisfies $\mathcal{D}_{sa}(x^+, x^+) > \mathcal{D}_{sa}(x^+, x^-) > \mathcal{D}_{sa}(x^-, x^-)$.

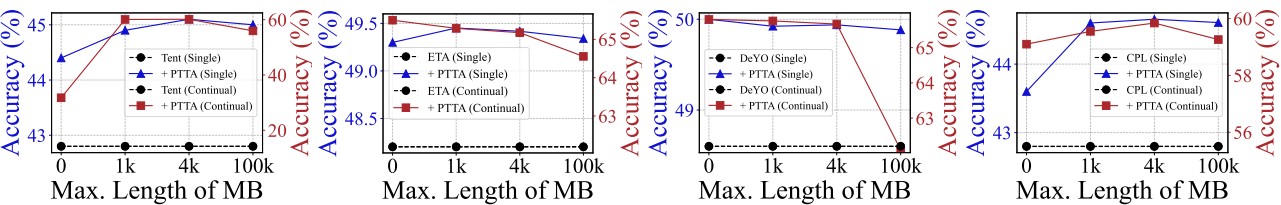

Figure 7: Ablation study of different maximum lengths of memory bank (MB).

We conduct experiments on single and continual TTA tasks using ResNet and ViT to verify the generality of PTTA across different DNN architectures in Table 3. The single task emphasizes optimizing the model within a static target distribution with limited samples. Therefore, PTTA utilizes malicious samples that should otherwise be filtered out, improving the model's optimization efficiency and avoiding sub-optimal state. Applying PTTA leads to an average accuracy increase of $1.5\%$ for ResNet50 and $1.2\%$ for ViT-B/16. Meanwhile, PTTA addresses the performance degradation of ResNet50 in dynamically changing environments, caused by the weak robustness of ResNet50 to distribution shifts, and achieves improvements of $37.0\%$ and $26.5\%$ for Tent and DeYO respectively. Experiments on CIFAR100-C also validate the above conclusions, referred to the Table 15.

We also conduct single and episodic TTA experiments on datasets representing natural distribution shifts, namely ImageNet and its variants, as shown in Table 3 and 4. PTTA achieves an additional accuracy improvement of up to $2.0\%$ over existing TTA methods. Meanwhile, on average, PTTA increases the zero-shot accuracy of CLIP models by $1.8\%$ for CLIP-RN50 and $1.4\%$ for CLIP-ViT/B-16. Beyond image classification, we also validate PTTA on semantic segmentation that simulates practical autonomous driving requirements. The experimental results, as shown in Table 5, indicate that PTTA achieves an mIoU improvement of up to $1.8\%$ in scenarios of day-to-night and clear-to-rainy/foggy weather changes, as well as urban-to-highway transitions. The efficiency of PTTA is analyzed in Appendix C.4.

### 4.3. Extended Analysis

**Effect of Saliency Indicator.** We compare the logit-, feature-, and pixel-saliency indicators mentioned in Sec. 3.2 by visualizing the saliency distance $\mathcal{D}_{sa}$ among test samples, as shown in Fig. 6. We hope that all test samples can be matched with benign samples for Mixup. Therefore, a good saliency indicator should satisfy $\mathcal{D}_{sa}(x^+, x^+) > \mathcal{D}_{sa}(x^+, x^-) > \mathcal{D}_{sa}(x^-, x^-)$, where $x^+$ and $x^-$ represent benign and malicious samples respectively. A typical example is shown in Fig. 6 (a). In conclusion, the logit-saliency indicator can better encode benign and malicious samples than the other two, incurs lower computational overhead, and is more suitable for purifying malicious samples. Additionally, the results of CoTTA+PTTA in Tables 2 and 3 show that the teacher-student consistency loss function can also be effectively applied to the saliency indicator, demonstrating the flexibility in choosing objective functions.

**Necessity of Benign Sample Retrieval.** Considering the usage of Mixup technique in PTTA, we compare with the vanilla Mixup method (Zhang et al., 2018), which stores historical samples in a first-in-first-out memory queue (maximum length of $1,000$) and randomly selects samples from it to mixup with mini-batch samples. Results in Table 18 show that vanilla Mixup provides significantly weaker performance improvements to base TTA algorithms than PTTA. This demonstrates the necessity of benign sample retrieval.

**Unnecessary of Sample Selection Criteria.** Sample selection is employed to screen benign samples for purifying

Table 7: Ablation study on retrieving top-$K$ benign samples for malicious sample purification.

| METHODS | SINGLE | | | |
| | $K = 1$ | $K = 2$ | $K = 3$ | AVG. |
| --- | --- | --- | --- | --- |
| TENT | $42.8_{\pm0.04}$ | $42.8_{\pm0.04}$ | $42.8_{\pm0.04}$ | $42.8_{\pm0.04}$ |
| + PTTA | $44.9_{\pm0.05}$ | $44.7_{\pm0.04}$ | $44.5_{\pm0.03}$ | $44.7_{\pm0.22}$ |
| ETA | $48.2_{\pm0.04}$ | $48.2_{\pm0.04}$ | $48.2_{\pm0.04}$ | $48.2_{\pm0.04}$ |
| + PTTA | $\underline{49.0}_{\pm0.05}$ | $\underline{49.1}_{\pm0.09}$ | $\underline{49.1}_{\pm0.06}$ | $\underline{49.1}_{\pm0.06}$ |
| DEYO | $48.6_{\pm0.11}$ | $48.6_{\pm0.11}$ | $48.6_{\pm0.11}$ | $48.6_{\pm0.11}$ |
| + PTTA | $\mathbf{49.3}_{\pm0.06}$ | $\mathbf{49.3}_{\pm0.16}$ | $\mathbf{49.4}_{\pm0.00}$ | $\mathbf{49.3}_{\pm0.03}$ |
| CPL | $42.8_{\pm0.04}$ | $42.8_{\pm0.04}$ | $42.8_{\pm0.04}$ | $42.8_{\pm0.04}$ |
| + PTTA | $44.6_{\pm0.06}$ | $44.4_{\pm0.05}$ | $44.1_{\pm0.08}$ | $44.4_{\pm0.25}$ |
| | CONTINUAL | | | |
| TENT | $14.2_{\pm10.5}$ | $14.2_{\pm10.5}$ | $14.2_{\pm10.5}$ | $14.2_{\pm10.5}$ |
| + PTTA | $60.1_{\pm0.08}$ | $59.7_{\pm0.01}$ | $59.4_{\pm0.05}$ | $59.7_{\pm0.32}$ |
| ETA | $62.2_{\pm0.07}$ | $62.2_{\pm0.07}$ | $62.2_{\pm0.07}$ | $62.2_{\pm0.07}$ |
| + PTTA | $\underline{64.7}_{\pm0.04}$ | $\underline{64.3}_{\pm0.07}$ | $\underline{63.9}_{\pm0.07}$ | $\underline{64.2}_{\pm0.43}$ |
| DEYO | $62.2_{\pm0.35}$ | $62.2_{\pm0.35}$ | $62.2_{\pm0.35}$ | $62.2_{\pm0.35}$ |
| + PTTA | $\mathbf{65.1}_{\pm0.03}$ | $\mathbf{64.9}_{\pm0.01}$ | $\mathbf{64.4}_{\pm0.32}$ | $\mathbf{64.9}_{\pm0.21}$ |
| CPL | $55.5_{\pm0.13}$ | $55.5_{\pm0.13}$ | $55.5_{\pm0.13}$ | $55.5_{\pm0.13}$ |
| + PTTA | $59.6_{\pm0.01}$ | $59.1_{\pm0.02}$ | $58.4_{\pm0.04}$ | $59.0_{\pm0.58}$ |

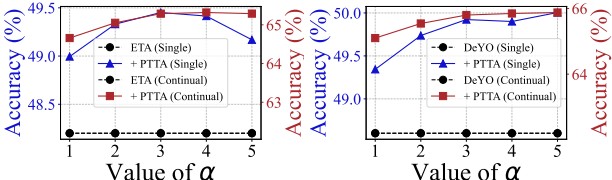

Figure 8: Ablation Study of $\alpha$ in Eq. 8.

malicious samples. However, we find that an excellent saliency indicator can retrieve correctly matched benign samples from candidates mixed with both benign and malicious samples. Thus, PTTA can be applied without the assistance of sample selection, and meanwhile, avoid the impact of selection thresholds. The experimental results in Fig. 5 indicate that Tent+PTTA exhibits comparable performance with or without sample selection. It further validates the effectiveness of the logit-saliency indicator.

**Adversarial Defense.** We also verify the performance of PTTA under the indiscriminate and instant attack scenarios, as shown in Table 6. The classification error rate of attacked samples is reduced by applying PTTA on BN Adapt (Schneider et al., 2020) and MedBN (Park et al., 2024). It indicates that PTTA can be effectively used for adversarial defense.

### 4.4. Ablation Studies

**Ablation Study of $\alpha$ in Eq. 8.** $\alpha$ is introduced to balance the original TTA objective function $\mathcal{L}_{\text{tta}}$ and the purification loss we proposed $\mathcal{L}_{\text{pur}}$. As shown in Fig. 8, we compare the impacts of $\alpha$ on applying PTTA to existing TTA methods. Since sample-selection-based TTA methods assign greater weights to lower-entropy samples, a larger $\alpha$ is appropriate.

**Different Size of Memory Bank.** The size of a Memory Bank (MB) determines the scope of benign sample retrieval and the required additional storage overhead. In Fig. 7, we compare the impacts of different MB sizes, where a maximum length of $0$ indicates that we use In-Batch retrieval. The results show that a small MB, such as $1,000$ by default, is appropriate, while an extremely large MB is not suitable for TTA tasks in dynamically changing environments.

**Comparison of Retrieving Top-$K$ Benign Samples.** The $K$ determines the number of benign samples used for Mixup and the linear interpolation coefficient $\lambda = 1/(K + 1)$. As depicted in Table 7, PTTA is insensitive to the value of $K$. However, as the value of $K$ approaches infinity, it will dilute the proportion of malicious data information in purified samples. To maximize the utilization of test data, we set $K = 1$ as the default.

## 5. Conclusion

In this paper, we show that malicious samples severely degrade the performance of TTA methods and highlight the inherent difficulty of perfectly detecting them from test data. Instead of selecting and filtering out malicious samples, we propose a plug-and-play solution, PTTA. It retrieves benign samples, exerting the most opposite influence on the objective function based on a saliency indicator, to purify malicious samples. PTTA improves the utilization rate of test data, delivers more stable and robust test-time predictions of DNNs, and reduces existing TTA algorithms' sensitivity to sample selection thresholds. We further discuss the necessity of benign sample retrieval and the unnecessary of sample selection criteria. PTTA achieves superior performance in four types of TTA tasks, as well as in classification, segmentation, and adversarial defense tasks.

## Acknowledgements

This work was supported by the 111 Project on Computational Intelligence and Intelligent Control under Grant B18024, the Foundation for Outstanding Research Groups of Hubei Province under Grant 2025AFA012, the project of Peng Cheng Lab under Grant PCL2023A08, and the Natural Science Fund of Hubei Province under Grant 2022CFB823.

## Impact Statement

This paper presents work whose goal is to advance the field of Machine Learning. There are many potential societal consequences of our work, none which we feel must be specifically highlighted here.

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

# APPENDIX

## Contents

# A. Related Work

## A.1. Test-Time Adaptation

TTA is a task of online adapting Deep Neural Networks (DNNs) pretrained on the source domain to arbitrary target data distributions while achieving performance improvement at test time. Unlike classical Unsupervised Domain Adaptation, TTA does not assume the stationarity of the target distribution. Instead, it conducts online learning in a dynamic environment that changes over time (Wang et al., 2022). The optimization objective of TTA is to maximize the adaptation of current batch in the data stream while preventing the catastrophic forgetting of DNNs. Recent research on TTA focuses on utilizing the self-supervised paradigm to update part of parameters in DNNs under the condition that source data and its statistics are unavailable (Lee et al., 2024; Wang et al., 2022; Niu et al., 2022a; Wang et al., 2021). Tent (Wang et al., 2021) connects the distribution shift and prediction error to the entropy. More severe shifts and more errors all-in-all lead to higher entropy. On this basis, entropy minimization is the sole loss function to update Batch Normalization layers in DNNs. Following Tent, EATA (Niu et al., 2022a) shows that high-entropy samples lead to noisy gradients that could disrupt the model. Therefore, EATA employs an active sample selection criterion to identify reliable and non-redundant samples for TTA. Additionally, DeYO (Lee et al., 2024) points out that entropy alone is an unreliable confidence metric, and proposes applying an object-destructive transformation to filter predictions that are more grounded in the object's shape. CPL (Zhang et al., 2024) is another approach developed for fine-tuning vision-language models. It leverages intra- and inter-instance label selection based on a confidence score matrix to obtain appropriate candidate pseudo-labels for unlabeled data.

Unlike leveraging sample selection criteria to filter malicious samples, some TTA methods aim to devise general optimization strategies that utilize all test samples which achieving significant performance improvements, such as FOA (Niu et al., 2024), ROID (Marsden et al., 2024b), MGTTA (Deng et al., 2025), and EATA-C (Tan et al., 2025). Some studies investigate enhancing TTA performance by improving the model adaptation process, such as improving entropy loss (Yuan et al., 2024; Wang et al., 2025), refining batch normalization (Su et al., 2024; Park et al., 2024), correcting the pseudo-labels (Ma, 2024), and optimizing the inference process (Niu et al., 2022b). We demonstrate that the purification strategy effectively reduces malicious content in all test samples. Meanwhile, we show that the PTTA serves as a versatile plug-and-play method compatible with existing TTA approaches.

Other than improving TTA methods, a series of studies focus on the security vulnerabilities of TTA algorithms (Wu et al., 2023; Cong et al., 2024). Distribution Invading Attack (Wu et al., 2023) is proposed based on the insight that the predictions of benign samples can be affected by malicious samples within the same batch. These malicious samples can easily be exploited by adversaries to create poisoned samples, which are injected into the test batch to make DNNs misclassify benign and unperturbed test data. Cong *et al.* (2024) show that as few as 10 poisoned samples can degrade the performance of TTA by 35%. Inspired by these findings, we propose an effective purification strategy for malicious samples to make full use of test data, avoid sample filtering, thus further improving the performance of existing TTA methods.

Furthermore, the TTA framework has been widely adopted across various Artificial Intelligence tasks, including image super-resolution (Deng et al., 2023), video classification (Lin et al., 2023; Zeng et al., 2023), visual question answering (Wen et al., 2023), *etc.*. Notably, the malicious sample hazard is inherently persistent across these tasks. Extending PTTA to these tasks in practical application scenarios holds potential for yielding additional benefits.

## A.2. Adversarial Attack and Defense

Adversarial Attack aims to generate adversarial samples that are indistinguishable from natural data, by applying small yet intentionally worst-case perturbations, and manipulate the input to fool DNNs into producing incorrect output. The best practice is to utilize the gradient direction of image pixels with respect to maximizing the objective function, which measures the contribution of each pixel to the success of the attack, to generate adversarial noise for one-step or multi-step attacks (Szegedy, 2013; Wu et al., 2023; Cong et al., 2024). Adversarial Attacks have inspired the research community to focus on how to train robust models so that there are no adversarial samples or at least an adversary cannot easily find them (Madry et al., 2018). These defense methods include adversarial training (Goodfellow et al., 2014), adversarial distillation (Goldblum et al., 2020), and adversarial purification (Shi et al., 2021; Tang & Zhang, 2024; Goodfellow et al., 2014). SOAP (Shi et al., 2021) exploits an auxiliary self-supervised task to perform gradient descent for image pixels to purify adversarial samples at test time. TPAP (Tang & Zhang, 2024) uses the Fast Gradient Sign Method (FGSM) (Goodfellow et al., 2014) as a way to purify unknown adversarial perturbations to obtain correctly classified purified samples. Yang *et al.* (2024) introduce the manifold hypothesis to adversarial defense and propose an adversarial purification method based on manifold learning and variational inference. In this paper, we focus on solving the systematic issues that affect the performance

bottleneck of TTA. We are motivated by the idea of image purification and propose to utilize benign samples to purify malicious samples. Instead of noise, we find that image Mixup (Zhang et al., 2018) can be more effectively and efficiently applied as a part of the purification strategy for test-time model adaptation.

## B. More Design Details of PTTA

### B.1. Proof of Logit-Saliency Indicator

Given a sample $x$, the predicted output of model $f_\theta$ for $x$ is defined as logits $\mathbf{z} = f_\theta(x) = [z_1, ..., z_C]$, where $C$ is the number of classes. The predicted probability $\mathbf{p} = \sigma(f_\theta(x)) = [p_1, ..., p_C]$ is obtained by applying the Softmax function $\sigma(\cdot)$, *i.e.*,

$$p_i = \frac{\exp(z_i)}{\sum_j^C \exp(z_j)}. \tag{9}$$

The partial derivative of $p_i$ with respect to $z_i$ and $z_j$ $(i \neq j)$ is

$$
\begin{aligned}
\frac{\partial p_i}{\partial z_i} &= \frac{\exp(z_i) \cdot \sum_j^C \exp(z_j) - \exp(z_i) \cdot \exp(z_i)}{(\sum_j^C \exp(z_j))^2} = p_i - p_i^2, \\
\frac{\partial p_i}{\partial z_j} &= \frac{-\exp(z_i) \cdot \exp(z_j)}{(\sum_j^C \exp(z_j))^2} = -p_i \cdot p_j.
\end{aligned}
\tag{10}
$$

According to the formula of conditional entropy, $\mathcal{L}_{\text{Ent}}(f_\theta(x)) = -\sum_i^C p_i \log p_i$, the partial derivative of $\mathcal{L}_{\text{Ent}}(f_\theta(x))$ with respect to $z_i$ is

$$
\begin{aligned}
\frac{\partial \mathcal{L}_{\text{Ent}}(f_\theta(x))}{\partial z_i} &= -\frac{\partial \sum_j^C p_j \log p_j}{\partial z_i} \\
&= -\left( \frac{\partial p_i \log p_i}{\partial z_i} + \sum_{j \neq i}^C \frac{\partial p_j \log p_j}{\partial z_i} \right) \\
&= -\left( \log p_i \frac{\partial p_i}{\partial z_i} + p_i \frac{\partial \log p_i}{\partial z_i} + \sum_{j \neq i}^C \log p_j \frac{\partial p_j}{\partial z_i} + \sum_{j \neq i}^C p_j \frac{\partial \log p_j}{\partial z_i} \right) \\
&= -\left( \log p_i \frac{\partial p_i}{\partial z_i} + \frac{\partial p_i}{\partial z_i} + \sum_{j \neq i}^C \log p_j \frac{\partial p_j}{\partial z_i} + \sum_{j \neq i}^C \frac{\partial p_j}{\partial z_i} \right) \\
&= -\left( (1 + \log p_i)(p_i - p_i^2) + \sum_{j \neq i}^C (1 + \log p_j)(-p_i p_j) \right) \\
&= -\left( (1 + \log p_i)p_i + \sum_j^C (1 + \log p_j)(-p_i p_j) \right) \\
&= -\left( p_i(1 + \log p_i) - p_i \sum_j^C p_j(1 + \log p_j) \right).
\end{aligned}
\tag{11}
$$

Since we have $1 + \log p_i = z_i + (1 - \log \sum_j^C \exp(z_j))$, where $Q = 1 - \log \sum_j^C \exp(z_j)$ is independent of index $i$. Then,

the Eq. 11 can be rewritten as

$$
\begin{aligned}
\frac{\partial \mathcal{L}_{\mathrm{Ent}}(f_\theta(x))}{\partial z_i} &= -\left( p_i z_i + p_i Q - p_i \sum_j^C p_j z_j - p_i \sum_j^C p_j Q \right) \\
&= -\left( p_i z_i - p_i \sum_j^C p_j z_j + p_i Q(1 - \sum_j^C p_j) \right) \\
&= -\left( p_i z_i - p_i \sum_j^C p_j z_j \right) \\
&= -p_i \left( z_i - \sum_j^C p_j z_j \right).
\end{aligned}
\tag{12}
$$

Next, we use vectors to express the gradient of Entropy Minimization as

$$
\nabla_z \mathcal{L}_{\mathrm{Ent}}(f_\theta(x)) = [\frac{\partial \mathcal{L}_{\mathrm{Ent}}(f_\theta(x))}{\partial z_i}, ..., \frac{\partial \mathcal{L}_{\mathrm{Ent}}(f_\theta(x))}{\partial z_C}] = -(\mathbf{z} - \mathbf{p} \cdot \mathbf{z}) \odot \mathbf{p}
\tag{13}
$$

## B.2. Pseudo Code

---

**Algorithm 1** Purification for Test-Time Adaptation (PTTA)

---

**Input:** Source model $f_\theta$ with parameters $\theta$. Test samples $\mathbf{x}_t = \{x_{ti}\}_{i=1}^{N_{bs}} \in \mathcal{D}^{\mathrm{test}}$ arrived at time step $t$. A memory bank $\mathcal{M}$. Benign samples $\{x_{ti}^+, y_{ti}^+\}$ selected by basic TTA methods. Hyperparameters $\alpha$. Learning rate $\eta$.
**Output:** Predictions $\hat{\mathbf{y}}_t = \{\hat{y}_{ti}\}_{i=1}^{N_{bs}}$ for test samples $\mathbf{x}_t$.
**for** $\mathbf{x}_t \in \mathcal{D}^{test}$ **do**

    Compute predictions $\hat{\mathbf{y}}_t = f_\theta(\mathbf{x}_t)$ and logit-saliency indicator $\nabla_z \mathcal{L}_{\mathrm{Ent}}(f_\theta(\mathbf{x}_t))$ ;    // (Eq. 4)
    Incorporate selected benign samples $\{x_{ti}^+, y_{ti}^+, \nabla_z \mathcal{L}_{\mathrm{Ent}}(f_\theta(x_{ti}^+))\}$ into $\mathcal{M}$;
    Retrieve $x_j^*$ from $\mathcal{M}$ using saliency distance $\mathcal{D}_{sa}(x_i, x_j)$ ;    // (Eq. 3 and 5)
    Generate purified image $x_{ij}^*$ and its pseudo-label $y_{ij}^*$ using Mixup ;    // (Eq. 6)
    Compute total loss $\mathcal{L}_{\mathrm{total}} = \mathcal{L}_{\mathrm{tta}} + \alpha \mathcal{L}_{\mathrm{pur}}(x_{ij}^*, y_{ij}^*)$ ;    // (Eq. 7)
    Update $\theta$ with $\theta \leftarrow \theta - \eta \nabla_\theta \mathcal{L}_{\mathrm{total}}$;
**end**

---

# C. More Implementation Details

## C.1. Benchmarks

**CIFAR-100-C & ImageNet-C** (Hendrycks & Dietterich, 2018) respectively comprise corrupted variants of the validation images from CIFAR-100 (Krizhevsky et al., 2009) and ImageNet (Deng et al., 2009). These datasets include 15 types of corruptions: Gaussian noise (Gauss.), Shot noise (Shot), Impulse noise (Impul.), Defocus blur (Defoc.), Frosted Glass blur (Glass), Motion blur (Motion), Zoom blur (Zoom), Snow (Snow), Frost (Frost), Fog (Fog), Brightness (Brit.), Contrast (Contr.), Elastic (Elastic), Pixelation (Pixel), and JPEG (JPEG). Each corruption consists of 5 levels of severity. Our experiments and evaluation concentrate on the most severe corruption level (5-th level).

**ImageNet-1K**(Deng et al., 2009) is a widely used and highly influential dataset for evaluating the performance of deep learning algorithms. It encompasses $1,281,167$ images distributed across $1,000$ different object categories. In Test-Time Adaptation (TTA), the val set is often used to evaluate the performance of TTA algorithms on *i.i.d.* data and to measure the importance of parameters (*e.g.*, EATA). We use the validation images to construct an in-distribution (ID) memory bank.

**ImageNet-V2** (Recht et al., 2019) is an independent test set that consists of natural images. These images are gathered from diverse sources. It encompasses $10,000$ images which are distributed across $1,000$ ImageNet categories.

**ImageNet-A** (Hendrycks et al., 2021b) serves as a challenging test set. It is composed of "natural adversarial examples" that are misclassified by a standard ResNet50. This set contains $7,500$ images belonging to 200 ImageNet categories.

**ImageNet-R** (Hendrycks et al., 2021a) is a dataset that collects images of ImageNet categories but in the form of artistic renditions. In total, it has $30,000$ images, and these images cover 200 ImageNet categories.

**ImageNet-Sketch** (Wang et al., 2019) is a dataset made up of black-and-white sketches. It is collected independently from the original ImageNet validation set. Altogether, it includes $50,000$ images that span $1,000$ ImageNet categories.

**CarlaTTA**(Marsden et al., 2024a) is a synthetic autonomous driving dataset for exploring progressive test time adaptation in urban scene segmentation. It is created based on CARLA (Dosovitskiy et al., 2017), an open source simulator for autonomous driving research. It contains 5 progressive test sequences evolved from the sunny-noon source domain, namely day2night, clear2fog, clear2rain, dynamic, and highway.

## C.2. Experimental Protocols

**Episodic Test-Time Adaptation.** In the episodic task, TTA algorithms are required to optimize the model using the test samples within a single mini-batch. The updated model is then utilized to predict the current mini-batch. Subsequently, the model's parameters are reset to the source in preparation for learning the next batch of samples. We mainly use CLIP models (Radford et al., 2021), including CLIP-RN50 and CLIP-ViT-B/16, for the episodic tasks. Instead of updating the parameters of CLIP models, we follow TPT (Shu et al., 2022) to update the learnable text prompt. We conduct episodic tasks constructed on ImageNet and its variants. We utilize two distinct methods for initializing the text prompts: 1) the default prompt, "a photo of a {CLASS}", and 2) the prompt optimized through CoOP (Zhou et al., 2022). We select top-$10\%$ low-entropy test samples for TTA. We employ the AdamW with a learning rate of $0.005$ and a batch size of $64$. We run experiments on 3 random seeds and report the mean and standard deviation of the accuracy.

**Single Test-Time Adaptation.** We use 15 types of corruption in ImageNet-C and CIFAR100-C, as well as ImageNet and its variants to construct single tasks. A corruption or a variant represents a static distribution shift relative to ImageNet-1K. The model parameters are continuously updated within a static distribution without being reset. On ImageNet-C, we use ResNet50 and ViT-B/16, which are sourced from the torchvision[2] library and the timm[3] library, respectively. For ResNet50, we consistently use the SGD with a learning rate of $0.00025$, a momentum of $0.9$, a batch size of $64$, and no weight decay. For ViT-B/16, we consistently use the SGD with a learning rate of $0.001$, a momentum of $0.9$, a batch size of $64$, and no weight decay. In experiments with a batch size of 1, the learning rates for ResNet50 and ViT-B/16 are adjusted to $0.00025/64$ and $0.001/64$, except for DeYO and SAR, the learning rates are set to $0.00025/32$ for ResNet50 and $0.001/32$ for ViT-B/16. On ImageNet-R, ImageNet-V2, ImageNet-A, and ImageNet-Sketch, we conduct experiments using ViT-B/16. We employ the SGD with a learning rate of $0.005$, a momentum of $0.9$, a batch size of $64$, and no weight decay. On CIFAR100-C, we employ Wide ResNet-40 (WRN-40) with pre-trained weights from the RobustBench (Croce et al., 2021). All methods consistently use the SGD with a learning rate of $0.0005$, a momentum of $0.9$, a batch size of $64$, and no weight decay. We run experiments on 3 random seeds and report the mean and standard deviation of the accuracy.

**Continual Test-Time Adaptation.** The continual task considers a dynamically changing environment, which is constructed with 15 types of corruptions concatenated in the order of Gaussian noise $\rightarrow$ Shot noise $\rightarrow$ Impulse noise $\rightarrow$ Defocus blur $\rightarrow$ Frosted Glass blur $\rightarrow$ Motion blur $\rightarrow$ Zoom blur $\rightarrow$ Snow $\rightarrow$ Frost $\rightarrow$ Fog $\rightarrow$ Brightness $\rightarrow$ Contrast $\rightarrow$ Elastic $\rightarrow$ Pixelation $\rightarrow$ JPEG. The model is continuously optimized in such environment, and its parameters are never reset. We construct the continual tasks on ImageNet-C and CIFAR100-C, and we report the Top-1 classification accuracy on 15 consecutive corruptions. On ImageNet-C, we use ResNet50 and ViT-B/16. For ResNet50, we consistently use the SGD with a learning rate of $0.00025$, a momentum of $0.9$, a batch size of $64$, and no weight decay. For ViT-B/16, we consistently use the SGD with a learning rate of $0.001$, a momentum of $0.9$, a batch size of $64$, and no weight decay. In experiments with a batch size of 1, the learning rates for ResNet50 and ViT-B/16 are adjusted to $0.00025/64$ and $0.001/64$ except for DeYO and SAR, the learning rates are set to $0.00025/32$ for ResNet50 and $0.001/32$ for ViT-B/16. On CIFAR100-C, we use Wide ResNet-40 (WRN-40) with pre-trained weights from the RobustBench (Croce et al., 2021). All methods consistently use the SGD with a learning rate of $0.0005$, a momentum of $0.9$, a batch size of $64$, and no weight decay. We run experiments on 3 random seeds and report the mean and variance of the accuracy.

**Lifelong Test-Time Adaptation.** The lifelong task focuses on running TTA algorithms indefinitely to optimize the model in

---

[2]https://github.com/pytorch/vision
[3]https://github.com/pprp/timm

a dynamically changing environment. We define the 15 consecutive corruptions in the continual task as 1 round. Meanwhile, we use 10 rounds to compose the lifelong task, which includes 150 consecutive corruptions. We construct the lifelong tasks on ImageNet-C and report the Top-1 classification accuracy. We use ViT-B/16 with the SGD with a learning rate of $0.001$, a momentum of $0.9$, a batch size of $64$, and no weight decay. We run experiments on 3 random seeds and report the mean and variance of the accuracy.

**Semantic Segmentation.** Beyond image classification, we demonstrate the effectiveness of PTTA in the semantic segmentation task. We employ CarlaTTA as the benchmark and use the DeepLab-V2 (Chen et al., 2017) with ResNet-101 as the backbone. We use the SGD with a learning rate of $2.5 \times 10^{-4}$, a momentum of $0.9$, and a weight decay of $5 \times 10^{-4}$. We use the model's one-hot output as the pseudo-labels to calculate the KL-divergence loss. We set $N_{mb}$ to 10 and $\alpha$ to $1.0$. We adopt the standard mean-Intersection-over-Union (mIoU) as the evaluation metric, and report the average results and standard deviation over 3 random seeds.

**Adversarial Defense.** To more closely simulate the performance of different methods under real-world attack scenarios, we focus on the indiscriminate and instant attack scenario, where the attacker injects a set of attacked samples into the $t$-th mini-batch after adapting to the previous $t - 1$ unattacked batches. Based on the experimental setup of DIA (Wu et al., 2023), we set the batch size to 100, the number of attack steps to 50, the attack optimization rate to $1/255$, the initial perturbation to $0.5$, and the perturbation constraint to $1.0$. For CIFAR100-C, we use a ResNet-26 with EMA, and the malicious sample ratio is set to $20\%$. For ImageNet-C, we use a ResNet50 with a malicious sample ratio of $10\%$. In all experiments, we employ the Adam with a learning rate of $0.0005$ and no weight decay. We employ MedBN (Park et al., 2024) and BN Adapt (Schneider et al., 2020) as the baselines. We replace the original BN layers with MedBN or BN Adapt, and combine these two approaches with various TTA methods.

### C.3. Method Implementations

**PTTA** (Ours). The PTTA can be applied to other TTA methods. The sample selection criteria of existing methods are used to distinguish benign and malicious samples, and the benign ones are added to the memory bank. For sample-selection-based Tent variant, we employ the criterion of ETA. In general, the maximum length of the memory bank is $N_{mb} = 1,000$ for ID and OOD retrieval, and the values of $\alpha$ are set to 1/3/3/3/1 for Tent/ETA/EATA/DeYO/CPL, respectively.

**Tent**[4] (Wang et al., 2021). Tent minimizes the entropy of model predictions during testing. We use the same hyperparameters reported in the paper (Wang et al., 2019) unless otherwise specified.

**CoTTA**[5] (Wang et al., 2022). The hyperparameters of our implementations are consistent with the paper (Wang et al., 2022). The same random augmentation composition is applied, including color jitter, random affine, gaussian blur, random horizonal flip, and gaussian noise. We use 32 augmented versions for experiments and a restoration probability of $p = 0.01$.

**SAR**[6] (Niu et al., 2023). We follow all hyperparameters that are set in SAR unless it does not provide. The reset threshold $E_0$ is set to $0.2$. For ResNet50 that has 4 layer groups, we freeze the 4-th layer. For ViT-B/16 that has 11 blocks groups, we freeze the 9-th, 10-th, and 11-th blocks.

**ETA & EATA**[7] (Niu et al., 2022a). We configure the entropy threshold $E_0$ to be $0.4 \times \ln C$, where $C$ is the number of classes. The threshold of cosine similarities $\epsilon$ is set to 0.4/0.05 for CIFAR-10/ImageNet. For the anti-forgetting regularizer, its weight $\beta$ is assigned a value of $1/2000$. In total, $2,000$ samples are used to compute the anti-forgetting regularizer.

**DeYO**[8] (Lee et al., 2024). We set the entropy threshold $E_0 = 0.4 \times \ln C$ and the entropy factor $\tau_{\text{Ent}} = 0.5 \times \ln C$, where $C$ represents the number of classes. In terms of the Pseudo-Label Probability Difference (PLPD) threshold $\tau_{PLPD}$, we set $\tau_{PLPD} = 0.2$ for ViT-B/16. As for other models, it is set to $\tau_{PLPD} = 0.3$. For ResNet50 which has 4 layer groups, we freeze the 4-th layer. For ViT-B/16 which has 11 blocks groups, we freeze the 9-th, 10-th, and 11-th blocks.

**CPL**[9] (Zhang et al., 2024). CPL aims to fine-tune vision-language models with unlabeled data. Through experiments, we find that using generated pseudo-labels and multi-label loss is not suitable for TTA tasks. Therefore, the pseudo-label

---

[4] https://github.com/DequanWang/tent
[5] https://github.com/qinenergy/cotta
[6] https://github.com/mr-eggplant/SAR
[7] https://github.com/mr-eggplant/EATA
[8] https://github.com/Jhyun17/DeYO
[9] https://github.com/vanillaer/CPL-ICML2024

selection mechanism of CPL is employed to select benign samples, and then entropy minimization is used to optimize the model. The selection ratio $\beta$ is set to 0.9. The top-$K_t$, which represents the maximum number of selected samples per class, is originally set to 16 in CPL. For datasets with a data volume less than $16 \times C$ (where $C$ is the number of classes), to ensure an effective filtering, the top-$K_t$ is set to 4, 3, and 10 for ImageNet-V2, ImageNet-A, and ImageNet-R, respectively. The specified quantile $\alpha$ is set to 0.7.

**TPT**[10] (Shu et al., 2022). TPT addresses the zero-shot generalization issue of vision-language models by proposing the Test-Time Prompt Tuning method. Due to time constraints, we focus on a larger batch size for faster running speed. Specifically, we adopt a batch size of 64, which differs from the original setting of 1. Instead of augmenting a single sample 63 times to form a batch of 64, we directly input 64 samples into the model, thus achieving higher efficiency.

### C.4. Efficiency of PTTA

We evaluate the running time of different TTA methods and their PTTA-applied versions under the following configuration:

Hardware: CPU: Intel® Xeon® Silver 4210 @ 2.20GHz | GPU: NVIDIA GeForce RTX 3090 | RAM: 256GB

Software: PyTorch 1.9.0 | CUDA 11.1

All experiments use a single GPU without Automatic Mixed Precision (AMP), except: 1) CoTTA & CoTTA+PTTA employ AMP by default, 2) CoTTA & CoTTA+PTTA for ViT-B/16 leverage dual GPUs via Distributed Data Parallel (DDP).

As shown in Table 8, the PTTA-applied versions incur approximately $10\% \sim 40\%$ additional running time compared to the base TTA algorithms, primarily due to the extra forward passes required per batch. Additionally, we quantify the storage overhead of the memory bank for OOD-retrieval purposes, implemented via a first-in-first-out queue with a maximum length of $1,000$, as detailed in Table 9.

For a fair comparison, we evaluate the performance of base TTA algorithms and their PTTA-applied versions under comparable computational constraints, as shown in Table 10. Specifically, for base TTA methods, we maintain a memory bank and randomly select benign samples from it to replace malicious samples during adaptation. Consistent with PTTA's implementation, we utilize a first-in-first-out queue with a maximum length of $1,000$ to store benign samples, with additional forward passes required for randomly replaced benign samples.

Table 8: The running time (seconds) per batch for different TTA algorithms and their PTTA-applied versions. The batch size is set to 64. $\Delta$ denotes the runtime increase ratio of PTTA-applied versions compared to the base TTA methods.

| METHODS | RESNET50 | VIT-B/16 | | METHODS | RESNET50 | VIT-B/16 |
|---------|----------|----------|---|---------|----------|----------|
| TENT | 0.157 | 0.264 | | CPL | 0.157 | 0.310 |
| + PTTA | 0.219 | 0.376 | | + PTTA | 0.225 | 0.400 |
| $\Delta$ | 39.5% | 42.4% | | $\Delta$ | 43.3% | 29.0% |
| ETA | 0.158 | 0.270 | | SAR | 0.225 | 0.540 |
| + PTTA | 0.225 | 0.384 | | + PTTA | 0.328 | 0.770 |
| $\Delta$ | 42.4% | 42.2% | | $\Delta$ | 45.8% | 42.6% |
| EATA | 0.171 | 0.291 | | COTTA | 0.530 | 1.440 |
| + PTTA | 0.236 | 0.410 | | + PTTA | 0.673 | 1.597 |
| $\Delta$ | 38.0% | 40.9% | | $\Delta$ | 27.0% | 10.9% |
| DEYO | 0.177 | 0.328 | | SOTTA | 0.794 | 1.379 |
| + PTTA | 0.254 | 0.420 | | + PTTA | 0.872 | 1.597 |
| $\Delta$ | 43.5% | 28.0% | | $\Delta$ | 9.80% | 15.8% |

Table 9: The storage overhead of the memory bank for a first-in-first-out queue with a maximum length of $1,000$.

| LOGIT-SALIENCY INDICATOR | PREDICTED PROBABILITIES | RAW IMAGES |
|---------|---------|---------|
| 28.0 KB | 28.0 KB | 35.0 MB |

---

[10] https://github.com/azshue/TPT

Table 10: Comparison of base TTA algorithms with their PTTA-applied versions under comparable computational constraints.

| METHODS | SINGLE | | | | CONTINUAL | | | |
|---|---|---|---|---|---|---|---|---|
| | ETA | DEYO | CPL | AVG. | ETA | DEYO | CPL | AVG. |
| BASE ALGORITHM | 48.2 | 48.6 | 42.8 | 46.5 | 62.2 | 62.2 | 55.5 | 60.0 |
| + RANDOMLY REPLACE | 48.6 | 48.8 | 44.0 | 47.1 | 62.2 | 57.8 | 58.1 | 59.4 |
| + PTTA | **49.5** | **49.9** | **44.6** | **48.0** | **65.3** | **65.8** | **59.6** | **63.5** |

Table 11: Detailed experimental results (top-1 classification accuracy (%)) for the 1-st round of the lifelong TTA task. We highlight the highest accuracy in **bold** and the second best as underline.

| METHOD | GAUSS. | SHOT | IMPUL. | DEFOC. | GLASS | MOTION | ZOOM | SNOW | FROST | FOG | BRIT. | CONTR. | ELASTIC | PIXEL | JPEG | AVG. |
|---|---|---|---|---|---|---|---|---|---|---|---|---|---|---|---|---|
| NOADAPT | $31.6_{\pm0.00}$ | $31.6_{\pm0.00}$ | $31.6_{\pm0.00}$ | $31.6_{\pm0.00}$ | $31.6_{\pm0.00}$ | $31.6_{\pm0.00}$ | $31.6_{\pm0.00}$ | $31.6_{\pm0.00}$ | $31.6_{\pm0.00}$ | $31.6_{\pm0.00}$ | $31.6_{\pm0.00}$ | $31.6_{\pm0.00}$ | $31.6_{\pm0.00}$ | $31.6_{\pm0.00}$ | $31.6_{\pm0.00}$ | $31.6_{\pm0.00}$ |
| TENT | $44.4_{\pm0.06}$ | $51.0_{\pm0.17}$ | $22.1_{\pm0.59}$ | $2.2_{\pm0.09}$ | $0.1_{\pm0.00}$ | $0.2_{\pm0.00}$ | $0.2_{\pm0.00}$ | $0.3_{\pm0.01}$ | $0.1_{\pm0.00}$ | $0.1_{\pm0.00}$ | $0.4_{\pm0.01}$ | $0.1_{\pm0.00}$ | $0.1_{\pm0.00}$ | $0.1_{\pm0.00}$ | $0.2_{\pm0.00}$ | $8.1_{\pm0.06}$ |
| + PTTA | $44.6_{\pm0.14}$ | $51.6_{\pm0.12}$ | $55.4_{\pm0.03}$ | $51.4_{\pm0.18}$ | $53.5_{\pm0.02}$ | $58.3_{\pm0.02}$ | $57.0_{\pm0.14}$ | $58.8_{\pm0.13}$ | $63.3_{\pm0.01}$ | $69.0_{\pm0.18}$ | $76.2_{\pm0.11}$ | $64.8_{\pm0.02}$ | $58.9_{\pm0.26}$ | $69.6_{\pm0.02}$ | $68.0_{\pm0.03}$ | $60.0_{\pm0.05}$ |
| COTTA | $23.6_{\pm0.10}$ | $33.0_{\pm0.49}$ | $45.5_{\pm0.16}$ | $35.7_{\pm0.46}$ | $41.8_{\pm0.38}$ | $45.6_{\pm0.25}$ | $38.1_{\pm0.45}$ | $36.2_{\pm0.73}$ | $47.3_{\pm0.10}$ | $46.9_{\pm0.38}$ | $64.0_{\pm0.47}$ | $41.4_{\pm1.94}$ | $40.1_{\pm0.27}$ | $52.4_{\pm0.50}$ | $51.6_{\pm0.48}$ | $42.9_{\pm0.25}$ |
| + PTTA | $46.5_{\pm1.22}$ | $58.2_{\pm0.41}$ | $58.6_{\pm0.02}$ | $43.3_{\pm0.08}$ | $52.9_{\pm0.25}$ | $52.3_{\pm0.20}$ | $44.7_{\pm0.52}$ | $45.1_{\pm0.22}$ | $51.7_{\pm0.32}$ | $53.7_{\pm0.14}$ | $61.0_{\pm0.29}$ | $48.0_{\pm1.05}$ | $51.3_{\pm0.03}$ | $57.4_{\pm0.01}$ | $56.2_{\pm0.02}$ | $52.1_{\pm0.28}$ |
| SOTTA | $44.5_{\pm0.92}$ | $51.8_{\pm0.63}$ | $55.2_{\pm0.75}$ | $51.0_{\pm0.23}$ | $53.7_{\pm0.08}$ | $57.3_{\pm0.04}$ | $55.6_{\pm0.10}$ | $59.5_{\pm0.03}$ | $62.0_{\pm0.00}$ | $66.9_{\pm0.00}$ | $76.2_{\pm0.04}$ | $62.9_{\pm0.18}$ | $60.6_{\pm0.35}$ | $68.4_{\pm0.06}$ | $67.3_{\pm0.24}$ | $59.5_{\pm0.22}$ |
| + PTTA | $44.8_{\pm1.00}$ | $52.4_{\pm0.32}$ | $56.2_{\pm0.62}$ | $53.4_{\pm0.61}$ | $55.3_{\pm0.42}$ | $59.4_{\pm0.28}$ | $58.7_{\pm0.38}$ | $61.3_{\pm0.49}$ | $64.5_{\pm0.22}$ | $70.1_{\pm0.18}$ | $76.5_{\pm0.10}$ | $65.5_{\pm0.26}$ | $62.1_{\pm0.38}$ | $70.4_{\pm0.22}$ | $68.4_{\pm0.35}$ | $61.3_{\pm0.39}$ |
| SAR | $45.9_{\pm0.19}$ | $53.3_{\pm0.09}$ | $56.4_{\pm0.08}$ | $51.8_{\pm0.25}$ | $54.0_{\pm0.11}$ | $57.9_{\pm0.07}$ | $55.0_{\pm0.12}$ | $59.6_{\pm0.03}$ | $62.4_{\pm0.07}$ | $67.1_{\pm0.25}$ | $76.5_{\pm0.04}$ | $63.3_{\pm0.09}$ | $59.8_{\pm0.15}$ | $68.6_{\pm0.03}$ | $67.8_{\pm0.02}$ | $60.0_{\pm0.02}$ |
| + PTTA | $46.4_{\pm0.02}$ | $53.7_{\pm0.09}$ | $57.5_{\pm0.00}$ | $53.7_{\pm0.01}$ | $55.7_{\pm0.14}$ | $59.6_{\pm0.02}$ | $58.5_{\pm0.03}$ | $61.4_{\pm0.03}$ | $64.9_{\pm0.02}$ | $70.1_{\pm0.09}$ | $76.7_{\pm0.04}$ | $65.6_{\pm0.07}$ | $61.4_{\pm0.08}$ | $70.6_{\pm0.02}$ | $68.8_{\pm0.03}$ | $61.6_{\pm0.01}$ |
| ETA | $53.3_{\pm0.01}$ | $57.6_{\pm0.05}$ | $58.3_{\pm0.13}$ | $53.6_{\pm0.08}$ | $56.9_{\pm0.18}$ | $59.4_{\pm0.05}$ | $59.2_{\pm0.18}$ | $62.1_{\pm0.11}$ | $63.1_{\pm0.48}$ | $68.8_{\pm0.07}$ | $75.8_{\pm0.02}$ | $62.8_{\pm0.52}$ | $64.3_{\pm0.17}$ | $70.1_{\pm0.05}$ | $67.6_{\pm0.07}$ | $62.2_{\pm0.07}$ |
| + PTTA | $54.0_{\pm0.07}$ | $59.7_{\pm0.09}$ | $60.5_{\pm0.08}$ | $57.0_{\pm0.13}$ | $60.2_{\pm0.10}$ | $63.5_{\pm0.20}$ | $63.6_{\pm0.09}$ | $65.7_{\pm0.09}$ | $66.4_{\pm0.07}$ | $72.4_{\pm0.05}$ | $77.6_{\pm0.01}$ | $67.5_{\pm0.14}$ | $67.5_{\pm0.33}$ | $73.1_{\pm0.03}$ | $70.7_{\pm0.09}$ | $65.3_{\pm0.06}$ |
| EATA | $52.0_{\pm0.42}$ | $55.4_{\pm0.83}$ | $57.0_{\pm0.52}$ | $54.0_{\pm0.33}$ | $56.6_{\pm0.26}$ | $59.8_{\pm0.13}$ | $58.9_{\pm0.29}$ | $62.7_{\pm0.89}$ | $64.3_{\pm0.56}$ | $70.7_{\pm0.31}$ | $76.3_{\pm0.43}$ | $66.3_{\pm0.01}$ | $64.5_{\pm0.58}$ | $70.4_{\pm0.34}$ | $68.9_{\pm0.33}$ | $62.5_{\pm0.40}$ |
| + PTTA | $52.3_{\pm0.32}$ | $57.4_{\pm0.52}$ | $58.8_{\pm0.31}$ | $56.4_{\pm0.28}$ | $59.0_{\pm0.36}$ | $62.6_{\pm0.10}$ | $62.5_{\pm0.46}$ | $65.7_{\pm0.50}$ | $67.0_{\pm0.26}$ | $73.0_{\pm0.03}$ | $77.7_{\pm0.30}$ | $68.5_{\pm0.25}$ | $66.1_{\pm0.42}$ | $72.5_{\pm0.32}$ | $70.3_{\pm0.34}$ | $64.7_{\pm0.31}$ |
| DEYO | $54.4_{\pm0.15}$ | $58.6_{\pm0.06}$ | $59.4_{\pm0.07}$ | $54.8_{\pm0.29}$ | $58.1_{\pm0.02}$ | $61.4_{\pm0.04}$ | $37.6_{\pm4.81}$ | $61.1_{\pm3.04}$ | $64.1_{\pm0.39}$ | $70.4_{\pm0.27}$ | $76.8_{\pm0.16}$ | $64.8_{\pm0.06}$ | $67.3_{\pm0.02}$ | $71.9_{\pm0.06}$ | $69.5_{\pm0.08}$ | $62.0_{\pm0.50}$ |
| + PTTA | $55.0_{\pm0.03}$ | $59.8_{\pm0.02}$ | $60.5_{\pm0.04}$ | $57.0_{\pm0.09}$ | $60.4_{\pm0.15}$ | $64.0_{\pm0.12}$ | $63.7_{\pm0.37}$ | $66.6_{\pm0.13}$ | $67.0_{\pm0.15}$ | $73.0_{\pm0.02}$ | $78.0_{\pm0.03}$ | $67.8_{\pm0.23}$ | $68.7_{\pm0.30}$ | $73.7_{\pm0.17}$ | $71.2_{\pm0.04}$ | $65.8_{\pm0.01}$ |
| CPL | $42.4_{\pm0.19}$ | $47.9_{\pm0.14}$ | $52.5_{\pm0.11}$ | $46.0_{\pm0.41}$ | $47.4_{\pm0.45}$ | $53.4_{\pm0.14}$ | $50.7_{\pm0.22}$ | $54.9_{\pm0.41}$ | $57.8_{\pm0.17}$ | $60.6_{\pm0.44}$ | $75.1_{\pm0.07}$ | $56.1_{\pm0.32}$ | $57.1_{\pm0.21}$ | $65.1_{\pm0.11}$ | $65.3_{\pm0.12}$ | $55.5_{\pm0.13}$ |
| + PTTA | $42.8_{\pm0.15}$ | $49.6_{\pm0.15}$ | $54.6_{\pm0.10}$ | $51.9_{\pm0.24}$ | $52.8_{\pm0.10}$ | $57.6_{\pm0.03}$ | $56.2_{\pm0.10}$ | $59.4_{\pm0.04}$ | $64.6_{\pm0.05}$ | $69.4_{\pm0.04}$ | $76.2_{\pm0.02}$ | $64.6_{\pm0.15}$ | $59.2_{\pm0.07}$ | $68.8_{\pm0.02}$ | $67.6_{\pm0.07}$ | $59.7_{\pm0.02}$ |

# D. More Experimental Results

We conduct experiments on four types of TTA tasks, including lifelong, single, continual, and episodic tasks, and semantic segmentation as well as adversarial defense. In the following, we present the detailed experimental results of the baselines and our proposed PTTA for each task.

## D.1. Lifelong Test-Time Adaptation

We conduct experiments in a dynamically changing environment with 150 corruptions over 10 consecutive rounds, continuously updating the parameters of DNNs without resetting. We report the detailed experimental results of the 1-st, 5-th, and 10-th rounds in Tables 11, 12, and 13. Meanwhile, the average accuracy of each round is reported in Table 2. ETA applies prediction entropy and sample diversity as criteria for screening benign samples. This kind of sample selection method achieves excellent performance in short-term environmental changes. However, with the continuous change of the environments, the accuracy of the model will experience catastrophic degradation, *i.e.*, 62.2% in the 1-st round and 25.1% in the 10-th round. The integration of our PTTA can effectively alleviate the performance degradation problem of ETA, with 65.3% in the 1-st round and 64.1% in the 10-th round. EATA mitigates the model's catastrophic forgetting as well by introducing the Fisher information of in-distribution data for supervision, achieving 62.5% accuracy in the 1-st round and 61.0% in the 10-th round. However, applying PTTA to EATA enhances the utilization of test data, improves the calibration of DNNs, and reduces over-confident predictions. We find that using the prior information of data distribution along with purification strategy can address the issue of performance decline in lifelong tasks, *i.e.*, 64.7% accuracy in the 1-st round and 65.0% in the 10-th round. This conclusion is verified again when PTTA is applied to CPL.

## D.2. Single & Continual Test-Time Adaptation

We conduct experiments on single and continual TTA tasks with ResNet and ViT architectures to verify the generality of our proposed PTTA for different DNNs. The experimental results are shown in Table 14 and Fig. 9. Compared with continual TTA, the single task places more emphasis on the adaptability of TTA algorithms in static distribution shifts. Since

Table 12: Detailed experimental results (top-1 classification accuracy (%)) for the 5-th round of the lifelong TTA task. We highlight the highest accuracy in **bold** and the second best as underline..

| | $t$ $\longrightarrow$ | | | | | | | | | | | | | | | |
| METHOD | GAUSS. | SHOT | IMPUL. | DEFOC. | GLASS | MOTION | ZOOM | SNOW | FROST | FOG | BRIT. | CONTR. | ELASTIC | PIXEL | JPEG | AVG. |
|---|---|---|---|---|---|---|---|---|---|---|---|---|---|---|---|---|
| NoAdapt | 31.6±0.00 | 31.6±0.00 | 31.6±0.00 | 31.6±0.00 | 31.6±0.00 | 31.6±0.00 | 31.6±0.00 | 31.6±0.00 | 31.6±0.00 | 31.6±0.00 | 31.6±0.00 | 31.6±0.00 | 31.6±0.00 | 31.6±0.00 | 31.6±0.00 | 31.6±0.00 |
| Tent | 0.1±0.00 | 0.1±0.00 | 0.1±0.00 | 0.1±0.00 | 0.1±0.00 | 0.1±0.00 | 0.1±0.00 | 0.1±0.00 | 0.1±0.00 | 0.1±0.00 | 0.1±0.00 | 0.1±0.00 | 0.1±0.00 | 0.1±0.00 | 0.1±0.00 | 0.1±0.00 |
| + PTTA | 0.1±0.00 | 0.1±0.00 | 0.1±0.00 | 0.1±0.00 | 0.1±0.00 | 0.1±0.00 | 0.1±0.00 | 0.1±0.00 | 0.1±0.00 | 0.1±0.00 | 0.1±0.01 | 0.1±0.00 | 0.1±0.00 | 0.1±0.00 | 0.1±0.00 | 0.1±0.00 |
| CoTTA | 27.6±1.05 | 34.1±1.89 | 37.0±0.83 | 26.2±1.07 | 32.3±0.62 | 32.5±0.98 | 29.1±1.09 | 28.9±2.03 | 34.3±1.47 | 34.8±0.79 | 47.9±1.66 | 30.6±0.69 | 30.2±2.12 | 38.7±1.24 | 37.9±1.01 | 33.5±1.02 |
| + PTTA | 38.5±0.15 | 42.1±1.04 | 42.3±0.07 | 29.9±0.45 | 38.5±0.31 | 38.0±0.20 | 33.9±0.22 | 32.1±0.10 | 38.5±0.05 | 40.7±0.20 | 48.7±0.13 | 36.1±0.44 | 39.0±0.23 | 45.3±0.20 | 44.7±0.12 | 39.2±0.09 |
| SoTTA | 52.1±0.54 | 54.1±0.29 | 55.5±0.65 | 54.7±0.12 | 56.8±0.13 | 59.3±0.03 | 57.0±0.10 | 62.2±0.01 | 62.3±0.00 | 67.9±0.07 | 75.7±0.01 | 62.6±0.13 | 63.7±0.10 | 70.2±0.00 | 67.6±0.14 | 61.5±0.13 |
| + PTTA | 54.2±0.67 | 56.1±0.61 | 57.2±0.70 | 56.7±0.00 | 58.5±0.31 | 61.6±0.22 | 61.0±0.32 | 64.3±0.16 | 64.8±0.27 | 69.9±0.01 | 76.2±0.06 | 66.1±0.18 | 65.4±0.16 | 72.0±0.22 | 68.8±0.21 | 63.5±0.27 |
| SAR | 53.2±0.04 | 54.9±0.09 | 56.3±0.04 | 55.3±0.10 | 56.9±0.05 | 59.8±0.08 | 55.8±0.16 | 62.4±0.04 | 62.2±0.04 | 68.2±0.15 | 75.8±0.05 | 63.0±0.05 | 63.5±0.09 | 70.4±0.05 | 68.0±0.06 | 61.7±0.02 |
| + PTTA | 55.0±0.07 | 56.8±0.02 | 58.2±0.00 | 57.1±0.03 | 58.6±0.09 | 61.7±0.01 | 61.1±0.07 | 64.5±0.01 | 64.8±0.02 | 70.1±0.09 | 76.3±0.01 | 66.0±0.05 | 65.3±0.03 | 72.1±0.04 | 69.2±0.01 | 63.8±0.02 |
| ETA | 27.9±24.1 | 29.9±25.8 | 29.5±25.4 | 25.4±21.9 | 28.0±24.2 | 30.1±26.1 | 30.0±25.9 | 30.9±26.7 | 32.2±27.8 | 33.3±28.8 | 44.1±38.1 | 21.2±18.3 | 33.8±29.1 | 39.9±34.5 | 36.9±31.9 | 31.5±27.2 |
| + PTTA | 56.7±0.03 | 58.7±0.04 | 59.3±0.14 | 57.8±0.15 | 59.9±0.12 | 63.3±0.24 | 63.4±0.07 | 65.4±0.08 | 65.3±0.15 | 71.2±0.04 | 76.4±0.05 | 66.9±0.16 | 67.7±0.20 | 72.6±0.10 | 69.7±0.09 | 64.9±0.03 |
| EATA | 51.6±0.65 | 53.6±0.83 | 55.4±0.73 | 52.9±0.65 | 55.4±0.57 | 58.2±0.11 | 57.7±0.45 | 61.6±0.94 | 63.4±0.68 | 69.8±0.46 | 75.8±0.51 | 65.3±0.56 | 64.2±0.53 | 70.2±0.45 | 68.6±0.31 | 61.6±0.48 |
| + PTTA | 55.1±0.54 | 57.6±0.70 | 58.9±0.39 | 57.0±0.53 | 59.4±0.43 | 62.9±0.26 | 62.8±0.54 | 66.3±0.58 | 67.0±0.26 | 72.8±0.07 | 77.7±0.31 | 68.6±0.17 | 66.6±0.76 | 72.7±0.31 | 70.5±0.32 | 65.1±0.40 |
| DeYO | 17.4±29.9 | 17.8±30.6 | 17.7±30.5 | 12.5±21.4 | 15.8±27.2 | 18.1±31.2 | 4.6±7.87 | 15.6±26.9 | 18.3±31.5 | 21.2±36.6 | 25.0±43.1 | 19.3±33.3 | 19.9±34.2 | 22.7±39.1 | 22.0±37.8 | 17.9±30.8 |
| + PTTA | 57.2±0.03 | 59.7±0.03 | 60.1±0.10 | 58.1±0.20 | 60.8±0.13 | 64.5±0.04 | 64.1±0.46 | 66.7±0.11 | 66.2±0.05 | 72.2±0.13 | 77.2±0.09 | 67.4±0.13 | 69.5±0.12 | 73.5±0.07 | 70.7±0.08 | **65.9**±0.05 |
| CPL | 50.3±0.12 | 50.1±0.25 | 52.7±0.09 | 51.6±0.29 | 52.0±0.18 | 56.8±0.34 | 53.8±0.25 | 60.6±0.18 | 60.2±0.09 | 62.0±0.50 | 74.8±0.15 | 56.2±2.15 | 60.1±0.11 | 68.2±0.15 | 65.8±0.12 | 58.4±0.27 |
| + PTTA | 53.4±0.04 | 54.6±0.10 | 56.6±0.05 | 55.4±0.02 | 56.8±0.12 | 60.3±0.06 | 59.2±0.05 | 63.5±0.02 | 65.4±0.04 | 69.8±0.06 | 76.2±0.01 | 64.8±0.11 | 63.2±0.08 | 71.0±0.06 | 68.4±0.07 | 62.6±0.02 |

Table 13: Detailed experimental results (top-1 classification accuracy (%)) for the 10-th round of the lifelong TTA task. We highlight the highest accuracy in **bold** and the second best as underline.

| | $t$ $\longrightarrow$ | | | | | | | | | | | | | | | |
| METHOD | GAUSS. | SHOT | IMPUL. | DEFOC. | GLASS | MOTION | ZOOM | SNOW | FROST | FOG | BRIT. | CONTR. | ELASTIC | PIXEL | JPEG | AVG. |
|---|---|---|---|---|---|---|---|---|---|---|---|---|---|---|---|---|
| NoAdapt | 31.6±0.00 | 31.6±0.00 | 31.6±0.00 | 31.6±0.00 | 31.6±0.00 | 31.6±0.00 | 31.6±0.00 | 31.6±0.00 | 31.6±0.00 | 31.6±0.00 | 31.6±0.00 | 31.6±0.00 | 31.6±0.00 | 31.6±0.00 | 31.6±0.00 | 31.6±0.00 |
| Tent | 0.1±0.00 | 0.1±0.00 | 0.1±0.00 | 0.1±0.00 | 0.1±0.00 | 0.1±0.00 | 0.1±0.00 | 0.1±0.00 | 0.1±0.00 | 0.1±0.00 | 0.1±0.00 | 0.1±0.00 | 0.1±0.00 | 0.1±0.00 | 0.1±0.00 | 0.1±0.00 |
| + PTTA | 0.1±0.00 | 0.1±0.00 | 0.1±0.00 | 0.1±0.00 | 0.1±0.00 | 0.1±0.00 | 0.1±0.00 | 0.1±0.00 | 0.1±0.00 | 0.1±0.00 | 0.1±0.00 | 0.1±0.00 | 0.1±0.00 | 0.1±0.00 | 0.1±0.00 | 0.1±0.00 |
| CoTTA | 25.7±1.04 | 31.4±0.97 | 33.8±1.11 | 23.8±0.84 | 29.6±0.85 | 29.8±1.02 | 26.6±1.13 | 26.3±0.55 | 31.2±0.59 | 31.9±0.61 | 43.2±0.89 | 28.4±1.85 | 27.0±0.71 | 35.1±1.18 | 34.9±1.29 | 30.6±0.90 |
| + PTTA | 36.9±0.25 | 40.1±0.12 | 40.4±0.01 | 28.9±0.26 | 36.9±0.09 | 36.5±0.06 | 32.9±0.16 | 31.0±0.39 | 36.7±0.12 | 39.0±0.19 | 46.1±0.09 | 35.8±0.13 | 37.0±0.28 | 43.5±0.12 | 43.0±0.08 | 37.6±0.08 |
| SoTTA | 53.4±0.65 | 54.8±0.47 | 55.8±0.72 | 55.1±0.10 | 57.1±0.19 | 60.0±0.34 | 57.6±0.20 | 62.7±0.10 | 62.2±0.02 | 68.1±0.02 | 75.4±0.10 | 62.8±0.18 | 64.4±0.24 | 70.8±0.03 | 67.8±0.19 | 61.9±0.20 |
| + PTTA | 55.1±0.59 | 56.7±0.46 | 57.7±0.57 | 57.3±0.11 | 58.8±0.41 | 62.2±0.17 | 61.8±0.31 | 64.8±0.00 | 64.8±0.23 | 70.0±0.04 | 75.9±0.03 | 66.4±0.22 | 66.3±0.21 | 72.4±0.21 | 69.0±0.10 | 64.0±0.24 |
| SAR | 47.9±0.50 | 53.0±0.51 | 55.9±0.43 | 52.6±0.15 | 54.3±0.16 | 58.0±0.27 | 54.7±0.44 | 59.9±0.34 | 62.4±0.09 | 67.6±0.68 | 76.4±0.12 | 63.4±0.31 | 61.0±0.53 | 69.9±0.02 | 68.2±0.12 | 60.4±0.06 |
| + PTTA | 55.9±0.03 | 57.4±0.01 | 58.5±0.02 | 57.3±0.09 | 59.0±0.03 | 62.5±0.03 | 61.7±0.03 | 64.9±0.02 | 64.8±0.04 | 69.9±0.07 | 75.9±0.05 | 66.3±0.01 | 66.0±0.03 | 72.4±0.06 | 69.2±0.00 | 64.1±0.02 |
| ETA | 21.5±18.5 | 23.2±20.1 | 23.7±20.4 | 20.0±17.4 | 23.0±19.9 | 24.3±21.0 | 22.0±19.0 | 24.6±21.3 | 24.7±21.3 | 24.6±21.4 | 38.0±32.9 | 14.0±12.0 | 28.6±24.7 | 33.6±29.1 | 30.5±26.4 | 25.1±21.6 |
| + PTTA | 55.9±0.09 | 57.6±0.22 | 58.3±0.16 | 57.3±0.17 | 59.5±0.13 | 62.7±0.05 | 62.8±0.12 | 64.4±0.07 | 64.5±0.18 | 70.2±0.07 | 75.6±0.07 | 65.7±0.20 | 67.0±0.14 | 71.9±0.21 | 68.8±0.11 | 64.1±0.04 |
| EATA | 50.9±0.15 | 52.5±0.72 | 54.7±0.34 | 52.3±0.22 | 54.9±0.27 | 57.8±0.38 | 57.1±0.48 | 60.8±0.97 | 62.8±0.75 | 69.2±0.15 | 75.6±0.47 | 64.8±0.24 | 63.5±0.77 | 69.8±0.45 | 68.1±0.30 | 61.0±0.35 |
| + PTTA | 55.1±0.57 | 57.7±0.81 | 58.9±0.40 | 56.8±0.32 | 59.5±0.26 | 62.9±0.38 | 62.7±0.59 | 65.9±0.63 | 66.9±0.45 | 72.8±0.19 | 77.6±0.25 | 68.4±0.13 | 66.6±0.68 | 72.6±0.36 | 70.5±0.34 | **65.0**±0.41 |
| DeYO | 0.1±0.00 | 0.1±0.00 | 0.1±0.00 | 0.1±0.00 | 0.1±0.00 | 0.1±0.00 | 0.1±0.00 | 0.1±0.01 | 0.1±0.01 | 0.1±0.00 | 0.1±0.00 | 0.1±0.00 | 0.1±0.01 | 0.1±0.01 | 0.1±0.01 | 0.1±0.00 |
| + PTTA | 56.6±0.37 | 59.0±0.25 | 59.3±0.14 | 55.4±4.37 | 56.5±3.92 | 57.7±5.07 | 54.3±7.34 | 61.8±3.73 | 62.2±4.38 | 67.0±6.34 | 75.1±2.41 | 63.2±5.78 | 61.0±9.82 | 69.4±5.22 | 66.4±5.98 | 61.6±4.30 |
| CPL | 51.6±0.09 | 51.3±0.07 | 53.2±0.11 | 52.2±0.30 | 52.4±0.35 | 57.2±0.37 | 54.3±0.21 | 61.6±0.12 | 60.5±0.11 | 61.7±0.38 | 74.7±0.02 | 56.4±1.39 | 60.8±0.18 | 68.8±0.10 | 65.8±0.10 | 58.8±0.21 |
| + PTTA | 55.1±0.04 | 55.5±0.09 | 57.2±0.10 | 56.1±0.10 | 57.4±0.05 | 60.9±0.05 | 60.0±0.08 | 64.2±0.09 | 65.6±0.04 | 70.0±0.05 | 76.0±0.08 | 65.1±0.07 | 64.3±0.10 | 71.4±0.06 | 68.7±0.00 | 63.2±0.03 |

the dataset of a single target domain provides limited test samples, it is crucial to make full use of all samples. This can improve the optimization efficiency of TTA algorithms and avoid getting trapped in a sub-optimal state. Applying PTTA to Tent, ETA, EATA, DeYO, and CPL leads to an average accuracy increase of 1.5% for ResNet50 and 1.2% for ViT-B/16. As ResNet50 is significantly less robust to distribution shifts than ViT-B/16, common TTA algorithms such as Tent and DeYO experience severe performance degradation in continual tasks when using ResNet50. PTTA can address this issue by improving the calibration of DNNs and reducing over-confident predictions, enabling Tent and DeYO to achieve accuracy increases of 37.0% and 26.5% respectively. We further validate the above conclusions using WRN-40 on CIFAR100-C, as shown in Table 15.

### D.3. Episodic Test-Time Prompt Tuning

We conduct experiments on the episodic test-time prompt tuning task using CLIP models. Following TPT, we only update the learnable text prompt of text encoders for single-batch samples. The experimental results are referred to Table 4. TPT uses the default "a photo of a {CLASS}" or the text prompt learned from CoOp (Zhou et al., 2022) as the initial

Table 14: Detailed experimental results (top-1 classification accuracy (%)) on single TTA task. We highlight the highest accuracy in **bold** and the second best as underline.

| METHOD | NOISE | | | BLUR | | | | WEATHER | | | | DIGITAL | | | | AVG. |
|---|---|---|---|---|---|---|---|---|---|---|---|---|---|---|---|---|
| | Gauss. | Shot | Impul. | Defoc. | Glass | Motion | Zoom | Snow | Frost | Fog | Brit. | Contr. | Elastic | Pixel | JPEG | |
| RESNET50 | | | | | | | | | | | | | | | | |
| NOADAPT | $3.0_{\pm0.00}$ | $3.7_{\pm0.00}$ | $2.6_{\pm0.00}$ | $17.9_{\pm0.00}$ | $9.7_{\pm0.00}$ | $14.7_{\pm0.00}$ | $22.5_{\pm0.00}$ | $16.6_{\pm0.00}$ | $23.1_{\pm0.00}$ | $24.0_{\pm0.00}$ | $59.1_{\pm0.00}$ | $5.4_{\pm0.00}$ | $16.5_{\pm0.00}$ | $20.9_{\pm0.00}$ | $32.6_{\pm0.00}$ | $18.2_{\pm0.00}$ |
| TENT | $29.7_{\pm0.08}$ | $31.5_{\pm0.04}$ | $31.1_{\pm0.23}$ | $28.1_{\pm0.09}$ | $26.9_{\pm0.07}$ | $41.2_{\pm0.21}$ | $49.3_{\pm0.12}$ | $47.1_{\pm0.09}$ | $40.9_{\pm0.09}$ | $57.4_{\pm0.10}$ | $67.4_{\pm0.06}$ | $26.4_{\pm0.11}$ | $54.8_{\pm0.09}$ | $58.4_{\pm0.10}$ | $52.4_{\pm0.08}$ | $42.8_{\pm0.04}$ |
| + PTTA | $31.6_{\pm0.04}$ | $31.8_{\pm0.31}$ | $32.9_{\pm0.19}$ | $29.8_{\pm0.22}$ | $28.8_{\pm0.17}$ | $43.2_{\pm0.26}$ | $50.5_{\pm0.07}$ | $48.2_{\pm0.13}$ | $42.8_{\pm0.17}$ | $58.2_{\pm0.06}$ | $67.6_{\pm0.09}$ | $41.4_{\pm0.22}$ | $55.3_{\pm0.07}$ | $59.1_{\pm0.11}$ | $52.9_{\pm0.07}$ | $44.9_{\pm0.05}$ |
| COTTA | $17.1_{\pm0.05}$ | $17.9_{\pm0.13}$ | $17.9_{\pm0.06}$ | $16.1_{\pm0.11}$ | $16.3_{\pm0.10}$ | $27.8_{\pm0.05}$ | $40.4_{\pm0.02}$ | $35.6_{\pm0.07}$ | $34.1_{\pm0.13}$ | $49.5_{\pm0.16}$ | $65.8_{\pm0.10}$ | $17.9_{\pm0.08}$ | $45.3_{\pm0.09}$ | $50.5_{\pm0.07}$ | $41.6_{\pm0.11}$ | $32.9_{\pm0.01}$ |
| + PTTA | $17.9_{\pm0.19}$ | $18.2_{\pm0.20}$ | $19.1_{\pm0.37}$ | $16.0_{\pm1.33}$ | $16.5_{\pm0.69}$ | $29.7_{\pm1.20}$ | $42.1_{\pm1.04}$ | $36.9_{\pm1.02}$ | $35.3_{\pm0.75}$ | $50.9_{\pm1.38}$ | $66.2_{\pm0.36}$ | $20.1_{\pm1.40}$ | $46.1_{\pm0.35}$ | $52.1_{\pm1.21}$ | $42.8_{\pm0.84}$ | $34.0_{\pm0.53}$ |
| SOTTA | $27.4_{\pm0.82}$ | $29.9_{\pm0.11}$ | $28.5_{\pm0.24}$ | $25.7_{\pm0.07}$ | $25.2_{\pm0.56}$ | $38.4_{\pm0.15}$ | $47.6_{\pm0.09}$ | $45.4_{\pm0.41}$ | $40.3_{\pm0.17}$ | $56.1_{\pm0.08}$ | $66.9_{\pm0.09}$ | $28.1_{\pm0.26}$ | $53.9_{\pm0.26}$ | $57.5_{\pm0.00}$ | $51.4_{\pm0.03}$ | $41.5_{\pm0.01}$ |
| + PTTA | $29.1_{\pm1.36}$ | $31.4_{\pm1.08}$ | $29.9_{\pm1.07}$ | $27.4_{\pm0.01}$ | $26.8_{\pm0.06}$ | $39.9_{\pm0.72}$ | $48.3_{\pm0.17}$ | $46.2_{\pm0.21}$ | $41.3_{\pm0.25}$ | $56.5_{\pm0.05}$ | $66.6_{\pm0.07}$ | $37.1_{\pm1.55}$ | $54.0_{\pm0.23}$ | $57.4_{\pm0.12}$ | $51.4_{\pm0.29}$ | $42.9_{\pm0.36}$ |
| SAR | $31.4_{\pm0.28}$ | $31.6_{\pm0.24}$ | $32.4_{\pm0.44}$ | $29.0_{\pm0.25}$ | $28.3_{\pm0.48}$ | $41.9_{\pm0.11}$ | $49.3_{\pm0.28}$ | $47.3_{\pm0.03}$ | $42.0_{\pm0.17}$ | $57.4_{\pm0.02}$ | $67.4_{\pm0.17}$ | $36.9_{\pm2.34}$ | $54.5_{\pm0.08}$ | $58.4_{\pm0.12}$ | $52.3_{\pm0.09}$ | $44.0_{\pm0.15}$ |
| + PTTA | $32.5_{\pm2.12}$ | $32.7_{\pm1.60}$ | $33.4_{\pm1.37}$ | $31.5_{\pm1.39}$ | $30.8_{\pm1.48}$ | $44.2_{\pm1.16}$ | $50.7_{\pm0.84}$ | $48.5_{\pm0.36}$ | $43.6_{\pm0.53}$ | $58.1_{\pm0.44}$ | $67.5_{\pm0.20}$ | $41.5_{\pm2.92}$ | $55.2_{\pm0.02}$ | $59.3_{\pm0.44}$ | $52.5_{\pm0.06}$ | $45.5_{\pm0.93}$ |
| ETA | $36.0_{\pm0.02}$ | $38.2_{\pm0.23}$ | $37.5_{\pm0.23}$ | $33.4_{\pm0.27}$ | $32.8_{\pm0.09}$ | $47.7_{\pm0.26}$ | $52.6_{\pm0.08}$ | $51.9_{\pm0.14}$ | $45.8_{\pm0.19}$ | $59.9_{\pm0.04}$ | $67.8_{\pm0.13}$ | $45.4_{\pm0.19}$ | $57.9_{\pm0.07}$ | $60.8_{\pm0.10}$ | $55.1_{\pm0.05}$ | $48.2_{\pm0.04}$ |
| + PTTA | $37.5_{\pm0.14}$ | $39.6_{\pm0.11}$ | $39.0_{\pm0.11}$ | $35.4_{\pm0.08}$ | $34.5_{\pm0.16}$ | $48.9_{\pm0.26}$ | $53.7_{\pm0.20}$ | $52.8_{\pm0.05}$ | $47.1_{\pm0.19}$ | $60.9_{\pm0.10}$ | $68.4_{\pm0.14}$ | $48.2_{\pm0.32}$ | $58.5_{\pm0.06}$ | $61.5_{\pm0.18}$ | $55.9_{\pm0.05}$ | $49.5_{\pm0.00}$ |
| EATA | $35.5_{\pm0.31}$ | $37.6_{\pm0.18}$ | $36.8_{\pm0.13}$ | $33.3_{\pm0.23}$ | $33.2_{\pm0.09}$ | $47.2_{\pm0.15}$ | $52.7_{\pm0.03}$ | $51.6_{\pm0.12}$ | $45.5_{\pm0.26}$ | $59.9_{\pm0.01}$ | $68.1_{\pm0.11}$ | $44.6_{\pm0.06}$ | $58.0_{\pm0.07}$ | $60.6_{\pm0.10}$ | $55.1_{\pm0.04}$ | $48.0_{\pm0.06}$ |
| + PTTA | $37.1_{\pm0.26}$ | $39.0_{\pm0.24}$ | $38.3_{\pm0.22}$ | $35.0_{\pm0.34}$ | $34.7_{\pm0.22}$ | $48.6_{\pm0.16}$ | $53.6_{\pm0.05}$ | $52.5_{\pm0.09}$ | $46.7_{\pm0.20}$ | $60.6_{\pm0.09}$ | $68.4_{\pm0.09}$ | $47.3_{\pm0.13}$ | $58.3_{\pm0.17}$ | $61.2_{\pm0.11}$ | $55.7_{\pm0.05}$ | $49.1_{\pm0.08}$ |
| DEYO | $36.5_{\pm0.26}$ | $38.8_{\pm0.05}$ | $38.2_{\pm0.11}$ | $33.9_{\pm0.27}$ | $33.5_{\pm0.27}$ | $48.6_{\pm0.18}$ | $52.7_{\pm0.15}$ | $52.6_{\pm0.27}$ | $46.3_{\pm0.06}$ | $60.3_{\pm0.08}$ | $68.2_{\pm0.07}$ | $44.4_{\pm1.99}$ | $58.5_{\pm0.19}$ | $61.5_{\pm0.13}$ | $55.7_{\pm0.10}$ | $48.6_{\pm0.11}$ |
| + PTTA | $37.9_{\pm0.08}$ | $39.7_{\pm0.63}$ | $39.6_{\pm0.03}$ | $35.4_{\pm0.23}$ | $35.7_{\pm0.30}$ | $49.9_{\pm0.21}$ | $53.8_{\pm0.09}$ | $53.5_{\pm0.27}$ | $47.5_{\pm0.11}$ | $61.3_{\pm0.11}$ | $68.6_{\pm0.06}$ | $48.2_{\pm0.59}$ | $59.2_{\pm0.04}$ | $62.1_{\pm0.19}$ | $56.5_{\pm0.02}$ | **$49.9_{\pm0.10}$** |
| CPL | $29.9_{\pm0.15}$ | $32.0_{\pm0.08}$ | $30.8_{\pm0.16}$ | $27.8_{\pm0.14}$ | $27.7_{\pm0.13}$ | $40.0_{\pm0.13}$ | $48.3_{\pm0.14}$ | $45.1_{\pm0.19}$ | $41.1_{\pm0.14}$ | $55.9_{\pm0.09}$ | $66.8_{\pm0.11}$ | $35.0_{\pm0.20}$ | $54.1_{\pm0.13}$ | $56.8_{\pm0.08}$ | $50.6_{\pm0.13}$ | $42.8_{\pm0.04}$ |
| + PTTA | $31.9_{\pm0.30}$ | $32.8_{\pm0.11}$ | $33.1_{\pm0.11}$ | $29.5_{\pm0.25}$ | $29.4_{\pm0.20}$ | $42.5_{\pm0.20}$ | $50.0_{\pm0.16}$ | $46.8_{\pm0.22}$ | $42.9_{\pm0.07}$ | $57.2_{\pm0.08}$ | $67.1_{\pm0.10}$ | $41.2_{\pm0.10}$ | $55.0_{\pm0.17}$ | $58.3_{\pm0.08}$ | $51.5_{\pm0.11}$ | $44.6_{\pm0.06}$ |
| VIT-B/16 | | | | | | | | | | | | | | | | |
| NOADAPT | $16.8_{\pm0.00}$ | $12.0_{\pm0.00}$ | $16.5_{\pm0.00}$ | $29.2_{\pm0.00}$ | $23.6_{\pm0.00}$ | $33.9_{\pm0.00}$ | $27.3_{\pm0.00}$ | $15.8_{\pm0.00}$ | $26.6_{\pm0.00}$ | $47.5_{\pm0.00}$ | $55.4_{\pm0.00}$ | $44.3_{\pm0.00}$ | $31.0_{\pm0.00}$ | $45.1_{\pm0.00}$ | $49.1_{\pm0.00}$ | $31.6_{\pm0.00}$ |
| TENT | $44.4_{\pm0.10}$ | $42.7_{\pm0.09}$ | $45.3_{\pm0.18}$ | $52.2_{\pm0.15}$ | $47.7_{\pm0.04}$ | $55.4_{\pm0.15}$ | $50.5_{\pm0.00}$ | $18.5_{\pm1.51}$ | $21.7_{\pm2.78}$ | $66.4_{\pm0.03}$ | $74.9_{\pm0.14}$ | $64.7_{\pm0.06}$ | $53.2_{\pm0.11}$ | $66.9_{\pm0.18}$ | $64.5_{\pm0.16}$ | $51.3_{\pm0.31}$ |
| + PTTA | $44.7_{\pm0.15}$ | $39.0_{\pm6.88}$ | $46.0_{\pm0.15}$ | $52.1_{\pm0.17}$ | $47.9_{\pm0.14}$ | $55.9_{\pm0.13}$ | $51.8_{\pm0.15}$ | $18.7_{\pm2.73}$ | $31.6_{\pm5.02}$ | $67.4_{\pm0.21}$ | $75.0_{\pm0.08}$ | $65.5_{\pm0.02}$ | $54.3_{\pm0.05}$ | $67.4_{\pm0.07}$ | $64.3_{\pm0.21}$ | $52.1_{\pm0.26}$ |
| COTTA | $23.6_{\pm0.10}$ | $23.7_{\pm2.36}$ | $27.6_{\pm3.17}$ | $30.6_{\pm0.04}$ | $25.9_{\pm0.05}$ | $38.4_{\pm0.47}$ | $30.9_{\pm0.23}$ | $28.6_{\pm7.74}$ | $41.5_{\pm2.35}$ | $51.7_{\pm0.16}$ | $64.3_{\pm0.17}$ | $49.6_{\pm0.17}$ | $36.2_{\pm1.55}$ | $51.4_{\pm0.55}$ | $53.6_{\pm0.55}$ | $38.5_{\pm0.69}$ |
| + PTTA | $46.5_{\pm1.72}$ | $24.3_{\pm31.5}$ | $46.4_{\pm1.64}$ | $40.5_{\pm1.14}$ | $39.3_{\pm0.83}$ | $49.8_{\pm0.14}$ | $40.7_{\pm0.98}$ | $27.3_{\pm33.0}$ | $49.7_{\pm2.23}$ | $59.1_{\pm0.27}$ | $72.3_{\pm1.39}$ | $58.8_{\pm0.53}$ | $45.5_{\pm0.35}$ | $63.2_{\pm1.06}$ | $61.9_{\pm0.35}$ | $48.4_{\pm4.65}$ |
| SOTTA | $44.4_{\pm1.30}$ | $43.4_{\pm1.35}$ | $45.8_{\pm1.07}$ | $52.6_{\pm0.14}$ | $48.9_{\pm0.28}$ | $56.1_{\pm0.25}$ | $52.1_{\pm0.29}$ | $58.9_{\pm0.28}$ | $58.8_{\pm0.01}$ | $67.0_{\pm0.08}$ | $75.1_{\pm0.19}$ | $64.7_{\pm0.08}$ | $56.7_{\pm0.25}$ | $67.2_{\pm0.01}$ | $64.6_{\pm0.34}$ | $57.1_{\pm0.31}$ |
| + PTTA | $44.8_{\pm1.42}$ | $43.6_{\pm1.48}$ | $46.0_{\pm1.24}$ | $53.1_{\pm0.13}$ | $50.3_{\pm0.22}$ | $56.7_{\pm0.34}$ | $53.9_{\pm0.28}$ | $60.5_{\pm0.23}$ | $60.9_{\pm0.59}$ | $69.0_{\pm0.23}$ | $75.3_{\pm0.31}$ | $65.9_{\pm0.01}$ | $58.4_{\pm0.32}$ | $67.8_{\pm0.09}$ | $64.4_{\pm0.46}$ | $58.1_{\pm0.49}$ |
| SAR | $45.9_{\pm0.19}$ | $44.9_{\pm0.22}$ | $46.9_{\pm0.13}$ | $53.0_{\pm0.06}$ | $49.9_{\pm0.13}$ | $55.9_{\pm0.04}$ | $51.6_{\pm0.10}$ | $57.6_{\pm0.21}$ | $52.8_{\pm2.73}$ | $66.3_{\pm0.12}$ | $74.8_{\pm0.05}$ | $64.4_{\pm0.13}$ | $55.4_{\pm0.33}$ | $66.7_{\pm0.04}$ | $64.4_{\pm0.09}$ | $56.7_{\pm0.19}$ |
| + PTTA | $46.2_{\pm0.19}$ | $45.3_{\pm0.23}$ | $47.4_{\pm0.11}$ | $53.3_{\pm0.01}$ | $50.6_{\pm0.11}$ | $56.5_{\pm0.06}$ | $53.3_{\pm0.07}$ | $59.3_{\pm0.16}$ | $59.8_{\pm0.07}$ | $68.2_{\pm0.12}$ | $75.2_{\pm0.02}$ | $65.4_{\pm0.06}$ | $57.2_{\pm0.31}$ | $67.4_{\pm0.07}$ | $64.5_{\pm0.06}$ | $58.0_{\pm0.04}$ |
| ETA | $53.3_{\pm0.01}$ | $53.4_{\pm0.14}$ | $54.3_{\pm0.00}$ | $57.6_{\pm0.13}$ | $57.8_{\pm0.16}$ | $62.1_{\pm0.06}$ | $59.9_{\pm0.17}$ | $66.0_{\pm0.04}$ | $64.8_{\pm0.05}$ | $72.5_{\pm0.09}$ | $77.5_{\pm0.04}$ | $67.8_{\pm0.13}$ | $66.6_{\pm0.03}$ | $72.2_{\pm0.15}$ | $69.5_{\pm0.10}$ | $63.7_{\pm0.03}$ |
| + PTTA | $54.0_{\pm0.07}$ | $53.8_{\pm0.28}$ | $55.1_{\pm0.09}$ | $58.2_{\pm0.17}$ | $58.3_{\pm0.15}$ | $63.1_{\pm0.12}$ | $61.2_{\pm0.12}$ | $67.1_{\pm0.02}$ | $65.9_{\pm0.06}$ | $73.3_{\pm0.15}$ | $77.8_{\pm0.06}$ | $68.8_{\pm0.11}$ | $66.9_{\pm0.06}$ | $73.0_{\pm0.08}$ | $69.6_{\pm0.09}$ | $64.4_{\pm0.04}$ |
| EATA | $51.7_{\pm0.61}$ | $51.2_{\pm0.70}$ | $52.6_{\pm0.63}$ | $55.6_{\pm0.37}$ | $55.5_{\pm0.62}$ | $59.8_{\pm0.48}$ | $57.5_{\pm0.65}$ | $63.0_{\pm0.91}$ | $61.6_{\pm0.66}$ | $71.0_{\pm0.26}$ | $76.0_{\pm0.46}$ | $67.0_{\pm0.17}$ | $64.3_{\pm0.52}$ | $70.3_{\pm0.47}$ | $67.7_{\pm0.28}$ | $61.7_{\pm0.51}$ |
| + PTTA | $52.3_{\pm0.32}$ | $51.7_{\pm0.82}$ | $53.3_{\pm0.42}$ | $56.4_{\pm0.29}$ | $56.2_{\pm0.45}$ | $61.1_{\pm0.33}$ | $59.0_{\pm0.45}$ | $65.3_{\pm0.50}$ | $63.7_{\pm0.41}$ | $72.3_{\pm0.06}$ | $76.9_{\pm0.28}$ | $68.0_{\pm0.13}$ | $65.0_{\pm0.29}$ | $71.2_{\pm0.43}$ | $68.1_{\pm0.26}$ | $62.7_{\pm0.35}$ |
| DEYO | $54.4_{\pm0.15}$ | $54.7_{\pm0.18}$ | $55.4_{\pm0.24}$ | $58.2_{\pm0.08}$ | $58.4_{\pm0.05}$ | $63.1_{\pm0.06}$ | $54.6_{\pm6.94}$ | $66.9_{\pm0.09}$ | $65.8_{\pm0.03}$ | $73.3_{\pm0.06}$ | $78.2_{\pm0.01}$ | $68.0_{\pm0.03}$ | $67.8_{\pm0.01}$ | $73.2_{\pm0.04}$ | $70.4_{\pm0.02}$ | $64.2_{\pm0.46}$ |
| + PTTA | $55.0_{\pm0.03}$ | $55.3_{\pm0.23}$ | $56.1_{\pm0.13}$ | $58.8_{\pm0.05}$ | $59.0_{\pm0.10}$ | $64.1_{\pm0.07}$ | $62.3_{\pm0.07}$ | $68.1_{\pm0.07}$ | $66.6_{\pm0.06}$ | $74.0_{\pm0.02}$ | $78.5_{\pm0.09}$ | $69.1_{\pm0.22}$ | $68.1_{\pm0.10}$ | $73.9_{\pm0.13}$ | $70.4_{\pm0.06}$ | **$65.3_{\pm0.01}$** |
| CPL | $42.4_{\pm0.19}$ | $41.0_{\pm0.19}$ | $42.5_{\pm0.61}$ | $48.6_{\pm0.38}$ | $45.3_{\pm0.05}$ | $49.5_{\pm0.12}$ | $47.6_{\pm0.33}$ | $55.8_{\pm0.15}$ | $52.8_{\pm0.39}$ | $59.9_{\pm0.12}$ | $70.1_{\pm0.39}$ | $59.1_{\pm0.21}$ | $52.7_{\pm0.24}$ | $59.5_{\pm0.29}$ | $59.0_{\pm0.07}$ | $52.4_{\pm0.08}$ |
| + PTTA | $41.9_{\pm0.15}$ | $40.6_{\pm0.21}$ | $43.0_{\pm0.23}$ | $50.0_{\pm0.12}$ | $47.2_{\pm0.14}$ | $52.2_{\pm0.34}$ | $49.4_{\pm0.33}$ | $56.9_{\pm0.24}$ | $58.3_{\pm0.23}$ | $64.5_{\pm0.05}$ | $73.6_{\pm0.13}$ | $63.1_{\pm0.06}$ | $54.1_{\pm0.09}$ | $63.1_{\pm0.05}$ | $60.6_{\pm0.14}$ | $54.6_{\pm0.06}$ |

parameters for test-time adaptation, which achieves a 1.2% accuracy improvement for CLIP-RN50 and a 1.1% improvement for CLIP-ViT/B-16. The introduction of our proposed PTTA improves the accuracy by 1.8% and 1.4% respectively. This validates the performance of our method on natural distribution shifts. Meanwhile, we also carry out experiments on the single TTA task on ImageNet variants to further demonstrate the robustness of PTTA to natural distribution shifts, as shown in Table 3, achieving a maximum accuracy improvement of 2.0%. It is worth noting that we compare the effects of using ID retrieval and OOD retrieval for the episodic TTA task. The experimental results show that OOD retrieval achieves comparable or higher performance than ID retrieval, with an average improvement of 0.3%. This benefits from the fact that OOD retrieval takes into account the prediction bias of DNNs caused by data distribution deviation.

## D.4. Semantic Segmentation

Beyond image classification, we conduct experiments on semantic segmentation to verify the effectiveness of our proposed PTTA in practical applications, as shown in Table 5. We use the CarlaTTA dataset to conduct experiments in 5 continuously changing scenarios, including day2night, clear2fog, clear2rain, dynamic, and highway. This task simulates the environmental changes of autonomous vehicles driving in day and night, sunny, foggy, and rainy days, as well as on highways and in cities. MEMO based on episodic TTA and Tent based on continual TTA are used as baselines for comparison. We apply PTTA to these methods and achieve a maximum mIoU improvement of 1.8%. This validates the applicability of our method to different application tasks and its expandability to different TTA methods.

Table 15: Detailed experimental results (top-1 classification accuracy (%)) on single & continual TTA tasks using WRN-40. We highlight the highest accuracy in **bold** and the second best as underline.

| Method | Noise | | | Blur | | | | Weather | | | | Digital | | | | Avg. |
|---|---|---|---|---|---|---|---|---|---|---|---|---|---|---|---|---|
| | Gauss. | Shot | Impul. | Defoc. | Glass | Motion | Zoom | Snow | Frost | Fog | Brit. | Contr. | Elastic | Pixel | JPEG | |
| | | | | | | SINGLE | | | | | | | | | | |
| NoAdapt | $34.3_{\pm0.00}$ | $40.0_{\pm0.00}$ | $40.9_{\pm0.00}$ | $68.0_{\pm0.00}$ | $49.0_{\pm0.00}$ | $66.5_{\pm0.00}$ | $67.0_{\pm0.00}$ | $58.6_{\pm0.00}$ | $54.8_{\pm0.00}$ | $48.6_{\pm0.00}$ | $68.4_{\pm0.00}$ | $44.5_{\pm0.00}$ | $59.7_{\pm0.00}$ | $40.3_{\pm0.00}$ | $57.6_{\pm0.00}$ | $53.2_{\pm0.00}$ |
| CoTTA | $55.0_{\pm0.09}$ | $55.4_{\pm0.46}$ | $51.9_{\pm0.16}$ | $66.7_{\pm0.27}$ | $52.9_{\pm0.18}$ | $66.1_{\pm0.10}$ | $66.4_{\pm0.08}$ | $60.4_{\pm0.19}$ | $60.8_{\pm0.06}$ | $53.8_{\pm0.09}$ | $69.1_{\pm0.10}$ | $62.9_{\pm0.22}$ | $58.5_{\pm0.14}$ | $62.2_{\pm0.20}$ | $55.3_{\pm0.06}$ | $59.8_{\pm0.03}$ |
| SAR | $58.0_{\pm0.12}$ | $58.6_{\pm0.17}$ | $56.0_{\pm0.42}$ | $68.9_{\pm0.22}$ | $55.6_{\pm0.13}$ | $67.6_{\pm0.27}$ | $68.3_{\pm0.28}$ | $62.9_{\pm0.16}$ | $63.3_{\pm0.47}$ | $58.5_{\pm0.07}$ | $70.8_{\pm0.39}$ | $64.9_{\pm0.12}$ | $60.4_{\pm0.20}$ | $64.9_{\pm0.24}$ | $57.2_{\pm0.21}$ | $62.4_{\pm0.02}$ |
| Tent | $58.3_{\pm0.11}$ | $58.9_{\pm0.17}$ | $56.0_{\pm0.41}$ | $69.0_{\pm0.17}$ | $55.9_{\pm0.06}$ | $67.7_{\pm0.19}$ | $68.5_{\pm0.23}$ | $63.2_{\pm0.25}$ | $63.7_{\pm0.42}$ | $58.5_{\pm0.09}$ | $71.0_{\pm0.34}$ | $65.4_{\pm0.30}$ | $60.7_{\pm0.16}$ | $65.4_{\pm0.31}$ | $57.4_{\pm0.39}$ | $62.6_{\pm0.01}$ |
| + PTTA | $58.6_{\pm0.17}$ | $59.4_{\pm0.57}$ | $56.8_{\pm0.60}$ | $69.1_{\pm0.15}$ | $56.1_{\pm0.20}$ | $68.0_{\pm0.33}$ | $68.7_{\pm0.22}$ | $63.5_{\pm0.26}$ | $64.0_{\pm0.26}$ | $59.1_{\pm0.11}$ | $71.1_{\pm0.49}$ | $66.3_{\pm0.50}$ | $60.9_{\pm0.24}$ | $66.0_{\pm0.49}$ | $57.5_{\pm0.59}$ | $63.0_{\pm0.01}$ |
| ETA | $59.2_{\pm0.21}$ | $60.2_{\pm0.33}$ | $58.4_{\pm0.14}$ | $69.6_{\pm0.12}$ | $56.9_{\pm0.31}$ | $68.3_{\pm0.20}$ | $69.0_{\pm0.14}$ | $64.3_{\pm0.10}$ | $64.2_{\pm0.45}$ | $61.0_{\pm0.34}$ | $71.2_{\pm0.59}$ | $66.7_{\pm0.25}$ | $61.4_{\pm0.13}$ | $66.5_{\pm0.40}$ | $58.4_{\pm0.22}$ | $63.7_{\pm0.03}$ |
| + PTTA | $59.6_{\pm0.15}$ | $60.4_{\pm0.54}$ | $59.0_{\pm0.56}$ | $70.0_{\pm0.32}$ | $57.5_{\pm0.49}$ | $68.5_{\pm0.25}$ | $69.4_{\pm0.07}$ | $64.6_{\pm0.12}$ | $64.7_{\pm0.52}$ | $61.4_{\pm0.21}$ | $71.6_{\pm0.40}$ | $67.6_{\pm0.31}$ | $61.8_{\pm0.04}$ | $67.0_{\pm0.48}$ | $58.9_{\pm0.26}$ | $64.1_{\pm0.04}$ |
| EATA | $54.8_{\pm0.16}$ | $55.5_{\pm0.03}$ | $52.2_{\pm0.07}$ | $67.4_{\pm0.33}$ | $53.4_{\pm0.15}$ | $66.2_{\pm0.05}$ | $66.5_{\pm0.24}$ | $60.5_{\pm0.46}$ | $61.1_{\pm0.22}$ | $54.3_{\pm0.08}$ | $69.0_{\pm0.18}$ | $62.8_{\pm0.12}$ | $58.6_{\pm0.23}$ | $62.3_{\pm0.36}$ | $55.3_{\pm0.29}$ | $60.0_{\pm0.02}$ |
| + PTTA | $58.1_{\pm0.15}$ | $58.7_{\pm0.34}$ | $56.8_{\pm0.22}$ | $68.5_{\pm0.23}$ | $55.9_{\pm0.27}$ | $67.4_{\pm0.24}$ | $67.8_{\pm0.18}$ | $63.1_{\pm0.23}$ | $62.9_{\pm0.19}$ | $58.1_{\pm0.43}$ | $70.5_{\pm0.14}$ | $65.7_{\pm0.33}$ | $60.5_{\pm0.14}$ | $65.3_{\pm0.29}$ | $57.5_{\pm0.36}$ | $62.5_{\pm0.17}$ |
| DeYO | $59.7_{\pm0.19}$ | $60.7_{\pm0.14}$ | $61.1_{\pm0.13}$ | $69.6_{\pm0.19}$ | $57.5_{\pm0.25}$ | $68.6_{\pm0.12}$ | $69.4_{\pm0.19}$ | $65.4_{\pm0.26}$ | $64.5_{\pm0.53}$ | $62.8_{\pm0.33}$ | $71.5_{\pm0.24}$ | $68.8_{\pm0.37}$ | $61.3_{\pm0.25}$ | $67.2_{\pm0.06}$ | $59.3_{\pm0.12}$ | $\underline{64.5}_{\pm0.06}$ |
| + PTTA | $60.1_{\pm0.25}$ | $60.9_{\pm0.26}$ | $61.4_{\pm0.31}$ | $70.2_{\pm0.34}$ | $58.1_{\pm0.20}$ | $68.6_{\pm0.28}$ | $69.6_{\pm0.24}$ | $65.5_{\pm0.31}$ | $65.3_{\pm0.18}$ | $62.7_{\pm0.28}$ | $71.8_{\pm0.20}$ | $69.8_{\pm0.13}$ | $62.1_{\pm0.53}$ | $67.5_{\pm0.22}$ | $59.7_{\pm0.36}$ | $\mathbf{64.9}_{\pm0.06}$ |
| CPL | $55.8_{\pm0.37}$ | $56.8_{\pm0.24}$ | $53.3_{\pm0.14}$ | $67.4_{\pm0.38}$ | $54.5_{\pm0.32}$ | $66.4_{\pm0.26}$ | $66.5_{\pm0.08}$ | $61.4_{\pm0.37}$ | $61.6_{\pm0.16}$ | $55.4_{\pm0.27}$ | $69.6_{\pm0.20}$ | $63.9_{\pm0.14}$ | $58.9_{\pm0.30}$ | $62.9_{\pm0.31}$ | $56.0_{\pm0.49}$ | $60.7_{\pm0.02}$ |
| + PTTA | $56.5_{\pm0.22}$ | $57.3_{\pm0.13}$ | $54.1_{\pm0.11}$ | $67.7_{\pm0.32}$ | $54.9_{\pm0.43}$ | $66.8_{\pm0.37}$ | $66.8_{\pm0.10}$ | $61.7_{\pm0.38}$ | $61.9_{\pm0.10}$ | $56.0_{\pm0.42}$ | $69.8_{\pm0.22}$ | $64.5_{\pm0.13}$ | $59.3_{\pm0.09}$ | $63.5_{\pm0.41}$ | $56.4_{\pm0.40}$ | $61.1_{\pm0.01}$ |
| | | | | | | CONTINUAL | | | | | | | | | | |
| NoAdapt | $34.3_{\pm0.00}$ | $40.0_{\pm0.00}$ | $40.9_{\pm0.00}$ | $68.0_{\pm0.00}$ | $49.0_{\pm0.00}$ | $66.5_{\pm0.00}$ | $67.0_{\pm0.00}$ | $58.6_{\pm0.00}$ | $54.8_{\pm0.00}$ | $48.6_{\pm0.00}$ | $68.4_{\pm0.00}$ | $44.5_{\pm0.00}$ | $59.7_{\pm0.00}$ | $40.3_{\pm0.00}$ | $57.6_{\pm0.00}$ | $53.2_{\pm0.00}$ |
| CoTTA | $55.0_{\pm0.09}$ | $55.8_{\pm0.46}$ | $52.5_{\pm0.10}$ | $67.1_{\pm0.30}$ | $53.6_{\pm0.19}$ | $66.7_{\pm0.03}$ | $67.2_{\pm0.05}$ | $61.2_{\pm0.34}$ | $61.8_{\pm0.28}$ | $55.5_{\pm0.09}$ | $70.1_{\pm0.10}$ | $64.0_{\pm0.08}$ | $60.3_{\pm0.09}$ | $64.3_{\pm0.31}$ | $57.2_{\pm0.13}$ | $60.8_{\pm0.04}$ |
| SAR | $58.0_{\pm0.12}$ | $61.3_{\pm0.35}$ | $59.0_{\pm0.20}$ | $68.8_{\pm0.12}$ | $57.7_{\pm0.14}$ | $67.5_{\pm0.20}$ | $69.3_{\pm0.21}$ | $63.6_{\pm0.04}$ | $64.9_{\pm0.40}$ | $60.7_{\pm0.02}$ | $70.6_{\pm0.27}$ | $67.0_{\pm0.33}$ | $61.8_{\pm0.10}$ | $67.0_{\pm0.23}$ | $58.2_{\pm0.25}$ | $63.7_{\pm0.03}$ |
| Tent | $58.3_{\pm0.11}$ | $61.2_{\pm0.39}$ | $58.6_{\pm0.22}$ | $68.8_{\pm0.19}$ | $57.4_{\pm0.15}$ | $67.3_{\pm0.23}$ | $69.2_{\pm0.18}$ | $63.5_{\pm0.14}$ | $64.8_{\pm0.43}$ | $60.3_{\pm0.22}$ | $70.1_{\pm0.21}$ | $67.0_{\pm0.43}$ | $61.8_{\pm0.10}$ | $67.0_{\pm0.23}$ | $58.0_{\pm0.26}$ | $63.6_{\pm0.07}$ |
| + PTTA | $58.7_{\pm0.20}$ | $61.6_{\pm0.37}$ | $59.6_{\pm0.32}$ | $68.9_{\pm0.35}$ | $57.9_{\pm0.50}$ | $67.9_{\pm0.11}$ | $69.7_{\pm0.04}$ | $64.2_{\pm0.17}$ | $65.3_{\pm0.42}$ | $61.2_{\pm0.39}$ | $71.1_{\pm0.12}$ | $68.8_{\pm0.38}$ | $63.0_{\pm0.05}$ | $68.0_{\pm0.24}$ | $59.0_{\pm0.18}$ | $\mathbf{64.3}_{\pm0.03}$ |
| ETA | $59.2_{\pm0.21}$ | $62.2_{\pm0.38}$ | $59.2_{\pm0.22}$ | $67.8_{\pm0.22}$ | $56.9_{\pm0.43}$ | $66.4_{\pm0.32}$ | $68.5_{\pm0.35}$ | $63.6_{\pm0.46}$ | $64.0_{\pm0.40}$ | $60.6_{\pm0.15}$ | $68.3_{\pm0.15}$ | $66.4_{\pm0.44}$ | $60.1_{\pm0.15}$ | $65.0_{\pm0.24}$ | $56.4_{\pm0.54}$ | $63.0_{\pm0.14}$ |
| + PTTA | $59.6_{\pm0.15}$ | $62.6_{\pm0.43}$ | $60.0_{\pm0.21}$ | $68.5_{\pm0.19}$ | $58.2_{\pm0.11}$ | $67.3_{\pm0.19}$ | $69.7_{\pm0.06}$ | $64.4_{\pm0.28}$ | $65.2_{\pm0.39}$ | $62.3_{\pm0.21}$ | $69.8_{\pm0.28}$ | $69.1_{\pm0.11}$ | $62.2_{\pm0.15}$ | $66.8_{\pm0.08}$ | $58.3_{\pm0.20}$ | $\underline{64.3}_{\pm0.07}$ |
| EATA | $54.8_{\pm0.16}$ | $55.6_{\pm0.02}$ | $52.1_{\pm0.05}$ | $67.4_{\pm0.33}$ | $53.4_{\pm0.14}$ | $66.2_{\pm0.07}$ | $66.5_{\pm0.23}$ | $60.5_{\pm0.51}$ | $61.1_{\pm0.23}$ | $54.4_{\pm0.10}$ | $69.0_{\pm0.18}$ | $62.9_{\pm0.17}$ | $58.6_{\pm0.24}$ | $62.3_{\pm0.39}$ | $55.3_{\pm0.25}$ | $60.0_{\pm0.02}$ |
| + PTTA | $58.1_{\pm0.15}$ | $59.9_{\pm0.13}$ | $57.4_{\pm0.22}$ | $69.0_{\pm0.17}$ | $56.8_{\pm0.16}$ | $67.9_{\pm0.12}$ | $68.6_{\pm0.26}$ | $63.6_{\pm0.26}$ | $63.9_{\pm0.21}$ | $59.3_{\pm0.26}$ | $71.3_{\pm0.24}$ | $66.8_{\pm0.37}$ | $61.1_{\pm0.35}$ | $66.2_{\pm0.52}$ | $58.2_{\pm0.43}$ | $63.2_{\pm0.21}$ |
| DeYO | $59.7_{\pm0.19}$ | $60.9_{\pm0.24}$ | $58.1_{\pm0.53}$ | $65.0_{\pm0.60}$ | $53.6_{\pm0.65}$ | $63.0_{\pm1.04}$ | $64.9_{\pm0.74}$ | $59.8_{\pm0.57}$ | $59.6_{\pm1.05}$ | $56.7_{\pm0.90}$ | $63.0_{\pm1.31}$ | $62.8_{\pm1.67}$ | $53.9_{\pm1.60}$ | $57.9_{\pm1.78}$ | $48.0_{\pm1.77}$ | $59.1_{\pm0.83}$ |
| + PTTA | $60.1_{\pm0.16}$ | $61.5_{\pm0.52}$ | $59.4_{\pm0.60}$ | $66.3_{\pm0.36}$ | $56.0_{\pm0.51}$ | $64.6_{\pm0.18}$ | $66.8_{\pm1.04}$ | $62.3_{\pm0.56}$ | $62.8_{\pm0.41}$ | $60.8_{\pm0.94}$ | $66.9_{\pm1.35}$ | $66.6_{\pm1.38}$ | $58.5_{\pm1.57}$ | $63.3_{\pm2.29}$ | $54.8_{\pm1.59}$ | $62.0_{\pm0.73}$ |
| CPL | $55.8_{\pm0.37}$ | $58.3_{\pm0.36}$ | $56.1_{\pm0.23}$ | $68.2_{\pm0.16}$ | $56.2_{\pm0.31}$ | $67.8_{\pm0.35}$ | $68.5_{\pm0.21}$ | $63.4_{\pm0.19}$ | $64.3_{\pm0.38}$ | $57.0_{\pm0.19}$ | $70.7_{\pm0.25}$ | $66.0_{\pm0.08}$ | $61.1_{\pm0.17}$ | $65.8_{\pm0.41}$ | $57.8_{\pm0.23}$ | $62.5_{\pm0.13}$ |
| + PTTA | $56.5_{\pm0.30}$ | $59.7_{\pm0.39}$ | $57.6_{\pm0.35}$ | $68.5_{\pm0.21}$ | $57.2_{\pm0.19}$ | $67.3_{\pm0.31}$ | $68.3_{\pm0.16}$ | $62.7_{\pm0.15}$ | $63.6_{\pm0.37}$ | $58.3_{\pm0.28}$ | $70.1_{\pm0.13}$ | $66.7_{\pm0.36}$ | $60.9_{\pm0.18}$ | $65.8_{\pm0.15}$ | $56.8_{\pm0.68}$ | $62.7_{\pm0.11}$ |

## D.5. Adversarial Defense

Table 16 shows the error rates of samples under the Indiscriminate and Instant Attack Scenario, while Table 17 presents the error rates of samples without any attack. MedBN (Park et al., 2024) effectively defends against attacks, for example, on CIFAR100-C, the error rates for Tent, ETA, DeYO, and CPL decrease by 6.7%, 6.5%, 5.5%, and 7.7% under attack. However, MedBN hinders the model's adaptation process, leading to higher error rates for samples without attacks compared to using BN Adapt (Schneider et al., 2020). Specifically, on CIFAR100-C, for Tent, the average error rate of samples without attacks is 39.4% for BN Adapt, while it is 40.9% for MedBN. Incorporating PTTA can mitigate this issue. PTTA not only reduces the error rates of attacked samples but also those of samples without attacks. On ImageNet-C, PTTA also shows positive effects. For Tent, the error rate of attacked samples reduces from 69.4% to 65.4% with PTTA, and for non-attacked samples, it decreases from 60.8% to 55.9%. This indicates that PTTA can enhance the model's robustness against adversarial attacks while maintaining good performance on normal samples, thereby validating its effectiveness in the adversarial defense scenario.

# E. Additional Discussions

## E.1. Effect of Saliency Indicator

In Fig. 10, we visualize the saliency distance among test samples under 4 types of image corruptions, namely Gaussian noise (Gauss.), Defocus blur (Defoc.), Snow (Snow), and Contrast (Contr.). We sort the test samples in each type of image corruption in ascending order of the entropy predicted by DNNs and split them according to percentages. We calculate the average saliency distance between samples in each split and those in other splits, which is normalized to the range of $0 \sim 1$. According to the sample selection criterion based on entropy, low-entropy samples can be regarded as benign samples for purifying malicious high-entropy samples. Therefore, we hope that all malicious samples can retrieve corresponding benign samples rather than malicious ones in the memory bank or within the mini-batch. A good saliency indicator should make the saliency distance between malicious and benign samples greater than that between malicious samples and less than that

Table 16: Error Rate (%) of adversarial attacked samples under Indiscriminate and Instant Attack Scenario.

| | NOISE | | | BLUR | | | | WEATHER | | | | DIGITAL | | | | |
|---|---|---|---|---|---|---|---|---|---|---|---|---|---|---|---|---|
| METHOD | GAUSS. | SHOT | IMPUL. | DEFOC. | GLASS | MOTION | ZOOM | SNOW | FROST | FOG | BRIT. | CONTR. | ELASTIC | PIXEL | JPEG | AVG. ↓ |
| **CIFAR100-C** | | | | | | | | | | | | | | | | |
| **TENT** | | | | | | | | | | | | | | | | |
| • BN ADAPT | 60.7±0.06 | 59.3±0.66 | 67.4±0.75 | 45.1±0.17 | 59.8±0.11 | 48.5±0.48 | 43.1±0.04 | 51.9±0.09 | 53.2±0.31 | 55.9±0.83 | 44.8±0.62 | 50.7±0.12 | 56.0±0.21 | 46.4±0.04 | 55.9±0.02 | 53.3±0.16 |
| + PTTA | 58.1±0.40 | 57.0±0.17 | 65.5±0.02 | 44.5±0.06 | 58.3±0.43 | 48.2±2.02 | 42.0±0.34 | 50.0±0.23 | 51.0±0.10 | 55.3±0.20 | 43.3±0.52 | 51.5±0.82 | 55.0±0.74 | 44.6±0.27 | 55.5±0.31 | 52.0±0.18 |
| • MEDBN | 54.4±0.69 | 52.5±0.76 | 61.7±0.14 | 38.3±0.72 | 54.5±0.08 | 41.4±0.04 | 36.0±0.21 | 46.1±0.30 | 45.7±0.34 | 48.1±1.05 | 38.2±2.57 | 42.4±0.02 | 49.6±0.70 | 39.5±0.04 | 50.1±0.33 | 46.6±0.34 |
| + PTTA | 52.7±0.54 | 51.8±0.83 | 61.7±0.21 | 37.9±0.33 | 52.4±0.59 | 41.1±0.11 | 35.3±0.70 | 44.3±0.42 | 44.3±0.39 | 47.2±0.08 | 37.3±1.38 | 42.3±0.50 | 49.2±1.22 | 37.9±0.55 | 49.7±0.26 | **45.7±0.30** |
| **ETA** | | | | | | | | | | | | | | | | |
| • BN ADAPT | 59.4±0.28 | 58.3±1.08 | 66.7±1.41 | 44.5±0.19 | 58.7±0.28 | 47.9±0.57 | 42.5±0.34 | 50.8±0.19 | 52.6±0.34 | 54.8±0.71 | 44.1±0.08 | 49.9±0.36 | 55.5±0.42 | 45.6±0.07 | 55.4±0.10 | 52.4±0.24 |
| + PTTA | 57.4±0.22 | 56.8±0.76 | 65.2±0.73 | 44.4±0.00 | 57.1±0.47 | 47.5±0.33 | 41.6±0.10 | 49.7±0.08 | 50.5±0.54 | 54.5±0.14 | 42.7±0.41 | 51.3±0.54 | 54.4±0.64 | 44.2±0.28 | 54.5±0.12 | 51.5±0.23 |
| • MEDBN | 53.9±0.42 | 52.3±0.24 | 60.8±0.27 | 37.8±0.21 | 52.3±0.38 | 40.8±0.17 | 36.0±0.16 | 45.3±0.82 | 46.0±0.14 | 47.4±0.22 | 38.2±2.57 | 41.9±0.37 | 48.6±0.33 | 38.2±0.02 | 49.4±0.87 | 45.9±0.14 |
| + PTTA | 52.7±0.69 | 51.5±0.06 | 61.1±0.04 | 37.1±0.23 | 51.1±1.04 | 40.6±0.06 | 35.6±0.57 | 44.5±0.46 | 44.4±0.16 | 46.9±0.26 | 37.2±0.56 | 41.7±0.76 | 48.3±0.71 | 37.4±0.45 | 49.1±0.15 | **45.3±0.02** |
| **DEYO** | | | | | | | | | | | | | | | | |
| • BN ADAPT | 61.8±0.44 | 60.0±0.50 | 68.3±0.94 | 45.9±0.13 | 61.8±0.04 | 50.3±0.66 | 43.4±0.24 | 53.1±0.14 | 54.2±0.31 | 56.5±0.27 | 45.0±0.21 | 51.4±0.31 | 57.3±0.27 | 46.8±0.35 | 56.9±0.40 | 54.2±0.09 |
| + PTTA | 58.8±0.04 | 57.3±0.12 | 65.5±0.88 | 44.9±0.17 | 58.3±0.81 | 48.1±0.29 | 42.2±0.51 | 50.7±0.01 | 51.5±0.11 | 55.5±0.27 | 43.5±0.97 | 51.5±0.31 | 56.0±0.95 | 45.0±0.57 | 55.2±0.02 | 52.3±0.24 |
| • MEDBN | 56.1±0.15 | 54.4±0.49 | 64.6±1.31 | 40.9±0.15 | 56.7±0.56 | 45.0±0.02 | 38.3±0.36 | 47.5±0.48 | 48.1±0.05 | 49.6±0.27 | 39.5±0.36 | 44.6±0.05 | 52.0±0.57 | 40.7±0.11 | 51.9±0.06 | 48.7±0.01 |
| + PTTA | 54.0±0.35 | 53.1±0.13 | 62.3±0.38 | 38.8±0.71 | 53.8±0.34 | 41.7±0.60 | 37.1±0.34 | 44.9±0.01 | 45.7±0.64 | 47.8±0.27 | 37.8±0.74 | 42.4±0.02 | 50.2±1.01 | 38.8±0.45 | 50.2±0.27 | 46.6±0.10 |
| **CPL** | | | | | | | | | | | | | | | | |
| • BN ADAPT | 61.4±0.41 | 60.6±0.10 | 68.7±0.43 | 46.4±0.30 | 60.8±0.34 | 51.0±0.48 | 43.7±0.22 | 53.7±0.79 | 53.9±0.28 | 61.5±0.46 | 45.6±0.21 | 54.3±0.15 | 56.7±0.24 | 46.9±0.57 | 57.9±0.48 | 54.9±0.02 |
| + PTTA | 60.0±0.40 | 59.5±0.04 | 67.0±0.00 | 47.0±0.50 | 59.2±0.13 | 51.4±0.31 | 44.2±0.15 | 53.0±0.88 | 53.0±0.10 | 61.9±0.12 | 46.3±0.08 | 56.1±0.42 | 56.2±0.09 | 46.8±0.71 | 57.7±0.04 | 54.6±0.12 |
| • MEDBN | 54.0±0.57 | 53.9±0.29 | 62.0±0.17 | 39.4±0.01 | 52.6±0.27 | 42.9±0.50 | 36.4±0.12 | 46.1±0.88 | 45.4±0.52 | 51.8±0.19 | 38.7±0.47 | 44.5±0.01 | 48.8±0.38 | 39.3±0.89 | 51.7±0.77 | 47.2±0.03 |
| + PTTA | 53.7±0.47 | 53.4±0.57 | 61.2±0.61 | 39.9±0.83 | 52.4±0.10 | 42.9±0.65 | 37.4±0.55 | 45.4±0.06 | 44.9±0.11 | 52.7±0.64 | 38.9±0.73 | 44.7±0.41 | 48.5±0.07 | 39.5±0.58 | 51.1±0.71 | 47.1±0.21 |
| **IMAGENET-C** | | | | | | | | | | | | | | | | |
| **TENT** | | | | | | | | | | | | | | | | |
| • BN ADAPT | 84.4±0.49 | 80.4±0.06 | 81.8±0.15 | 87.4±0.33 | 88.4±0.44 | 69.2±1.03 | 61.0±0.35 | 60.8±0.19 | 76.1±0.44 | 55.2±0.11 | 41.7±0.10 | 95.0±0.20 | 53.5±0.33 | 50.5±0.27 | 55.1±0.16 | 69.4±0.11 |
| + PTTA | 78.8±1.64 | 76.8±1.44 | 74.5±0.36 | 83.6±0.90 | 84.0±0.70 | 65.0±0.20 | 59.3±0.37 | 58.5±0.22 | 71.4±1.26 | 54.4±0.20 | 41.9±0.17 | 75.6±5.70 | 53.0±0.08 | 50.1±0.20 | 54.4±0.28 | 65.4±0.79 |
| • MEDBN | 85.7±0.81 | 84.0±0.24 | 83.6±0.13 | 85.5±0.30 | 86.4±1.02 | 72.5±1.30 | 58.0±0.28 | 61.6±0.80 | 70.7±0.03 | 48.5±0.24 | 35.1±0.05 | 88.8±0.45 | 53.9±0.33 | 47.0±0.04 | 55.0±0.41 | 67.8±0.17 |
| + PTTA | 85.4±0.00 | 85.8±0.00 | 82.0±0.00 | 81.0±0.00 | 84.2±0.00 | 69.0±0.00 | 57.3±0.00 | 60.6±0.00 | 65.3±0.00 | 48.8±0.00 | 36.2±0.00 | 76.2±0.00 | 53.7±0.00 | 48.0±0.00 | 56.4±0.00 | 66.0±0.00 |
| **ETA** | | | | | | | | | | | | | | | | |
| • BN ADAPT | 76.5±0.11 | 72.1±0.48 | 75.3±0.36 | 78.6±0.02 | 78.5±0.24 | 65.6±0.14 | 58.6±0.15 | 57.2±0.06 | 64.4±0.03 | 53.9±0.05 | 40.5±0.07 | 72.2±0.15 | 51.8±0.19 | 49.0±0.15 | 53.5±0.18 | 63.2±0.02 |
| + PTTA | 75.1±0.15 | 71.3±0.38 | 73.9±0.30 | 77.4±0.19 | 77.4±0.03 | 64.1±0.27 | 57.5±0.23 | 56.5±0.17 | 63.3±0.23 | 53.2±0.03 | 40.2±0.01 | 69.5±0.43 | 51.4±0.22 | 48.5±0.15 | 52.7±0.31 | 62.1±0.03 |
| • MEDBN | 82.0±0.02 | 80.5±0.09 | 80.7±0.14 | 80.9±0.39 | 81.4±0.01 | 69.4±0.31 | 56.9±0.11 | 59.9±0.36 | 62.8±0.15 | 47.9±0.16 | 34.9±0.15 | 76.3±0.27 | 52.2±0.30 | 47.0±0.48 | 54.9±0.38 | 64.5±0.12 |
| + PTTA | 83.3±0.00 | 82.6±0.00 | 82.8±0.00 | 81.6±0.00 | 82.3±0.00 | 68.7±0.00 | 56.7±0.00 | 58.6±0.00 | 62.9±0.00 | 48.4±0.00 | 35.5±0.00 | 75.3±0.00 | 53.7±0.00 | 47.7±0.00 | 56.0±0.00 | 65.1±0.00 |
| **DEYO** | | | | | | | | | | | | | | | | |
| • BN ADAPT | 74.3±0.11 | 70.7±0.48 | 73.4±0.25 | 77.8±0.08 | 77.2±0.06 | 64.6±0.20 | 59.1±0.10 | 56.6±0.03 | 64.3±0.03 | 54.1±0.11 | 41.2±0.00 | 71.6±0.51 | 51.8±0.05 | 49.3±0.29 | 53.2±0.17 | 62.6±0.03 |
| + PTTA | 73.8±0.72 | 70.1±0.47 | 73.0±0.47 | 76.9±0.96 | 76.9±0.39 | 63.8±0.72 | 58.5±0.71 | 56.3±0.46 | 63.8±0.72 | 53.7±0.48 | 41.0±0.35 | 70.1±1.29 | 51.5±0.19 | 48.9±0.41 | 52.9±0.15 | **62.1±0.57** |
| • MEDBN | 83.0±1.27 | 82.1±0.28 | 81.7±0.49 | 82.2±0.38 | 82.8±0.13 | 70.0±0.53 | 57.3±0.24 | 60.4±0.29 | 63.7±0.26 | 48.2±0.46 | 35.2±0.24 | 78.0±0.74 | 52.3±0.22 | 47.6±0.42 | 55.0±0.19 | 65.3±0.11 |
| + PTTA | 84.0±0.00 | 83.1±0.00 | 82.6±0.00 | 81.4±0.00 | 82.2±0.00 | 69.9±0.00 | 57.4±0.00 | 59.8±0.00 | 64.3±0.00 | 49.3±0.00 | 35.7±0.00 | 76.9±0.00 | 53.5±0.00 | 47.7±0.00 | 56.1±0.00 | 65.6±0.00 |
| **CPL** | | | | | | | | | | | | | | | | |
| • BN ADAPT | 78.6±0.11 | 73.8±0.11 | 76.9±0.65 | 81.0±0.15 | 81.2±0.69 | 68.9±0.09 | 61.2±0.00 | 60.8±0.20 | 67.6±0.07 | 57.4±0.14 | 42.0±0.03 | 79.6±0.80 | 54.4±0.16 | 51.9±0.27 | 56.3±0.08 | 66.1±0.07 |
| + PTTA | 75.9±0.09 | 75.4±0.34 | 74.3±0.18 | 78.7±0.30 | 79.1±0.37 | 66.0±0.24 | 59.3±0.25 | 59.0±0.11 | 65.6±0.04 | 55.9±0.10 | 42.2±0.05 | 70.9±0.14 | 53.8±0.07 | 50.8±0.26 | 55.6±0.02 | 64.2±0.01 |
| • MEDBN | 80.7±0.12 | 79.9±0.21 | 78.8±0.25 | 80.8±0.56 | 81.4±0.32 | 69.9±0.11 | 57.2±0.05 | 61.4±0.08 | 63.6±0.24 | 49.8±0.31 | 36.3±0.23 | 78.0±1.86 | 52.6±0.34 | 48.7±0.59 | 55.2±0.17 | 64.9±0.18 |
| + PTTA | 82.5±0.00 | 80.8±0.00 | 80.8±0.00 | 80.9±0.00 | 81.8±0.00 | 70.7±0.00 | 58.3±0.00 | 61.8±0.00 | 64.4±0.00 | 50.7±0.00 | 37.8±0.00 | 76.9±0.00 | 54.0±0.00 | 49.6±0.00 | 56.5±0.00 | 65.8±0.00 |

between benign samples. This can prevent malicious samples from retrieving malicious samples and benign samples from retrieving malicious samples for Mixup. A typical example is shown in Fig. 10 (d). We compare the logit-, feature-, and pixel-saliency mentioned in Sec. 3.2, as shown in Fig. 10. In general, the logit-saliency indicator has characteristics that better meet the above requirements compared to the other two and is applicable to different DNNs, including ResNet50 and ViT-B/16. However, it is not very stable in representing different distribution shifts. The feature-saliency indicator is slightly weaker than the logit one but can still meet the above requirements. The pixel-saliency indicator fails to accurately represent the above requirements. Moreover, the effect of pixel-saliency distance on ResNet50 cannot meet the basic requirement of retrieving benign samples for malicious samples, while ViT-B/16 can meet this basic requirement. Overall, the logit-saliency indicator has a better effect in encoding benign and malicious samples, incurs lower computational overhead, and is more suitable for purifying malicious samples.

### E.2. ID, OOD, and IB Retrieval

In-Distribution (ID) retrieval uses benign samples sampled from the source training distribution as candidate samples. Out-Of-Distribution (OOD) retrieval uses samples from the test data distribution sampled during the test-time adaptation process as candidate samples, which are updated in real-time. In-Batch (IB) Retrieval, without building an additional memory bank, uses test samples in the mini-batch at time step $t$ as benign candidate samples. For TTA tasks that continuously update the parameters of DNNs, including single, continual, and lifelong tasks, OOD Retrieval utilizes test samples in the recent historical period as candidates and maintains consistency with the model updates. However, for episodic tasks, where the

Table 17: Error Rate (%) of samples without being attacked.

| METHOD | NOISE | | | BLUR | | | | WEATHER | | | | DIGITAL | | | | |
|---|---|---|---|---|---|---|---|---|---|---|---|---|---|---|---|---|
| | GAUSS. | SHOT | IMPUL. | DEFOC. | GLASS | MOTION | ZOOM | SNOW | FROST | FOG | BRIT. | CONTR. | ELASTIC | PIXEL | JPEG | AVG. ↓ |
| **CIFAR100-C** | | | | | | | | | | | | | | | | |
| **TENT** | | | | | | | | | | | | | | | | |
| • BN ADAPT | 46.8±0.36 | 45.9±0.16 | 54.9±0.44 | 31.4±0.54 | 46.5±0.10 | 34.7±0.09 | 29.5±0.20 | 39.2±0.34 | 38.5±0.21 | 40.2±0.93 | 31.9±0.23 | 32.6±0.12 | 42.1±0.52 | 33.3±0.38 | 44.1±0.20 | 39.4±0.21 |
| + PTTA | 45.2±0.29 | 44.4±0.27 | 53.8±0.35 | 30.1±0.45 | 45.5±0.24 | 33.6±0.19 | 28.6±0.26 | 37.5±0.24 | 37.0±0.00 | 38.8±0.22 | 30.9±0.61 | 31.4±0.50 | 41.0±0.62 | 31.6±0.11 | 43.2±0.57 | _38.2±0.16_ |
| • MEDBN | 49.7±0.33 | 47.9±0.57 | 57.5±0.00 | 32.0±0.93 | 48.7±0.11 | 35.2±0.05 | 30.5±0.05 | 40.7±0.14 | 39.8±0.09 | 41.8±1.59 | 33.0±0.04 | 33.7±0.19 | 43.5±0.13 | 33.8±0.07 | 45.9±0.62 | 40.9±0.30 |
| + PTTA | 47.5±1.43 | 46.5±1.05 | 56.7±0.13 | 31.0±0.65 | 46.9±0.35 | 34.3±0.07 | 29.4±0.55 | 39.3±0.09 | 38.1±0.12 | 40.2±0.01 | 31.5±0.35 | 33.0±0.89 | 42.8±0.81 | 32.4±0.33 | 44.5±0.51 | 39.6±0.12 |
| **ETA** | | | | | | | | | | | | | | | | |
| • BN ADAPT | 45.6±0.18 | 44.7±0.54 | 54.1±0.99 | 31.5±0.31 | 45.3±0.38 | 34.0±0.04 | 29.5±0.13 | 38.2±0.24 | 37.8±0.20 | 39.0±0.63 | 31.6±0.08 | 31.8±0.16 | 41.5±0.75 | 32.6±0.20 | 43.3±0.18 | 38.7±0.26 |
| + PTTA | 44.6±0.22 | 44.7±0.13 | 53.6±0.47 | 29.8±0.36 | 43.9±0.01 | 33.3±0.22 | 28.5±0.50 | 37.0±0.47 | 36.1±0.43 | 37.7±0.25 | 30.4±0.39 | 31.3±0.01 | 40.4±1.20 | 31.9±0.14 | 42.3±0.04 | **37.7±0.27** |
| • MEDBN | 48.2±0.16 | 47.4±0.53 | 56.1±1.08 | 32.2±0.33 | 47.0±0.21 | 34.9±0.09 | 30.2±0.23 | 39.9±0.54 | 39.7±0.27 | 40.8±0.08 | 32.5±0.30 | 33.5±0.59 | 42.6±0.04 | 33.4±0.25 | 45.4±0.68 | 40.2±0.18 |
| + PTTA | 46.6±0.13 | 45.9±0.18 | 56.0±0.31 | 30.4±0.49 | 45.6±0.11 | 33.9±0.08 | 29.2±0.14 | 38.9±0.19 | 37.9±0.28 | 39.3±0.10 | 31.5±0.16 | 32.3±0.04 | 41.9±1.02 | 31.7±0.18 | 44.2±0.41 | 39.0±0.06 |
| **DEYO** | | | | | | | | | | | | | | | | |
| • BN ADAPT | 48.9±0.06 | 47.4±0.35 | 56.3±0.57 | 34.1±0.21 | 49.3±0.07 | 37.9±0.47 | 31.9±0.19 | 40.6±0.13 | 40.0±0.03 | 42.2±0.06 | 33.3±0.05 | 33.6±0.43 | 45.2±0.02 | 34.6±0.30 | 46.4±0.19 | 41.4±0.03 |
| + PTTA | 46.3±0.98 | 45.0±0.05 | 54.2±0.23 | 31.1±0.52 | 45.6±0.29 | 34.0±0.19 | 29.7±0.17 | 37.8±0.30 | 37.1±0.02 | 39.0±0.64 | 31.1±0.35 | 31.1±0.11 | 42.1±0.45 | 32.5±0.07 | 43.3±0.42 | 38.7±0.15 |
| • MEDBN | 51.1±0.25 | 49.3±0.33 | 60.6±1.52 | 35.3±0.68 | 51.0±0.18 | 39.3±0.28 | 33.4±0.48 | 42.4±0.23 | 41.8±0.16 | 43.6±0.22 | 34.7±0.13 | 35.2±0.62 | 46.9±0.08 | 35.8±0.03 | 47.9±0.32 | 43.2±0.11 |
| + PTTA | 48.5±0.71 | 47.5±0.13 | 57.2±0.82 | 31.5±0.13 | 47.5±0.24 | 34.8±0.95 | 30.8±0.59 | 39.1±0.24 | 39.1±0.23 | 40.5±0.05 | 32.0±0.34 | 32.4±0.20 | 43.9±0.23 | 33.5±0.83 | 45.0±0.73 | 40.2±0.10 |
| **CPL** | | | | | | | | | | | | | | | | |
| • BN ADAPT | 45.7±0.17 | 45.5±0.08 | 55.1±0.01 | 31.7±0.08 | 45.3±0.27 | 35.7±0.24 | 30.0±0.21 | 40.0±0.93 | 38.5±0.35 | 43.6±0.11 | 32.2±0.36 | 34.6±0.24 | 42.1±0.22 | 33.5±0.45 | 45.2±0.37 | 39.9±0.04 |
| + PTTA | 45.7±0.25 | 45.8±0.03 | 54.5±0.36 | 31.5±0.25 | 44.6±0.13 | 35.7±0.22 | 29.9±0.01 | 40.1±0.83 | 38.6±0.30 | 44.0±0.41 | 32.5±0.01 | 34.7±0.01 | 41.6±0.02 | 33.2±0.27 | 45.3±0.41 | 39.8±0.09 |
| • MEDBN | 51.2±4.59 | 51.0±4.35 | 59.2±3.82 | 35.6±5.40 | 49.5±4.67 | 39.8±4.95 | 33.3±4.15 | 43.4±2.93 | 42.4±3.76 | 48.2±4.92 | 35.8±4.50 | 39.8±6.54 | 45.9±4.46 | 36.3±3.28 | 49.3±4.09 | 44.0±4.43 |
| + PTTA | 47.1±0.54 | 47.2±0.07 | 55.8±0.07 | 32.0±0.56 | 45.8±0.18 | 36.0±0.42 | 30.2±0.49 | 40.6±0.15 | 39.5±0.25 | 44.7±0.62 | 32.9±0.21 | 35.1±0.65 | 42.3±0.24 | 33.6±0.28 | 46.2±0.54 | 40.6±0.19 |
| **IMAGENET-C** | | | | | | | | | | | | | | | | |
| **TENT** | | | | | | | | | | | | | | | | |
| • BN ADAPT | 77.4±0.64 | 74.6±0.10 | 74.5±0.24 | 80.1±0.22 | 81.7±0.64 | 58.4±0.87 | 49.9±0.40 | 52.6±0.45 | 67.3±0.21 | 41.5±0.18 | 33.0±0.04 | 89.6±0.16 | 43.8±0.10 | 40.8±0.25 | 46.8±0.05 | 60.8±0.12 |
| + PTTA | 71.0±1.51 | 73.2±1.28 | 66.3±0.28 | 75.1±1.05 | 76.2±1.04 | 54.0±0.11 | 47.8±0.36 | 49.8±0.04 | 61.7±1.51 | 40.3±0.15 | 32.4±0.06 | 62.6±6.23 | 42.9±0.15 | 40.0±0.05 | 45.4±0.01 | 55.9±0.80 |
| • MEDBN | 84.9±0.59 | 83.8±0.08 | 83.4±0.59 | 85.0±0.30 | 85.8±1.16 | 71.4±1.38 | 56.6±0.17 | 61.1±0.81 | 70.2±0.04 | 47.1±0.32 | 34.3±0.05 | 88.9±0.58 | 52.4±0.27 | 46.2±0.17 | 53.8±0.13 | 67.0±0.15 |
| + PTTA | 84.3±0.00 | 85.7±0.00 | 81.5±0.00 | 80.3±0.00 | 83.3±0.00 | 67.3±0.00 | 55.7±0.00 | 59.7±0.00 | 64.2±0.00 | 47.3±0.00 | 35.0±0.00 | 76.2±0.00 | 51.7±0.00 | 46.8±0.00 | 54.8±0.00 | 64.9±0.00 |
| **ETA** | | | | | | | | | | | | | | | | |
| • BN ADAPT | 66.6±0.21 | 64.2±0.27 | 65.1±0.11 | 68.5±0.10 | 68.8±0.24 | 53.9±0.17 | 47.2±0.16 | 48.5±0.18 | 54.4±0.03 | 39.7±0.04 | 32.1±0.09 | 57.8±0.25 | 41.8±0.05 | 39.2±0.16 | 44.8±0.01 | 52.8±0.09 |
| + PTTA | 65.4±0.11 | 63.1±0.10 | 63.8±0.24 | 67.1±0.18 | 67.5±0.00 | 52.6±0.21 | 46.1±0.21 | 47.6±0.01 | 53.1±1.24 | 39.0±0.08 | 31.5±0.13 | 55.0±0.01 | 41.4±0.03 | 38.6±0.00 | 44.1±0.04 | _51.7±0.03_ |
| • MEDBN | 81.0±0.30 | 80.6±0.05 | 80.4±0.66 | 80.7±0.10 | 81.3±0.08 | 68.6±0.60 | 55.6±0.05 | 59.7±0.19 | 62.5±0.16 | 47.1±0.11 | 34.1±0.15 | 76.7±0.33 | 50.7±0.26 | 46.1±0.35 | 53.7±0.22 | 63.9±0.18 |
| + PTTA | 82.3±0.00 | 82.2±0.00 | 82.2±0.00 | 81.5±0.00 | 81.7±0.00 | 67.2±0.00 | 55.0±0.00 | 58.6±0.00 | 62.3±0.00 | 46.9±0.00 | 34.5±0.00 | 76.5±0.00 | 51.8±0.00 | 46.3±0.00 | 54.5±0.00 | 64.2±0.00 |
| **DEYO** | | | | | | | | | | | | | | | | |
| • BN ADAPT | 65.0±0.09 | 63.1±0.53 | 63.4±0.16 | 67.3±0.05 | 67.5±0.21 | 52.3±0.09 | 46.9±0.05 | 47.5±0.12 | 53.6±0.09 | 39.1±0.11 | 31.8±0.02 | 56.7±0.88 | 41.1±0.06 | 38.5±0.04 | 44.0±0.06 | 51.9±0.03 |
| + PTTA | 64.5±1.15 | 62.3±0.74 | 63.2±0.77 | 66.4±1.14 | 66.8±0.54 | 51.4±0.83 | 46.3±0.63 | 47.0±0.44 | 53.1±0.93 | 38.9±0.59 | 31.6±0.36 | 54.8±1.61 | 40.8±0.12 | 38.3±0.56 | 43.8±0.26 | **51.3±0.71** |
| • MEDBN | 82.4±1.32 | 82.5±0.40 | 81.5±0.21 | 81.6±0.03 | 82.6±0.03 | 68.8±0.61 | 56.0±0.23 | 60.0±0.19 | 63.2±0.28 | 47.2±0.39 | 34.2±0.04 | 78.7±0.49 | 50.7±0.11 | 46.5±0.16 | 53.8±0.13 | 64.6±0.05 |
| + PTTA | 82.9±0.00 | 82.7±0.00 | 81.8±0.00 | 80.9±0.00 | 81.7±0.00 | 68.5±0.00 | 55.6±0.00 | 59.4±0.00 | 63.2±0.00 | 47.2±0.00 | 34.5±0.00 | 77.4±0.00 | 51.5±0.00 | 46.0±0.00 | 54.4±0.00 | 64.5±0.00 |
| **CPL** | | | | | | | | | | | | | | | | |
| • BN ADAPT | 69.2±0.34 | 66.1±0.17 | 67.6±0.30 | 71.6±0.25 | 72.4±0.87 | 57.2±0.14 | 49.1±0.16 | 51.6±0.08 | 56.8±0.01 | 41.9±0.06 | 32.7±0.08 | 66.5±0.99 | 43.3±0.15 | 41.2±0.04 | 46.8±0.07 | 55.6±0.08 |
| + PTTA | 66.8±0.49 | 71.7±0.36 | 65.2±0.09 | 68.9±0.21 | 70.1±0.53 | 54.4±0.16 | 47.4±0.01 | 49.8±0.01 | 54.9±0.25 | 40.7±0.04 | 32.7±0.07 | 56.9±0.23 | 42.9±0.21 | 40.1±0.02 | 46.2±0.01 | 53.9±0.05 |
| • MEDBN | 80.0±0.13 | 79.6±0.56 | 78.8±0.11 | 80.4±0.48 | 81.1±0.21 | 68.9±0.05 | 56.1±0.11 | 61.1±0.16 | 63.4±0.22 | 48.9±0.28 | 35.5±0.12 | 78.7±1.56 | 51.1±0.28 | 47.8±0.46 | 54.3±0.11 | 64.4±0.10 |
| + PTTA | 81.2±0.00 | 80.2±0.00 | 80.0±0.00 | 80.1±0.00 | 81.2±0.00 | 69.0±0.00 | 56.4±0.00 | 61.2±0.00 | 63.5±0.00 | 48.9±0.00 | 36.5±0.00 | 76.8±0.00 | 51.8±0.00 | 48.5±0.00 | 55.0±0.00 | 64.7±0.00 |

model's parameters are reset after making predictions on the mini-batch at time step $t$, ID Retrieval better represents the data distribution related to DNNs' parameters, because DNNs generally have higher confidence in ID samples. The experimental results in Table 4 and Table 19 show that for episodic tasks, ID Retrieval performs better, while for the other three TTA tasks, OOD Retrieval has better performance. Meanwhile, due to the advantage of not requiring a memory bank, IB Retrieval reduces the storage and retrieval computational overhead.

### E.3. Necessity of Benign Sample Retrieval

Different from vanilla Mixup (Zhang et al., 2018), we employ a saliency indicator to retrieve benign samples that exert completely opposite influences on the objective function relative to individual malicious samples, forming the matched pairs for Mixup. Considering the use of Mixup technique (Zhang et al., 2018) to purify malicious samples, we compare with the vanilla Mixup (Zhang et al., 2018) to demonstrate the necessity of benign sample retrieval, which stores historical test samples in a memory bank (maximum length $1,000$) and randomly selects samples from it to mixup with the samples in current mini-batch at time step $t$. Results in Table 18 show that while vanilla Mixup can improve the performance of base TTA algorithms in some cases, our proposed PTTA demonstrates superior performance and robustness.

### E.4. Unnecessary of Sample Selection Criteria

We claim that benign samples selected based on the sample selection criteria should be used as candidates for purifying malicious samples. The reason is that benign samples cause less disruption to DNNs during test-time adaptation compared

Table 18: The comparison of PTTA with the vanilla Mixup method. Also the comparison of different purification scopes in PTTA, which purify all test samples by default.

| METHODS | SINGLE | | | | CONTINUAL | | | |
|---|---|---|---|---|---|---|---|---|
| | ETA | DEYO | CPL | AVG. | ETA | DEYO | CPL | AVG. |
| BASE ALGORITHM | 48.2 | 48.6 | 42.8 | 46.5 | 62.2 | 62.2 | 55.5 | 60.0 |
| + VANILLA MIXUP (ZHANG ET AL., 2018) (W/ $1K$ MEMORY BANK) | 48.6 | 49.1 | 38.2 | 45.3 | 64.7 | 64.5 | 55.9 | 61.7 |
| + PURIFY ONLY BENIGN SAMPLES (W/ OOD RETRIEVAL) | 48.9 | 49.6 | 44.4 | 47.6 | 65.2 | 65.3 | 59.5 | 63.3 |
| + PURIFY ONLY MALICIOUS SAMPLES (W/ OOD RETRIEVAL) | 49.1 | 49.6 | 44.3 | 47.7 | 64.7 | 65.3 | 59.0 | 63.0 |
| + PTTA (W/ OOD RETRIEVAL) | **49.5** | **49.9** | **44.6** | **48.0** | **65.3** | **65.8** | **59.6** | **63.5** |

to malicious samples, thus stabilizing the model's optimization and self-training processes. Existing sample selection criteria are highly dependent on the choice of thresholds. The threshold determines which samples are benign and which are malicious. However, different DNNs often require different thresholds for sample selection when facing different distribution shifts. Therefore, the sensitivity to the threshold determines the practicality and portability of existing TTA methods. To avoid the impact of sample selection thresholds on PTTA, we conduct experiments using all test samples as candidates for purifying malicious samples. That is, instead of using sample selection to distinguish between benign and malicious samples, all test samples are stored in the memory bank or within the mini-batch. Ideally, as stated in Appendix E.1, an excellent saliency indicator can retrieve benign samples that effectively purify malicious samples from a candidate pool mixed with benign and malicious samples, while ensuring that benign samples are not disrupted by malicious samples. Consequently, the experimental results in Fig. 5 verify the performance of Tent+PTTA with or without sample selection. With the logit-saliency indicator, our PTTA can effectively purify malicious samples without the assistance of sample selection. This further validates the effectiveness of the saliency indicator we proposed in Sec. 3.2.

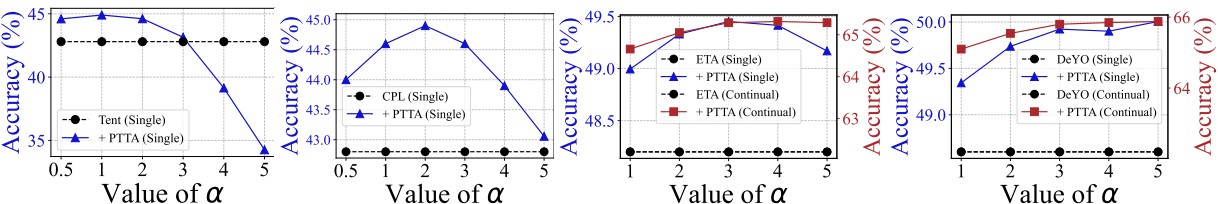

Figure 11: Ablation study of $\alpha$ in Eq. 8. We compare the performance of PTTA when applied to Tent, ETA, DeYO, and CPL for Single & Continual TTA tasks.

### E.5. Ablation Study of $\alpha$ in Eq. 8

The hyperparameter $\alpha$ in Eq. 8 is used to balance the loss function of the original TTA method $\mathcal{L}_{\text{tta}}$ and the purification loss we proposed $\mathcal{L}_{\text{pur}}$. In the main experiments, we set $\alpha = 3.0$ by default for sample-selection-based TTA methods and $\alpha = 1.0$ for selection-free methods. Since sample-selection-based methods, such as ETA and DeYO, assign larger weights to high-confident samples with low entropy, introducing $\alpha$ can balance the effects of these two losses. In Fig. 11, we compare the impacts of different values of $\alpha$ on the application of PTTA to Tent, ETA, DeYO, and CPL. The experimental results confirm that our default settings are appropriate.

### E.6. Different Size of Memory Bank

In the pipeline we proposed, a memory bank is designed to store selected benign samples. Subsequently, PTTA retrieves benign samples from the memory bank that match the malicious test samples for Mixup in the purification process. The memory bank is defined as a first-in-first-out (FIFO) queue structure with a limited length. The size of the memory bank directly affects the scope of benign sample retrieval, thus influencing the purification effect of PTTA. Meanwhile, it also determines the amount of storage space needed to store benign samples, their predicted pseudo-labels, and the calculated saliency indicator. However, we also propose an alternative without using the memory bank, where the retrieval is directly conducted within the test samples of the mini-batch arriving at the $t$-th time step. This In-Batch retrieval method eliminates

Table 19: We demonstrate the generality of PTTA across various sample selection criteria. Meanwhile, we compare the performance of PTTA with different saliency indicators, types of retrieval, and sample purification scopes. We count and compare the number of forward (#Fwds) and backward (#Bwds) propagations for different TTA methods.

| Methods | Single | | | | | Continual | | | | | #Fwds | #Bwds |
| | Tent | ETA | DeYO | CPL | Avg. | Tent | ETA | DeYO | CPL | Avg. | | |
|---|---|---|---|---|---|---|---|---|---|---|---|---|
| COMPARISON WITH EXISTING PURIFICATION (PURIF.) METHODS. | | | | | | | | | | | | |
| Base Algorithm | 42.8±0.04 | 48.2±0.04 | 48.6±0.11 | 42.8±0.04 | 45.6±0.06 | 14.2±10.5 | 62.2±0.07 | 62.2±0.35 | 55.5±0.13 | 48.5±2.77 | $N_{batch}$ | $N_{batch}$ |
| + Purif. w/ FGSM | 42.0±0.17 | 45.9±0.04 | 47.0±0.04 | 42.1±0.09 | 44.2±0.09 | 2.7±0.69 | 61.7±0.03 | 62.2±0.59 | 56.4±0.08 | 45.7±0.35 | $N_{batch} \times 2$ | $N_{batch} \times 2$ |
| + Purif. w/ SOAP (5 Iter) | 30.8±0.12 | 47.4±1.07 | 47.7±0.25 | 36.6±8.80 | 40.7±8.35 | 10.7±0.01 | 57.0±7.43 | 64.1±0.09 | 49.1±3.22 | 45.2±23.8 | $N_{batch} \times 6$ | $N_{batch} \times 6$ |
| ABLATION STUDY OF DIFFERENT SALIENCY INDICATORS (IND.). | | | | | | | | | | | | |
| + Pixel-Sali. Ind. | 44.2±0.02 | 49.0±0.03 | 50.0±0.09 | 43.0±0.06 | 46.5±0.05 | 28.8±1.42 | 65.6±0.02 | 66.0±0.01 | 59.3±0.05 | 54.9±0.38 | $N_{batch} \times 2$ | $N_{batch} \times 2$ |
| + Feature-Sali. Ind. | 44.1±0.04 | 49.1±0.04 | 49.9±0.03 | 43.4±0.06 | 46.6±0.04 | 14.3±11.6 | 65.4±0.05 | 65.8±0.02 | 58.9±0.06 | 51.1±2.94 | $N_{batch} \times 2$ | $N_{batch} \times 2$ |
| + Logit-Sali. Ind. | 44.9±0.05 | 49.5±0.00 | 49.9±0.10 | 44.6±0.06 | 47.2±0.05 | 60.1±0.08 | 65.3±0.06 | 65.8±0.02 | 59.6±0.01 | 62.7±0.04 | $N_{batch} \times 2$ | $N_{batch}$ |
| ABLATION STUDY OF DIFFERENT TYPES OF RETRIEVAL (RET.). | | | | | | | | | | | | |
| + ID Ret. ($N_{mb} = 1k$) | 42.6±0.30 | 48.2±0.25 | 48.3±0.22 | 43.1±0.18 | 45.5±0.24 | 30.0±17.9 | 64.4±0.10 | 63.3±0.49 | 58.2±0.08 | 54.0±4.65 | $N_{batch} \times 2$ | $N_{batch}$ |
| + OOD Ret. ($N_{mb} = 1k$) | 44.9±0.05 | 49.5±0.00 | 49.9±0.10 | 44.6±0.06 | 47.2±0.05 | 60.1±0.08 | 65.3±0.06 | 65.8±0.01 | 59.6±0.01 | 62.7±0.04 | $N_{batch} \times 2$ | $N_{batch}$ |
| + IB Ret. ($N_{mb} = 0$) | 44.4±0.03 | 49.3±0.08 | 50.0±0.04 | 43.6±0.06 | 46.8±0.05 | 31.8±1.27 | 65.5±0.03 | 65.8±0.02 | 59.1±0.06 | 55.5±0.35 | $N_{batch} \times 2$ | $N_{batch}$ |
| ABLATION STUDY OF DIFFERENT SAMPLE PURIFICATION SCOPES. | | | | | | | | | | | | |
| + Purif. All Samps. | 44.9±0.05 | 49.5±0.00 | 49.9±0.10 | 44.6±0.06 | 47.2±0.05 | 60.1±0.08 | 65.3±0.06 | 65.8±0.01 | 59.6±0.01 | 62.7±0.04 | $N_{batch} \times 2$ | $N_{batch}$ |
| + Purif. Mali. Samps. | 45.0±0.03 | 49.1±0.02 | 49.6±0.08 | 44.3±0.06 | 47.0±0.05 | 11.1±9.42 | 64.7±0.04 | 65.3±0.06 | 59.0±0.07 | 50.0±2.40 | $N_{batch} \times 2$ | $N_{batch}$ |

the need to construct a memory bank, reducing storage overhead. But it also restricts the scope of benign sample retrieval and degrades the performance of PTTA. In Fig. 7, we compare the effects of constructing memory banks of different sizes on the application of PTTA to Tent, ETA, DeYO, and CPL. Here, a maximum length of the memory bank (MB) equal to $0$ indicates that we use In-Batch retrieval for PTTA. The experimental results show that a memory bank of an appropriate size, such as the default maximum queue length of $1,000$, can achieve higher performance. While an extremely large-length memory bank will lead to performance degradation. Due to the impact of the dynamically changing environment in TTA tasks, storing historical samples adjacent to the current time step $t$ is the best choice.

### E.7. Comparison of Retrieving Top-$K$ Benign Samples

In Sec. 3.3, we retrieve the top-$K$ samples with the largest saliency distance from either the memory bank or within the mini-batch for Mixuping with malicious samples. We uniformly set $K = 1$, and the $\lambda$ used for calculating the linear interpolation in Mixup is set as $\lambda = 1/(K + 1)$. Intuitively, we think that the larger the value of $K$, the less the components of malicious samples retained in the mixed samples, and the less damage to PTTA. However, we compare the experimental results using $K = 1, 2$ and $3$, as shown in Table 7. The ablation experiment results show that the value of $K$ does not affect the performance of PTTA. On the contrary, a larger value of $K$ will dilute the information of malicious samples in the purified samples. When the value of $K$ approaches infinity, it can be approximately considered that the malicious samples have been completely filtered out, so that DNNs will not learn the potentially useful information of malicious samples. Therefore, we default to set $K = 1$ to maximize the retention of the original information in the purified samples.

### E.8. Single-Sample Test-Time Adaptation

To verify the performance of our proposed PTTA under the single-sample TTA setting, where the batch size equals 1, we conduct experiments on single and continual TTA tasks. Single-sample TTA requires that TTA algorithms learn information about the target data distribution from a single test sample and ensure that DNNs do not suffer from severe interference during online adaptation, leading to over-fitting and catastrophic forgetting. This is a more challenging task compared to multi-sample TTA and is closer to real-world applications. The experimental results are shown in Table 20. The selection-free Tent exhibits a significant performance degradation in continual tasks, while the application of PTTA can alleviate this problem. Meanwhile, the application of PTTA in sample-selection-based TTA methods can also enhance the performance of the original algorithms.

### E.9. Purification via FGSM, SOAP, and DDA

As stated in Sec. 3.2, motivated by Adversarial Purification (Tang & Zhang, 2024), we propose PTTA. A common approach in adversarial purification is based on FGSM (Goodfellow et al., 2014). Specifically, the partial derivative of the objective function with respect to pixels is used to determine a fixed step size noise superimposed to the original image. SOAP (Shi et al., 2021) is another adversarial purification method that iteratively superimposes purification noise onto the original

Table 20: Detailed experimental results (top-1 classification accuracy (%)) on single and continual TTA tasks with the batch size equals 1. We highlight the highest accuracy in **bold** and the second best as underline.

| | NOISE | | | BLUR | | | | WEATHER | | | | DIGITAL | | | | |
| METHOD | GAUSS. | SHOT | IMPUL. | DEFOC. | GLASS | MOTION | ZOOM | SNOW | FROST | FOG | BRIT. | CONTR. | ELASTIC | PIXEL | JPEG | AVG. |
|---|---|---|---|---|---|---|---|---|---|---|---|---|---|---|---|---|
| | | | | | | | SINGLE | | | | | | | | | |
| NOADAPT | 16.8±0.00 | 12.0±0.00 | 16.5±0.00 | 29.2±0.00 | 23.6±0.00 | 33.9±0.00 | 27.3±0.00 | 15.8±0.00 | 26.6±0.00 | 47.5±0.00 | 55.4±0.00 | 44.3±0.00 | 31.0±0.00 | 45.1±0.00 | 49.1±0.00 | 31.6±0.00 |
| SAR | 45.2±0.08 | 43.0±0.52 | 46.2±0.17 | 53.6±0.08 | 50.6±0.24 | 57.5±0.06 | 53.2±0.13 | 58.8±0.17 | 55.8±3.97 | 68.8±0.21 | 74.8±0.34 | 65.7±0.21 | 58.3±0.24 | 68.9±0.16 | 66.4±0.01 | 57.8±0.24 |
| TENT | 44.5±0.13 | 43.0±0.03 | 45.5±0.16 | 52.5±0.20 | 48.1±0.13 | 55.5±0.17 | 50.7±0.10 | 16.1±1.21 | 17.5±2.37 | 66.6±0.03 | 75.0±0.19 | 64.8±0.07 | 53.6±0.24 | 66.9±0.18 | 64.7±0.16 | 51.0±0.24 |
| + PTTA | 44.7±0.13 | 43.0±0.12 | 46.2±0.12 | 52.4±0.26 | 48.0±0.19 | 56.0±0.21 | 51.9±0.22 | 15.4±0.53 | 26.3±4.66 | 67.7±0.13 | 75.2±0.03 | 65.6±0.08 | 54.2±0.23 | 67.5±0.10 | 64.4±0.10 | 51.9±0.35 |
| ETA | 50.1±0.33 | 49.1±0.19 | 51.2±0.26 | 56.7±0.05 | 55.7±0.16 | 61.1±0.09 | 57.8±0.11 | 64.1±0.20 | 63.8±0.13 | 72.1±0.13 | 77.4±0.04 | 67.5±0.04 | 64.6±0.23 | 71.8±0.08 | 69.1±0.08 | 62.1±0.04 |
| + PTTA | 50.9±0.31 | 49.7±0.16 | 51.9±0.12 | 57.3±0.10 | 56.1±0.18 | 61.6±0.07 | 58.9±0.07 | 65.0±0.09 | 64.6±0.11 | 72.8±0.10 | 77.6±0.11 | 68.3±0.10 | 65.1±0.23 | 72.3±0.10 | 69.2±0.12 | 62.7±0.04 |
| EATA | 34.5±2.65 | 30.1±2.82 | 35.4±2.34 | 43.8±1.52 | 39.4±1.77 | 46.3±1.45 | 41.6±1.61 | 36.1±5.02 | 41.6±3.67 | 61.4±1.52 | 66.0±2.15 | 61.2±1.16 | 46.1±2.01 | 59.8±1.71 | 59.2±1.05 | 46.8±2.16 |
| + PTTA | 35.5±2.13 | 31.6±2.46 | 36.5±2.01 | 44.7±1.41 | 40.4±1.63 | 47.1±1.31 | 42.9±1.46 | 40.1±4.44 | 44.6±3.12 | 62.4±1.41 | 67.5±1.76 | 62.2±0.90 | 47.3±1.89 | 60.7±1.53 | 59.5±1.00 | 48.2±1.89 |
| DEYO | 55.1±0.10 | 55.6±0.21 | 56.4±0.16 | 58.9±0.07 | 59.3±0.16 | 64.3±0.03 | 39.8±13.74 | 68.0±0.03 | 66.5±0.19 | 73.7±0.18 | 78.4±0.09 | 68.3±0.12 | 68.8±0.03 | 74.0±0.03 | 71.0±0.06 | 63.9±0.93 |
| + PTTA | 55.9±0.08 | 56.3±0.22 | 56.9±0.06 | 59.5±0.12 | 60.1±0.11 | 64.9±0.08 | 63.3±0.14 | 68.9±0.13 | 67.2±0.10 | 74.3±0.04 | 78.5±0.15 | 69.2±0.14 | 69.4±0.07 | 74.3±0.09 | 71.2±0.18 | 66.0±0.05 |
| CPL | 34.0±0.14 | 30.6±0.06 | 32.7±0.12 | 42.5±0.23 | 36.4±0.05 | 43.9±0.28 | 40.2±0.23 | 44.0±0.11 | 42.6±0.15 | 54.7±0.04 | 65.0±0.21 | 54.0±0.12 | 42.9±0.12 | 54.2±0.08 | 54.6±0.06 | 44.8±0.05 |
| + PTTA | 35.4±0.13 | 33.2±0.23 | 36.1±0.30 | 45.5±0.20 | 39.7±0.12 | 46.8±0.05 | 43.8±0.17 | 49.2±0.22 | 50.5±0.19 | 59.2±0.17 | 69.9±0.22 | 58.8±0.21 | 46.8±0.07 | 57.7±0.17 | 56.8±0.13 | 48.6±0.03 |
| | | | | | | | CONTINUAL | | | | | | | | | |
| NOADAPT | 16.8±0.00 | 12.0±0.00 | 16.5±0.00 | 29.2±0.00 | 23.6±0.00 | 33.9±0.00 | 27.3±0.00 | 15.8±0.00 | 26.6±0.00 | 47.5±0.00 | 55.4±0.00 | 44.3±0.00 | 31.0±0.00 | 45.1±0.00 | 49.1±0.00 | 31.6±0.00 |
| SAR | 45.2±0.08 | 55.1±0.12 | 57.9±0.02 | 53.2±0.28 | 56.6±0.06 | 59.7±0.10 | 57.2±0.07 | 61.4±0.05 | 64.0±0.02 | 67.4±1.68 | 74.2±0.77 | 66.0±0.23 | 60.2±0.25 | 67.5±1.96 | 69.3±0.20 | 61.0±0.07 |
| TENT | 44.5±0.13 | 49.1±3.44 | 25.2±27.06 | 17.4±28.41 | 14.7±25.33 | 0.5±0.56 | 0.2±0.07 | 0.3±0.19 | 0.1±0.00 | 0.1±0.00 | 0.4±0.12 | 0.1±0.00 | 0.1±0.00 | 0.1±0.00 | 0.1±0.02 | 10.2±5.52 |
| + PTTA | 44.7±0.13 | 51.4±0.14 | 55.0±0.14 | 51.7±0.29 | 53.4±0.13 | 58.5±0.19 | 56.9±0.16 | 57.7±0.90 | 47.6±24.63 | 27.2±36.23 | 27.1±42.41 | 21.7±37.43 | 19.3±33.24 | 23.3±40.11 | 22.7±39.11 | 41.2±16.17 |
| ETA | 50.1±0.33 | 55.4±0.25 | 56.6±0.46 | 51.8±0.52 | 54.5±0.28 | 56.8±0.33 | 54.5±0.43 | 58.1±0.64 | 59.3±0.79 | 65.5±0.63 | 73.7±0.14 | 57.3±1.43 | 58.6±0.32 | 65.7±0.73 | 62.2±0.33 | 58.7±0.13 |
| + PTTA | 49.7±0.12 | 58.4±0.14 | 59.7±0.15 | 56.1±0.04 | 59.5±0.02 | 63.2±0.03 | 63.0±0.30 | 65.3±0.08 | 66.2±0.12 | 72.5±0.05 | 77.5±0.04 | 67.6±0.08 | 66.4±0.28 | 73.1±0.08 | 70.6±0.01 | 64.6±0.02 |
| EATA | 35.3±2.22 | 35.5±2.97 | 40.1±2.43 | 44.7±1.55 | 42.1±1.89 | 47.9±1.43 | 43.6±1.73 | 39.6±4.51 | 46.0±3.28 | 62.4±1.45 | 67.3±1.76 | 61.7±1.13 | 47.2±1.94 | 60.9±1.57 | 60.8±1.16 | 49.0±2.06 |
| + PTTA | 34.9±2.12 | 35.4±2.86 | 40.4±2.46 | 45.3±1.51 | 42.7±1.95 | 48.6±1.52 | 45.1±1.78 | 44.6±4.44 | 50.5±3.40 | 64.3±1.63 | 68.7±1.81 | 63.0±0.67 | 48.2±1.84 | 61.4±1.58 | 60.1±1.03 | 50.2±2.03 |
| DEYO | 55.1±0.02 | 59.0±0.12 | 59.4±0.12 | 55.2±0.13 | 58.2±0.25 | 61.3±0.29 | 25.0±4.93 | 63.4±0.37 | 64.1±0.20 | 70.3±0.21 | 76.9±0.10 | 62.7±2.91 | 67.9±0.29 | 71.9±0.12 | 69.6±0.07 | 61.3±0.30 |
| + PTTA | 55.3±0.09 | 60.3±0.06 | 60.7±0.07 | 57.9±0.11 | 61.1±0.08 | 65.0±0.23 | 64.6±0.32 | 67.6±0.31 | 67.2±0.02 | 73.2±0.17 | 78.0±0.06 | 68.2±0.09 | 69.8±0.18 | 74.2±0.17 | 71.5±0.15 | 66.3±0.05 |
| CPL | 34.0±0.14 | 39.9±0.01 | 46.5±0.07 | 43.0±0.26 | 43.7±0.12 | 51.5±0.07 | 47.6±0.12 | 50.9±0.19 | 56.2±0.02 | 61.3±0.04 | 75.2±0.04 | 56.3±0.12 | 53.4±0.15 | 63.6±0.08 | 64.8±0.02 | 52.5±0.04 |
| + PTTA | 35.4±0.13 | 42.6±0.03 | 49.9±0.09 | 47.6±0.15 | 49.0±0.06 | 55.2±0.04 | 52.3±0.04 | 55.7±0.04 | 61.8±0.06 | 67.2±0.07 | 75.8±0.04 | 62.2±0.18 | 55.6±0.14 | 66.1±0.04 | 66.2±0.04 | 56.2±0.02 |

image over multiple cycles (5 iterations by default). In Table 19, we compare with the performance of using Adversarial Purification to purify test samples for TTA, including FGSM and SOAP. The results show that the method based on Adversarial Purification is not suitable for existing TTA methods and may even disrupt the original algorithm. Since adding noise directly to pixels would damage the semantic information in the original image, causing the model to fit the noise components and ignore the objects themselves, resulting in biases in DNNs. Therefore, the sample purification method based on natural image Mixup achieves better accuracy and stability. Also, thanks to the data augmentation of MixUp, it can improve the calibration of DNNs and reduce over-confident predictions. We also compare with DDA (Gao et al., 2023), a diffusion-based image purification method. Due to time constraints, we only conduct experiments on Gaussian noise and Defocus blur corruptions, each requiring $4 \times 4090$ GPUs and approximately 45 hours. The results in Table 21 demonstrate that diffusion-based DDA is not consistently effective. More importantly, DDA incurs significantly higher runtime and computational overhead compared to existing TTA algorithms, limiting its applicability.

Table 21: The comparison of PTTA with purification via FGSM, SOAP, and Diffusion-based DDA.

| | RESNET50 | | | | | | VIT-B/16 | | | | | |
| | ETA | | DEYO | | CPL | | ETA | | DEYO | | CPL | |
| METHODS | GAUSS | DEFCS | GAUSS | DEFCS | GAUSS | DEFCS | GAUSS | DEFCS | GAUSS | DEFCS | GAUSS | DEFCS |
|---|---|---|---|---|---|---|---|---|---|---|---|---|
| BASE ALGORITHM | 36.0 | 33.4 | 36.5 | 33.9 | 29.9 | 27.8 | 53.3 | 57.6 | 54.4 | 58.2 | 42.4 | 48.6 |
| + PURIFY W/ FGSM | 36.1 | 31.5 | 36.6 | 32.0 | 30.2 | 28.1 | 53.3 | 52.8 | 54.3 | 54.0 | 42.7 | 46.8 |
| + PURIFY W/ SOAP | 35.5 | 30.0 | 36.4 | 31.6 | 26.9 | 9.16 | 53.8 | 48.3 | 54.7 | 55.0 | 45.1 | 35.8 |
| + PURIFY W/ DDA | **44.3** | 14.9 | **44.3** | 14.9 | **44.3** | 14.9 | **60.0** | 30.2 | **60.0** | 30.2 | **60.0** | 30.2 |
| + PTTA | 37.4 | **35.4** | 37.8 | **35.1** | 32.1 | **29.8** | 54.0 | 58.2 | 55.0 | 58.8 | 42.8 | **50.0** |

**Comparison of different purification scopes.** In Table 18 and 19, we compare purifying only malicious test samples, purifying only benign samples, and purifying all test samples. The results of comparative experiments show that purifying all test samples achieves higher accuracy. Through learning from more augmented samples using Mixup, DNNs can achieve better calibration. This helps reduce the over-confident predictions for out-of-distribution (OOD) samples and mitigates the risk of disrupting the model's optimization. From another perspective, in TTA tasks, OOD samples are malicious relative to in-distribution samples, due to the poor generalization of DNNs. Therefore, using samples that are benign to DNNs to purify all OOD samples can effectively alleviate the problem of degradation in the model's prediction performance.

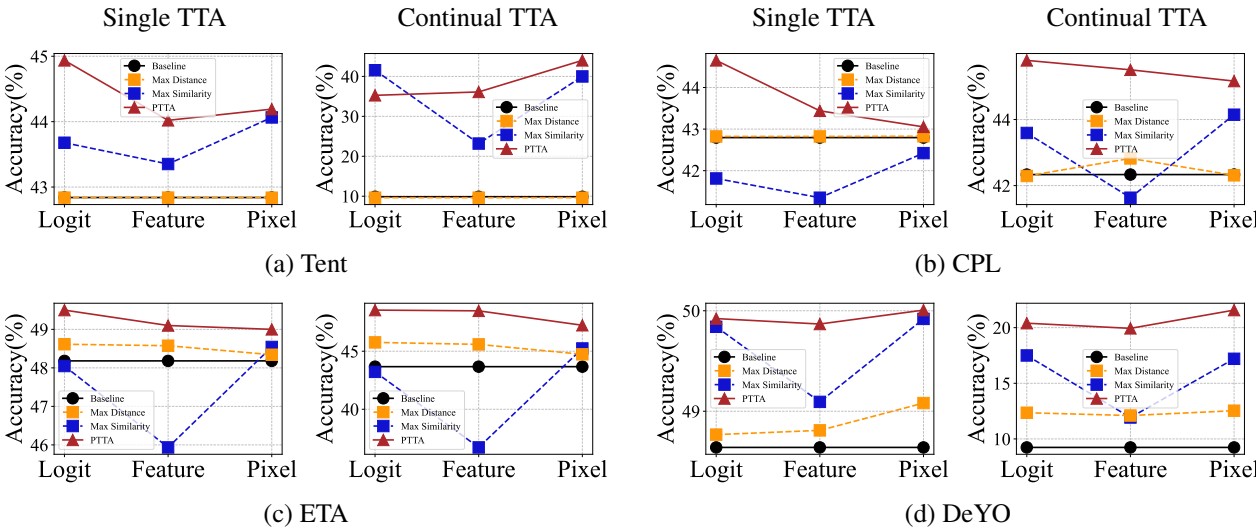

Figure 12: Comparison of maximum saliency distance, maximum direct cosine distance, and maximum direct cosine similarity at the logit, feature, and pixel levels

### E.10. Saliency Distance vs. Direct Cosine Distance

In Sec. 3.2, we propose using saliency distance as the metric for retrieving benign samples. Matching a malicious sample with the sample having the maximum saliency distance has a maximum opposite impact on the objective function. Another metric for retrieving benign samples is to directly calculate the cosine values between the logits or features output by the model, or image pixels for two samples $x_i$ and $x_j$, *i.e.*, $\text{Cosine}(f_\theta(x_i), f_\theta(x_j))$. Then, either the maximum cosine distance or the maximum cosine similarity is taken as the metric for retrieving benign samples. In Fig. 12, we compare the schemes of using maximum saliency distance, maximum cosine distance, and maximum cosine similarity as the metrics for retrieving benign samples. The results show that the benign samples retrieved by the maximum cosine distance cannot fully match the malicious samples, and thus cannot improve the performance of the original TTA algorithms. Furthermore, the retrieval based on the maximum cosine similarity will obtain samples that share similarities with malicious samples in terms of the objective function, and thus cannot achieve the purpose of purifying malicious samples. Therefore, the retrieval scheme based on our saliency distance has universality for logits, features, and pixels, as well as generality for different DNNs.

### E.11. Process of Continual Test-Time Adaptation

Fig. 13 visualizes the intermediate processes of two Test-Time Adaptation (TTA) algorithms, Tent (top) and DeYO (bottom), along with their PTTA-applied versions. Distribution shifts caused by abrupt corruption transitions induce oscillations in Deep Neural Network (DNN) optimization, undermining the effectiveness of TTA algorithms. Visualizations empirically validate that PTTA enhances DNN robustness against dynamically changing distributions and improves generalization on out-of-distribution samples by introducing Mixup.

## F. Limitations and Future Work

The saliency indicator we proposed is beneficial for encoding benign and malicious samples. And the saliency distance built upon it is an excellent metric for retrieving benign samples that have an opposite contribution to the objective function compared to malicious samples for the purpose of purification. However, the saliency indicator is not good enough. Developing an indicator with robust representation against arbitrary distribution shifts will be our future work.

We primarily validate the effectiveness of PTTA on vision models in the context of Test-Time Model Adaptation. Since the malicious sample hazards persist in a wide variety of machine learning and deep learning tasks, it is an interesting future research direction to extend purification strategies and PTTA to various tasks and applications.

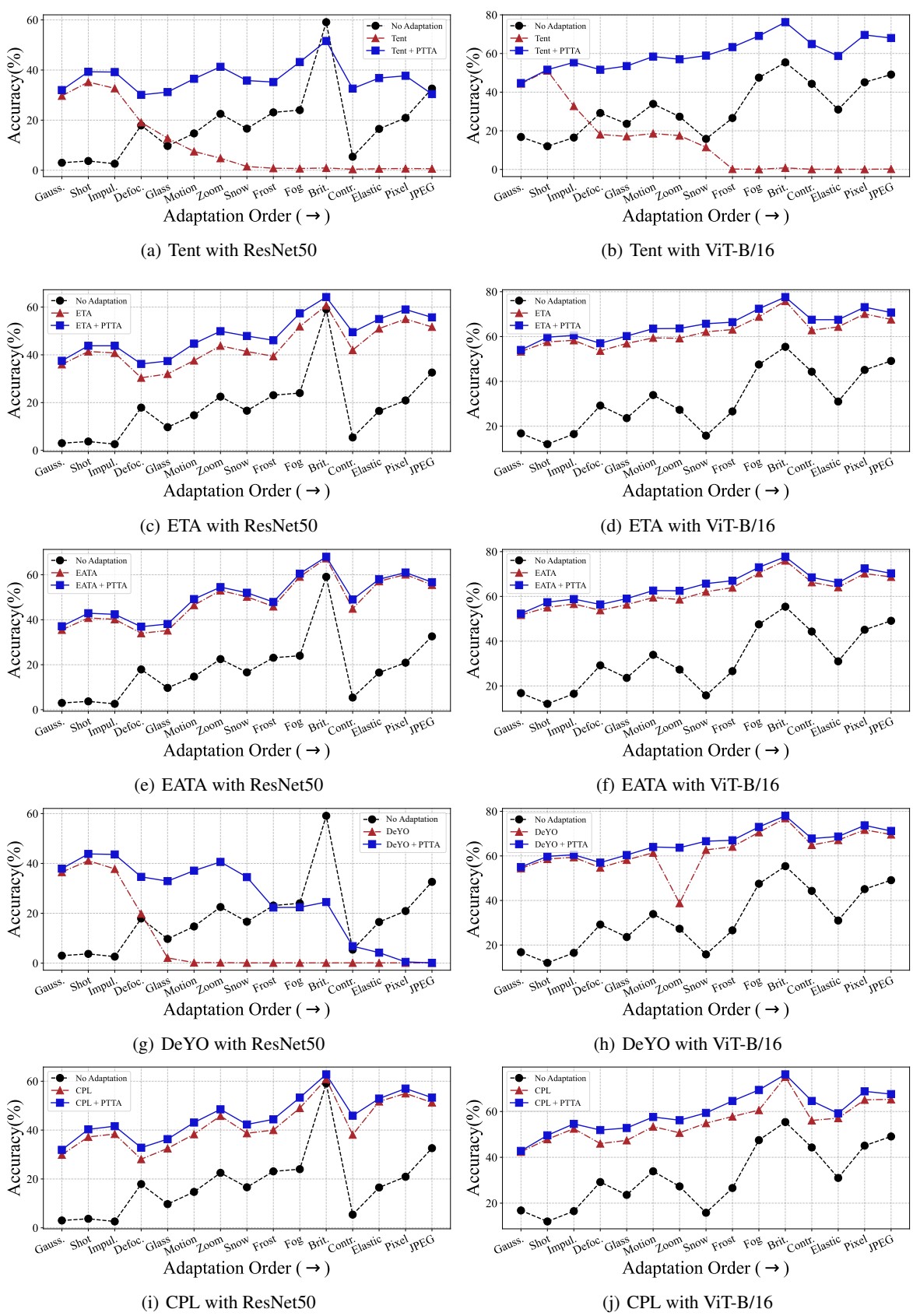

Figure 9: Detailed experimental results (top-1 classification accuracy (%)) on continual TTA task.

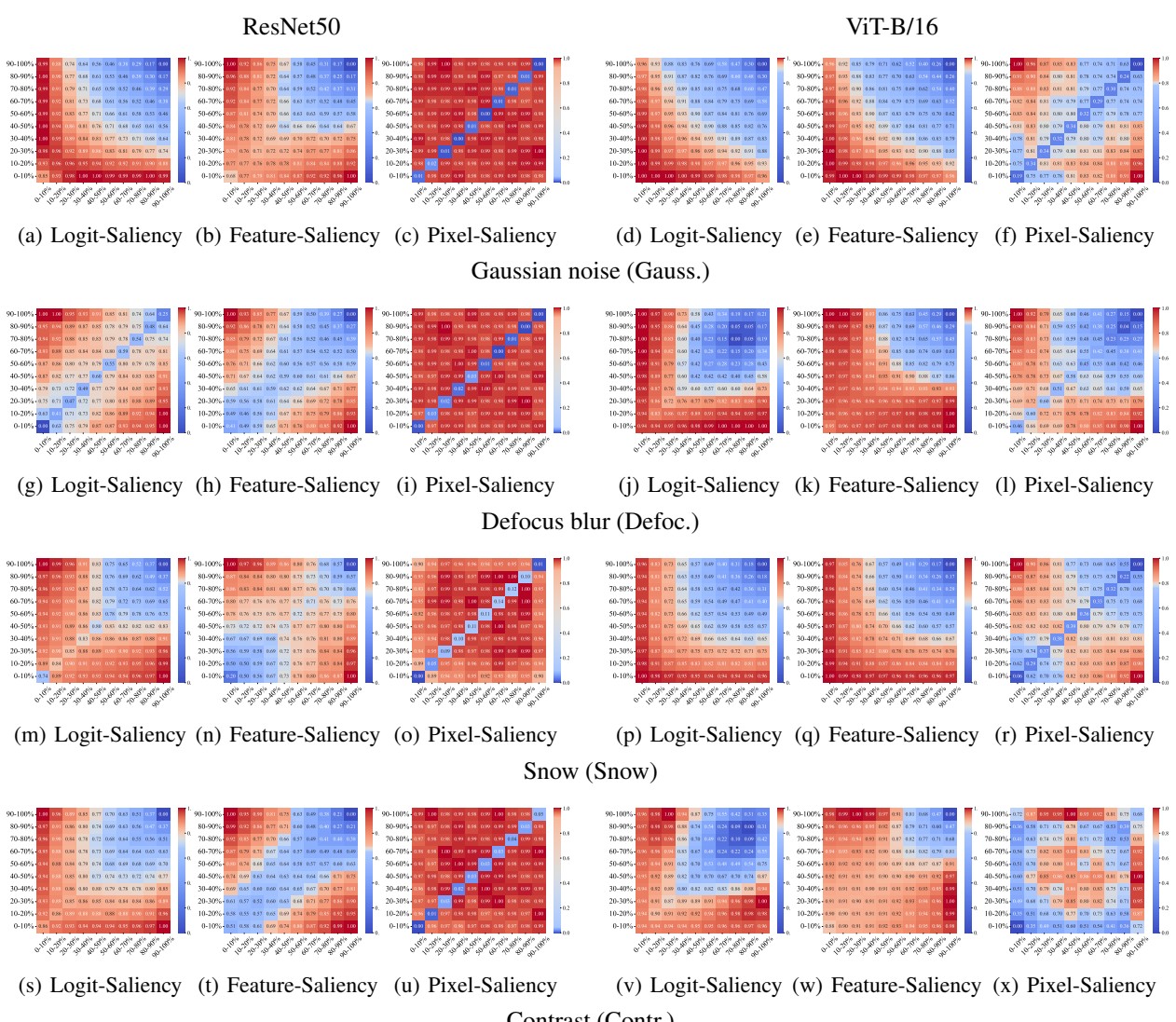

ResNet50 · ViT-B/16

(a) Logit-Saliency (b) Feature-Saliency (c) Pixel-Saliency   (d) Logit-Saliency (e) Feature-Saliency (f) Pixel-Saliency

Gaussian noise (Gauss.)

(g) Logit-Saliency (h) Feature-Saliency (i) Pixel-Saliency   (j) Logit-Saliency (k) Feature-Saliency (l) Pixel-Saliency

Defocus blur (Defoc.)

(m) Logit-Saliency (n) Feature-Saliency (o) Pixel-Saliency   (p) Logit-Saliency (q) Feature-Saliency (r) Pixel-Saliency

Snow (Snow)

(s) Logit-Saliency (t) Feature-Saliency (u) Pixel-Saliency   (v) Logit-Saliency (w) Feature-Saliency (x) Pixel-Saliency

Contrast (Contr.)

Figure 10: Saliency Distance (normalized to the range of $0 \sim 1$) among test samples sorted in ascending order of predicted entropy and split according to percentages. We visualize the logit-, feature-, and pixel-saliency distances of ResNet50 and ViT-B/16 for $4$ types of image corruptions: Gauss., Defoc., Snow, and Contr..

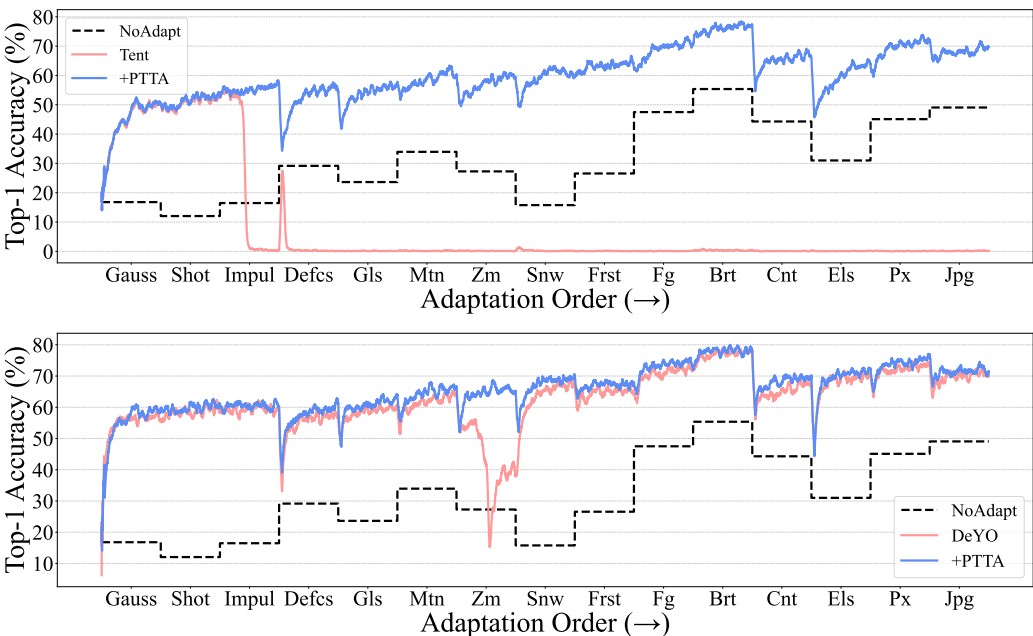

Figure 13: Visualization of intermediate Continual Test-Time Adaptation processes of Tent (upper) and DeYO (lower), along with their PTTA-applied versions. The figure intuitively demonstrates the impact of abrupt distribution shifts on DNN optimization trajectories.

