# OpenReview forum: "PTTA: Purifying Malicious Samples for Test-Time Model Adaptation"
_ICML.cc/2025/Conference — ICML 2025 poster_

### Official Review · Reviewer_yigZ · 2025-02-19

**Overall Recommendation:** 2

**Summary:**

The paper presents PTTA, a plug-and-play method for purifying malicious (unhelpful) samples for test-time adaptation.
PTTA selects benign samples by comparing the samplewise gradients.
Instead of simply filtering out malicious samples, PTTA transforms them into benign samples via Mixup with benign samples.
PTTA results in high accuracy improvements over various scenarios.

## update after rebuttal

The rebuttal addressed some of my concerns, including insufficient experiment results and computational complexity, so I raised the score from 1 to 2. However, I still worry (1) the problem (malicious sample hazard) is straightforward and (2) the solution lacks novelty.

**Claims And Evidence:**

1. The claim on the entropy-accuracy relationship is straightforward, but the literature (e.g., EATA) has partially discussed it.
1. The paper's key claims about PTTA's effectiveness are generally supported by the experimental evidence.

**Essential References Not Discussed:**

1. The paper does not discuss SoTTA [a] as a baseline, which robustly adapts to noisy data streams.


[a] Gong, Taesik, et al. "SoTTA: Robust Test-Time Adaptation on Noisy Data Streams." NeurIPS 2023.

**Experimental Designs Or Analyses:**

The paper experimented with full TTA (single) and continuous TTA (continuous) settings with various datasets, including ImageNet-C. Baselines include TENT, EATA, DeYO, and CPL.

Issues:

1. It is not specified why PTTA is not applied to CoTTA and SAR in Tables 2 and 3.

1. TENT is missing in Table 2.

1. It is unclear why ETA is used as the baseline in Table 1 instead of EATA.

**Methods And Evaluation Criteria:**

1. Please consider revising the manuscript regarding the methods. The current manuscript lists various approaches for saliency indicators, benign sample retrieval, and purification methods. It is unclear which version PTTA is using in the main experiments. Please improve the claims to support the best-working approach.

1. The evaluation setting of datasets aligns with standard benchmarks in the TTA field.

1. The experiment does not include the evaluation of computational overhead (e.g., memory, latency) with and without PTTA.

**Other Comments Or Suggestions:**

1. Please thoroughly check the citations. A few citations lack published years (e.g., Lee et al., Chen et al.).

**Other Strengths And Weaknesses:**

Minor Weakness
1. The technical novelty of sample purification is just a simple mixup technique, limiting the novelty.
1. Using source training data for in-distribution retrieval might limit the applicability.

**Questions For Authors:**

1. As stated in the abstract, sample purification is the main novelty of the paper, but purification is just a simple Mixup, limiting its novelty.

1. It is unclear which method will be the final method for PTTA. The current manuscript lists the potential methods and shows the ablation study. The writing should be improved to better convey the current method's importance.

1. The experiment is inconsistent across the evaluations. Please answer the concerns in the experimental setting.

1. Please report the computational overhead compared to the baselines (with and without PTTA).

**Relation To Broader Scientific Literature:**

PTTA can be applied in broader scenarios where test sample quality is low or includes noisy samples.

**Theoretical Claims:**

There are no novel theoretical claims that need to be checked.

---

> ### Author Rebuttal · Authors · 2025-04-01
>
> We sincerely appreciate the time and effort put into reviewing our paper and providing valuable feedback. We would like to address your questions below and provide a link to figures and tables.
>
> [link] https://anonymous.4open.science/r/PTTA/tab_fig.pdf
>
>
>
> > **R4Q1**: On the claim on entropy-accuracy relationship.
>
> We clarify that the entropy-accuracy relationship discussed in Sec. 3.1 is employed to demonstrate the **malicious sample hazards** in TTA methods. Our primary objective is to emphasize that directly utilizing malicious samples for TTA would undermine the stability of DNNs. Additionally, in contrast to EATA, we further demonstrate empirically that test data distributions with higher average entropy exhibit lower overall accuracy (Fig. 3 (down)), which is validated across both ResNet and ViT.
>
>
>
> > **R4Q2**: The key claims about PTTA are generally supported by the experimental evidence.
>
> We also provide a theoretical justification for PTTA. Please refer to **R2Q2** for details.
>
>
>
> > **R4Q3**: It is unclear which version of PTTA is used in the main experiments.
>
> We clarify that the default version of PTTA employed in the main experiments (**logit-saliency indicator & OOD retrieval**) is detailed in the **implementation part of Sec. 4.1** (lines 304–329). We hope it can address your concerns.
>
>
>
> > **R4Q4**: On the evaluation of computational overhead.
>
> We provide comparative analyses of **running times** between different TTA methods and their PTTA-applied versions, along with **quantified storage overhead** for PTTA's memory bank. Please refer to **R1Q1** for details.
>
>
>
> > **R4Q5**: Issues on experimental designs and analyses.
>
> We conduct experiments applying PTTA to CoTTA and SAR in Table C of [link], the results support our claims. We clarify that ETA, DeYO, and CPL were **selected as representative sample-selection-based TTA methods**. Due to EATA's foundational version ETA is **entirely based on sample selection**, so ETA is more suitable to be a baseline. Also, EATA use a Fisher regularizer measured in source domain, which is not source-free.
>
>
>
> > **R4Q6**: On some missing experimental results.
>
> We provide results for Tent on lifelong TTA task in Table C of [link], and MedBN+PTTA for adversarial defense on ImageNet-C in Table G of [link]. MedBN relies on a large batch size, which underperforms BN Adapt for adversarial defense. Therefore, BN Adapt+PTTA achieves the best performance on ImageNet-C for adversarial defense.
>
> SoTTA [1] was not selected as a baseline since its sample selection criterion aligns with CPL (Zhang et al., 2024), as both aim to maintain uniform class sampling. Additionally, CPL is a new state-of-the-art method in this field. However, we also provide a comparison with SoTTA [1] in Table C of [link]. We hope these studies can address your concerns.
>
> [1] Gong, et al. Sotta: Robust test-time adaptation on noisy data streams. In NeurIPS, 2023.
>
>
>
> > **R4Q7**: On the novelty of this paper.
>
> We emphasize that the main contributions of this paper include: 1) analysis of **malicious sample hazards** in TTA tasks, 2) a **saliency indicator** to effectively encode benign and malicious data, and 3) a **plug-and-play PTTA framework** for malicious sample purification.
>
> While PTTA employs Mixup technique, we clarify that **vanilla Mixup alone proves ineffective for TTA methods**. PTTA's effectiveness fundamentally stems from our proposed purification strategy and framework.
>
> Notably, PTTA uses OOD retrieval by default **without requiring any source data**.

---

> > ### Comment · Reviewer_yigZ · 2025-04-02
> >
> > Thank you for the rebuttal. I appreciate the updates on experimental settings and results.
> >
> > In terms of the novelty, an analysis of malicious sample hazards is not surprising. They can be easily inferred by the nature of entropy.
> >
> > Also, I found that PTTA results in about 40% increase in running time. Backpropagation on each sample for the saliency indicator must be a substantial computational burden. Could you specify the testing environment?
> >
> > ---
> >
> > Thank you for the quick response. I will increase the score and keep track of other reviewers' discussions to adjust the score further.

---

> > > ### Author Response · Authors · 2025-04-02
> > >
> > > Thank you for your professional feedback. We provide further explanations below.
> > >
> > >
> > > We would like to highlight that directly using malicious samples for TTA undermines the stability of DNNs, which **constitutes only one aspect of our analysis regarding malicious sample hazards**. Furthermore, we demonstrate that malicious samples are incorporated into mini-batches at uncertain proportions. **Existing sample selection criteria (employed in ETA, DeYO, CPL, etc.) fail to completely eliminate malicious samples from test data**. These criteria also exhibit **high sensitivity to threshold values** (typically treated as hyperparameters), where slight variations would lead to significant performance degradation, as illustrated in Fig. 2. Such limitations constrain the practical utility of sample-selection-based TTA methods across diverse tasks and scenarios, forming **another dimension of our analyzed malicious sample hazards** (lines 149-163). Consequently, rather than designing selection criteria to filter out malicious samples, we propose the purification strategy to transform malicious samples into benign ones.
> > >
> > >
> > >
> > > Regarding the running time, we clarify that the logit-saliency indicator (default in our PTTA) introduces **no additional gradient backpropagation** beyond base TTA algorithms. Instead, we explicitly compute this indicator through **Eq. 4**, which involves only **lightweight operations** (dot product and element-wise multiplication/subtraction) between two C-dimensional vectors (where C denotes the number of classes), making it **highly efficient**. Consequently, the logit-saliency indicator **converts a gradient-based method into a forward-only solution**, which is recognized by Reviewer iMq6. For comprehensive verification, Table 1 compares the forward/backward passes between base TTA methods and their PTTA-applied versions, while Appendix B.1 provides theoretical derivations for the logit-saliency indicator.
> > >
> > >
> > >
> > > Additionally, we specify the testing environment for evaluating the running time:
> > >
> > > **Hardware**: CPU: Intel® Xeon® Silver 4210 @ 2.20GHz | GPU: NVIDIA GeForce RTX 3090 | RAM: 256GB
> > >
> > > **Software**: PyTorch 1.9.0 | CUDA 11.1
> > >
> > > All experiments are conducted on a single GPU without Automatic Mixed Precision (AMP), with the following exceptions: 1) CoTTA \& CoTTA+PTTA utilize AMP by default, 2) CoTTA \& CoTTA+PTTA for ViT-B/16 employ dual GPUs via Distributed Data Parallel (DDP).
> > >
> > >
> > >
> > > We sincerely hope our clarifications above can improve your opinion of our work and can help you reconsider your score.
> > >
> > >
> > >
> > > Best Regards
> > >
> > >
> > > ----
> > >
> > > We are glad that our explanations are helpful in improving your opinion of our work. Thank you again for your expertise and invaluable feedback in enhancing the quality of our paper!
> > >
> > > Best Regards

---

### Official Review · Reviewer_iMq6 · 2025-03-11

**Overall Recommendation:** 3

**Summary:**

The paper introduces a method called Purifying Malicious Samples for Test-Time Model Adaptation (PTTA), a plug-and-play solution. Instead of filtering out, the authors identify that malicious samples in test data, though reflecting the data distribution, can undermine the stability of TTA algorithms. To address this, PTTA aims to transform malicious test samples into benign ones. It uses a saliency indicator to encode the impacts of benign and malicious samples on TTA, and retrieves benign samples with opposite contributions to the objective function compared to malicious samples. And then, the Mixup technique is employed for sample purification. Extensive experiments under various scenarios demonstrate that PTTA improves the performance of existing TTA methods.

**Claims And Evidence:**

Instead of filtering out, the authors identify that malicious samples in test data, though reflecting the data distribution, can undermine the stability of TTA algorithms. To address this, PTTA aims to transform malicious test samples into benign ones. Extensive experiments under various scenarios demonstrate PTTA improves the performance of existing TTA methods, validating the claim.

**Essential References Not Discussed:**

Comparing PTTA to diffusion-based malicious-to-benign sample transformation methods (Gao et al., 2024) would strengthen the evaluation.

[Gao’24] Gao, Jin, et al. "Back to the source: Diffusion-driven adaptation to test-time corruption." CVPR. 2023.

**Experimental Designs Or Analyses:**

Upon inspection of the experimental designs, I find them to be reasonable. The utilization of a diverse range of benchmarks offers comprehensive experimental support, effectively validating the proposed approach's efficacy.

**Methods And Evaluation Criteria:**

PTTA aims to transform malicious test samples into benign ones. It uses a saliency indicator to encode the impacts of benign and malicious samples on TTA, and retrieves benign samples with opposite contributions to the objective function compared to malicious samples. And then, the Mixup technique is employed for sample purification. Evaluations are also reasonable with experiments show that PTTA improves the performance of existing TTA methods under various scenarios.

**Other Comments Or Suggestions:**

N/A

**Other Strengths And Weaknesses:**

## Strengths:
1. The paper addresses a significant issue in TTA by focusing on malicious samples. PTTA's approach of transforming rather than discarding them, along with its non-sensitive threshold nature, makes it suitable for TTA scenarios, demonstrating innovative thinking.
2. The detailed experiments comprehensively validate PTTA across various settings like continual, lifelong, and adversarial, also with VLM backbones. Its excellent performance on multiple benchmarks and plug-and-play feature strongly prove its effectiveness, which is commendable.
3. The paper's explanation of the logit-level saliency indicator is profound. Converting a gradient-based method into a forward-only solution showcases technical sophistication.
4. The paper is written in a smooth and easily understandable manner.

## Weaknesses:
1. The paper lacks quantification of the memory and computational overhead comparison between the original methods and when PTTA is incorporated. For a plug-and-play method, especially in the TTA field, such quantification is important. It would help better understand its practicality and broad applicability in different resource-constrained environments.
2. Although PTTA has been well-validated in several scenarios, it would be better to verify whether it is effective when combined with teacher-student structure methods like COTTA. Also, comparing PTTA to diffusion-based malicious-to-benign sample transformation methods (Gao et al., 2024) would strengthen the evaluation.

**Questions For Authors:**

N/A

**Relation To Broader Scientific Literature:**

Most of previous TTA methods focused on filtering out malicious samples in the test data. The authors, however, pointed out that instead of simply discarding them, these malicious samples can potentially undermine the stability of TTA algorithms. Their approach is to transform these malicious samples into benign ones. Through comparisons with existing techniques such as the FGSM, the advantages of the proposed PTTA method are demonstrated.

**Theoretical Claims:**

The proof of logit-level saliency indicator is insightful, transforming a gradient-based method to a forward only solution.

---

> ### Author Rebuttal · Authors · 2025-04-01
>
> We sincerely appreciate the time and effort put into reviewing our paper and providing valuable feedback. We would like to address your questions below and provide a link to figures and tables.
>
> [link] https://anonymous.4open.science/r/PTTA/tab_fig.pdf
>
>
>
> > **R3Q1**: The paper lacks quantification of the memory and computational overhead comparison.
>
> We provide comparative analyses of **running times** between different TTA methods and their PTTA-applied versions, along with **quantified storage overhead** for PTTA's memory bank. Please refer to **R1Q1** for details.
>
>
>
> > **R3Q2**: It would be better to verify whether PTTA is effective when combined with teacher-student structure methods.
>
> We conduct experiments applying PTTA to two teacher-student structure methods, i.e., **CoTTA and CTDA [1]**, with results presented in Table C of [link]. These demonstrate that PTTA effectively enhances the performance of teacher-student structure TTA methods.
>
> [1] Wang, et al. Continual test-time domain adaptation via dynamic sample selection. In WACV, 2024.
>
>
>
> > **R3Q3**: On comparison between PTTA and diffusion-based malicious-to-benign sample transformation methods.
>
> We provide comparative experiments between PTTA and Diffusion-based DDA [2] when applied to base TTA methods (ETA, DeYO, and CPL) in Table D of [link]. The results show that Diffusion-based DDA **significantly degrades the performance of base TTA methods in some cases**, while incurring **prohibitive computational overhead** (requiring $4\times 4090$ GPUs and approximately 45 hours to complete an experiment for a single corruption type), rendering Diffusion-based DDA impractical for TTA tasks.
>
> [2] Gao, et al. Back to the source: Diffusion-driven adaptation to test-time corruption. In CVPR, 2023.

---

### Official Review · Reviewer_kYsv · 2025-03-12

**Overall Recommendation:** 3

**Summary:**

Existing TTA algorithms often focus on selecting benign samples for self-training, which leads to wasted test data. To address this, the authors propose PTTA, which uses a saliency indicator to identify benign samples with opposing effects on the objective function and combines them with malicious samples via Mixup. This strategy effectively leverages the information in malicious samples, improving online test accuracy. Extensive experiments across four TTA tasks, as well as classification, segmentation, and adversarial defense, validate the method’s effectiveness.

**Claims And Evidence:**

The claims made in the submission seem to be supported by clear and convincing evidence. However, in the ablation study section, the author states that “as K approaches infinity, it will dilute the proportion of original data information in purified samples.” This conclusion, however, cannot be directly drawn from Figure 9. Furthermore, what would be the result if the harmful samples were directly removed and only the corresponding benign samples were used in the Mixup? Rather than the fact that using more samples can improve TTA performance, I am more interested in understanding why the samples generated by Mixup on the original samples are effective.

**Essential References Not Discussed:**

Work [1] applies mixup to test-time training to prevent performance degradation in the main task and mitigate the mismatch problem. Work [2] improves sample utilization by performing negative learning through complementary labels. I believe these works share similarities with the ideas presented in the paper, and the authors should discuss it further to better highlight the key contributions of the work.

[1] Mixup for Test-Time Training. arXiv:2210.01640

[2] Continual Test-time Domain Adaptation via Dynamic Sample Selection. WACV 2024

**Experimental Designs Or Analyses:**

The experiments in the paper are thorough, covering four types of TTA tasks as well as classification, segmentation, and adversarial defense, with well-executed analysis of the results.

**Methods And Evaluation Criteria:**

The proposed methods and evaluation criteria are appropriate for the problem at hand.

**Other Comments Or Suggestions:**

NA

**Other Strengths And Weaknesses:**

NA

**Questions For Authors:**

See above.

**Relation To Broader Scientific Literature:**

Mixup is a commonly used data augmentation technique, and the author applies it to malicious samples in TTA to achieve higher sample utilization.

**Theoretical Claims:**

I did not check the correctness of proofs for theoretical claims.

---

> ### Author Rebuttal · Authors · 2025-04-01
>
> We sincerely appreciate the time and effort put into reviewing our paper and providing valuable feedback. We would like to address your questions below and provide a link to figures and tables.
>
> [link] https://anonymous.4open.science/r/PTTA/tab_fig.pdf
>
>
>
> > **R2Q1**: What's the results of purifying only begin samples?
>
> We present an ablation study in Table F of [link] comparing PTTA and direct removal of malicious (harmful) samples with purification of benign samples via Mixup. The results indicate that purifying only benign samples yields performance improvements of base TTA methods. Furthermore, additionally leveraging information from malicious samples via PTTA further enhances the performance.
>
>
>
> > **R2Q2**: Why is Mixup for PTTA effective?
>
> In general, Mixup helps enhance model robustness against adversarial attacks and improves generalization on out-of-distribution data [1]. It has also been demonstrated to refine the calibration of DNNs and mitigate overfitting [2].
>
> Next, we perform a Taylor expansion of the purification loss (Eq. 7) at mixup ratio $\lambda$ equals 0:
>
> $$
> \mathcal{L}\_{pur}(x^-, x^+) = - (\lambda y^- + (1-\lambda)y^+)^T \log f(\lambda x^- + (1-\lambda) x^+)
> = -(\lambda y^- + (1-\lambda)y^+)^T\log f(x^+) + \lambda (x^- - x^+) \nabla\_{x^+} \mathcal{L}\_{ce}(f(x^+)) + \mathcal{O}(\lambda^2),
> $$
>
> where $x^-$ is a malicious sample, $x^+$ is a benign sample, $y$ is an output vector, $f$ denotes the model, $\mathcal{L}\_{ce}$ is the Cross-Entropy loss.
>
> Minimizing $\mathcal{L}\_{pur}(x^-, x^+)$ reduces the loss of $-(\lambda y^- - (\lambda-1)y^+)^T\log f(x^+)$ while mitigating the interference of perturbations on $x^+$ to the predictions of $f(x^+)$, thereby enhancing model robustness. Large differences between $x^-$ and $x^+$ could enhance the influence of the first-order term in $\mathcal{L}\_{pur}(x^-, x^+)$, which verify the necessity and effectiveness of our proposed benign sample retrieval.
>
> Furthermore, we provide intermediate continual test-time adaptation process in Figure A of [link], empirically validating the effectiveness of using Mixup for PTTA.
>
> [1] Zhang, et al. How does mixup help with robustness and generalization? In ICLR, 2021.
>
> [2] Thulasidasan, et al. On mixup training: Improved calibration and predictive uncertainty for deep neural networks. In NeurIPS, 2019.
>
>
>
> > **R2Q3**: On the claim of $K$ in the ablation study.
>
> Due to the lack of clear trends in Fig. 9, please refer to Table 14 for detailed results, which demonstrate that as $K$ approaches infinity, it will dilute the proportion of original malicious data information in purified samples. Because we set $λ=1/(K+1)$, as $K$ approaches infinity, $\lambda$ approaches 0, causing $\mathcal{L}\_{pur}(x^-, x^+)$ to degenerate into $\mathcal{L}\_{ce}(f(x^+))$, thereby reducing PTTA to base TTA methods that remove all malicious data information.
>
>
>
> > **R2Q4**: On comparison with works [3] and [4].
>
> Work [3] mixes test samples with randomly selected training samples to perform an auxiliary task, which is designed for auxiliary test-time training and diverges significantly from the TTA framework.
>
> Work [4] employs negative learning to improve the utilization of malicious data. However, purification strategy-based PTTA outperform work [4] in TTA tasks, as validated in Table C of [link].
>
> [3] Zhang, et al. Mixup for Test-Time Training. arXiv:2210.01640.
>
> [4] Wang, et al. Continual test-time domain adaptation via dynamic sample selection. In WACV, 2024.

---

> > ### Comment · Reviewer_kYsv · 2025-04-02
> >
> > Thanks to the author’s reply, most of my concerns have been resolved, and I will adjust my score accordingly.

---

> > > ### Author Response · Authors · 2025-04-07
> > >
> > > Thank you for upgrading your score and the expertise and invaluable feedback in enhancing the quality of our paper!
> > >
> > > Best Regards

---

### Official Review · Reviewer_G5c6 · 2025-03-15

**Overall Recommendation:** 4

**Summary:**

This paper focuses on leveraging malicious samples during Test-Time Adaptation (TTA) to improve data utilization. The authors propose PTTA, a plug-and-play method that retrieves benign samples with maximal divergence from malicious samples and employs a Mixup strategy to purify malicious samples for TTA. PTTA demonstrates compatibility with existing TTA methods, and extensive experiments validate its effectiveness across multiple datasets, backbones, and tasks.


## update after rebuttal
My concerns have been addressed. I think this paper can be accepted now. This paper provides a valuable exploration of malicious sample utilization in TTA, with extensive experiments validating the effectiveness of the proposed method. In my opinion, this work is well-executed and deserves acceptance. To further strengthen the contribution, I would suggest extending the discussion that could provide deeper insights for readers.

First, it would be further improved by including a brief discussion with more TTA approaches. The paper could consider some of the recent TTA methods, such as FOA [1], ROID [2], MGTTA [3] and EATA-C[4]. Unlike this work, these methods do not perform purification on malicious samples but instead design general optimization strategies using all samples, achieving significant performance improvements as well. Second, some studies explore enhancing TTA performance from the perspective of improving TTA process. For example, improving entropy loss [5-6], refining batch normalization[7], and optimizing the inference process [9]. Third, the paper primarily discusses the application of TTA in image classification and semantic segmentation. It would be worthwhile to explore whether the proposed method can be extended to other domains and tasks, such as image super-resolution [9], video classification [10-11], and visual question answering [12].

[1] Test-Time Model Adaptation with Only Forward Passes, ICML 2024.
[2] Universal test-time adaptation through weight ensembling, diversity weighting, and prior correction, WACV 2024.
[3] Learning to Generate Gradients for Test-Time Adaptation via Test-Time Training Layers, AAAI 2025.
[5] TEA: Test-time Energy Adaptation, CVPR 2024.
[6] Decoupled Prototype Learning for Reliable Test-Time Adaptation, arXiv 2025.
[7] Unraveling batch normalization for realistic test-time adaptation, AAAI2024.
[8] Boost test-time performance with closed-loop inference, arXiv 2022.
[9] Efficient test-time adaptation for super-resolution with second-order degradation and reconstruction, NuerIPS 2023.
[10] Video Test-Time Adaptation for Action Recognition, CVPR 2023.
[11] Exploring Motion Cues for Video Test-Time Adaptation, ACM MM 2023.
[12] Test-time model adaptation for visual question answering with debiased self-supervisions, TMM 2023.

**Claims And Evidence:**

Yes

**Essential References Not Discussed:**

No

**Experimental Designs Or Analyses:**

Yes. The experimental design rigorously evaluates common TTA baselines and tasks, with comprehensive validation on diverse benchmarks.

**Methods And Evaluation Criteria:**

Yes

**Other Comments Or Suggestions:**

1.	Mixup Rationale: A brief theoretical or empirical justification for using Mixup in Sec. 3.3  would strengthen the methodology.
2.	The main text seems to lack a reference and introduction to Figure 1.

**Other Strengths And Weaknesses:**

Strengths

1.	The paper is well-structured and clearly written.

2.	PTTA is a simple yet practical plug-and-play approach, seamlessly integrating with existing TTA frameworks.

3.	Thorough ablation studies and evaluations across diverse scenarios (datasets, backbones, tasks) strongly support the method’s efficacy.

Weaknesses

1.	Lacks detailed computational complexity analysis. While PTTA improves performance, Table 1 indicates increased time (additional forward passes) and memory costs (memory bank). A critical analysis is missing: How do SOTA methods perform under comparable computational constraints? (For example, maintain a memory bank and randomly select benign samples for adaptation when encountering malicious samples.)

2.	Motivation of the proposed method is incomplete. The connection between adversarial purification and the proposed saliency indicator (Sec. 3.2) requires stronger justification. The rationale for linking gradient directions of entropy minimization to noise encoding remains unclear.

**Questions For Authors:**

See the comments above

**Relation To Broader Scientific Literature:**

This paper is closely related to the field of test-time adaptation, sample selection, and the use of malicious samples.

**Theoretical Claims:**

Yes. I have checked the proof of Logit-saliency indicator.

---

> ### Author Rebuttal · Authors · 2025-04-01
>
> We sincerely appreciate the time and effort put into reviewing our paper and providing valuable feedback. We would like to address your questions below and provide a link to figures and tables.
>
> [link] https://anonymous.4open.science/r/PTTA/tab_fig.pdf
>
>
>
> >  **R1Q1**: Lacks detailed computational complexity analysis.
>
> We provide comparative analyses of **running times** between different TTA methods and their PTTA-applied versions in Table A of [link], along with **quantified storage overhead** for PTTA's memory bank in Table B of [link]. Additionally, Table E of [link] compares state-of-the-art TTA methods with their PTTA-applied versions **under comparable computational constraints**. The results support PTTA's superiority.
>
>
>
> > **R1Q2**: Motivation of the proposed method is incomplete.
>
> Adversarial purification methods typically leverage the first-order partial derivatives of the objective loss function with respect to the image $x$, i.e., **the saliency information**: $\gamma = \xi \cdot \text{sign}(\nabla_x \mathcal{L}(f_\theta(x), y(x))$, as an unit vector in the image space for sample purification.
>
> Saliency information quantifies which individual pixels require the most modification to minimize the objective loss function. Consequently, **samples contributing similarly to the objective loss function exhibit aligned directions of their saliency information unit vectors**, resulting in small Cosine distances that serve as the *saliency indicator*.
>
> While Entropy Minimization (EM) is a widely adopted objective loss function in TTA methods, we use EM-derived saliency indicator for noise encoding. Notably, we validate in Table C of [link] that the saliency indicator based on the teacher-student consistency loss function (used in CoTTA and CTDA [1]) also achieves strong efficacy, demonstrating the **flexibility in choosing objective loss functions**.
>
> [1] Wang, et al. Continual test-time domain adaptation via dynamic sample selection. In WACV, 2024.
>
>
>
> > **R1Q3**: On a brief theoretical or empirical justification for using Mixup.
>
> We provide a brief theoretical and empirical justification for using Mixup in PTTA. Please refer to **R2Q2** for details.

---

> > ### Comment · Reviewer_G5c6 · 2025-04-03
> >
> > All my concerns have been adequately addressed.
> >
> > Regarding Question 1: While the method increases forward passes, the authors have now supplemented time cost analysis and performance comparisons under equivalent time constraints.
> >
> > Regarding Question 2: The motivation clarification in the methodology section has been strengthened with additional explanations and justifications.
> >
> > I currently have no further questions and recommend maintaining my original assessment.

---

> > > ### Author Response · Authors · 2025-04-07
> > >
> > > We are delighted to learn that all your concerns have been addressed. Thank you again for your expertise and invaluable feedback in enhancing the quality of our paper!
> > >
> > > Best Regards

---

### Decision · Program_Chairs · 2025-05-01

**Decision:**

Accept (poster)

**Comment:**

This paper introduces PTTA, a plug-and-play method for test-time adaptation (TTA) that purifies malicious test samples via saliency-guided Mixup, avoiding their discard and improving data utilization. Reviewers acknowledge its practical novelty and empirical effectiveness: PTTA consistently enhances baseline TTA methods (e.g., ETA, DeYO) across classification, segmentation, and adversarial defense tasks. The logit-saliency indicator and Mixup strategy are praised for simplicity and compatibility with existing frameworks.

However, concerns include: computational overhead (≈40% increase in runtime due to memory bank and forward passes), limited theoretical depth (e.g., reliance on Mixup’s heuristic benefits), missing references with discussion and missed comparisons with diffusion-based methods like DDA. Despite these caveats, PTTA’s robust experimental validation across diverse scenarios (4 TTA tasks, multiple backbones) and clear practical value justify its contribution. The authors are recommended to revise the paper carefully according to the reviewers' comments.